# MULTI-LCB: EXTENDING LIVECODEBENCH TO MULTIPLE PROGRAMMING LANGUAGES

**Maria Ivanova**[*2]     **Pavel Zadorozhny**[*1]     **Rodion Levichev**[*1]     **Ivan Petrov**[1]

**Pavel Adamenko**[1]     **Ivan Lopatin**[1]     **Alexey Kutalev**[1]     **Dmitrii Babaev**[1]

[1] GigaCode      [2] Yandex School of Data Analysis, Applied AI Institute

https://github.com/Multi-LCB/Multi-LCB

## ABSTRACT

LiveCodeBench (LCB) has recently become a widely adopted benchmark for evaluating large language models (LLMs) on code-generation tasks. By curating competitive programming problems, constantly adding fresh problems to the set, and filtering them by release dates, LCB provides contamination-aware evaluation and offers a holistic view of coding capability. However, LCB remains restricted to Python, leaving open the question of whether LLMs can generalize across the diverse programming languages required in real-world software engineering.

We introduce Multi-LCB, a benchmark for evaluating LLMs across twelve programming languages, including Python. Multi-LCB transforms Python tasks from the LCB dataset into equivalent tasks in other languages while preserving LCB's contamination controls and evaluation protocol. Because it is fully compatible with the original LCB format, Multi-LCB will automatically track future LCB updates, enabling systematic assessment of cross-language code generation competence and requiring models to sustain performance well beyond Python.

We evaluated 24 LLMs for instruction and reasoning on Multi-LCB, uncovering evidence of Python overfitting, language-specific contamination, and substantial disparities in multilingual performance. Our results establish Multi-LCB as a rigorous new benchmark for multi-programming-language code evaluation, directly addressing LCB's primary limitation and exposing critical gaps in current LLM capabilities.

## 1 INTRODUCTION

Large language models (LLMs) have recently demonstrated impressive capabilities in code-related tasks (Ridnik et al., 2024; Lozhkov et al., 2024; Roziere et al., 2023; Li et al., 2022; Nijkamp et al., 2022), powering applications such as AI-assisted programming, automated debugging, and code translation. To measure these abilities, benchmarks such as HumanEval (Chen et al., 2021), MBPP (Austin et al., 2021), and APPS (Hendrycks et al., 2021) have been widely adopted. However, these datasets suffer from well-documented limitations, including contamination from training corpora, narrow task scope, and weak correlation with human judgment. LiveCodeBench (LCB) (Jain et al., 2024) addresses these shortcomings by continuously curating competitive-programming problems, filtering them by release date, and enabling *contamination-aware, continuously updatable evaluation*. As a result, LCB has quickly become a standard benchmark for evaluating LLMs on code-generation tasks (Google DeepMind, 2025; DeepSeek, 2025).

Despite these strengths, LCB (Jain et al., 2024) evaluates only Python. While convenient, this limitation overlooks a central reality of software engineering: developers routinely work across diverse programming languages, each with its own syntax, semantics, and idiomatic practices. An LLM capable of solving problems exclusively in Python may perform poorly when C++ is required

---

*Equal contribution. Correspondence to Dmitrii Babaev: dmitri.babaev@gmail.com

for systems programming, Java for enterprise software, or JavaScript for web development. Current evaluations therefore leave open a critical question: *can LLMs generalize coding competence across multiple programming languages, or are they overfitted to Python?*

In this work, we introduce **Multi-LCB**, an extension of LCB (Jain et al., 2024) to twelve programming languages while preserving its contamination controls and evaluation protocol. Multi-LCB replicates every LCB task across all supported languages, enabling direct comparison of model performance on identical problems in different programming languages and updating automatically as LCB evolves. We evaluate 24 reasoning- and instruction-oriented LLMs on Multi-LCB and uncover key findings:

1. *Python is not always a reliable proxy for individual non-Python languages*. Our results reveal substantial and practically meaningful performance gaps across languages. In several cases, models that are stronger on Python do not retain their advantage in other languages.

2. *Python overfitting*. Models that perform strongly in Python often degrade sharply in other languages.

3. *Language-specific contamination*. Evidence of data leakage varies by programming language, reflecting uneven distribution in pretraining corpora.

4. *Substantial multi-programming-language disparities*. Models show large performance gaps across languages, with weaker results in statically typed or less prevalent languages.

Our main contributions are:

1. We extend LCB (Jain et al., 2024) to 12 programming languages without task loss, enabling direct comparison of LLM abilities to solve identical problems across different languages.

2. We provide a comprehensive evaluation of 24 instruction- and reasoning-oriented LLMs across these languages, revealing systematic multi programming languages performance gaps and evidence of language-specific contamination.

3. We publicly release all prompts, source code and experimental configurations to facilitate reproducibility and future research.

These results establish Multi-LCB as a rigorous benchmark for multi-programming-language code evaluation, directly addressing LCB's Python-only limitation and providing a foundation for developing more robust, programming language agnostic coding models.

## 2 RELATED WORK

**Single-language code benchmarks**. Early code-generation benchmarks evaluate functional correctness almost exclusively in Python. *HumanEval* (Chen et al., 2021) contains 164 hand-written problems, each defined by a natural language prompt, a fixed function signature, and hidden unit tests; tasks are short, single-function programs created specifically for evaluation rather than drawn from programming contests. *MBPP* (Austin et al., 2021) likewise offers small Python exercises aimed at introductory programming and interview practice. Subsequent datasets expanded scale and difficulty: *APPS* (Hendrycks et al., 2021) aggregates competition and interview style problems with hidden test suites, *CodeContests* (Li et al., 2022) compiles algorithmic contest tasks with official judge input/output data, and *CodeXGLUE* (Lu et al., 2021) provides a broad suite of generation, translation, and retrieval tasks. Despite their influence, these resources are static snapshots, lack release date filtering to prevent training set contamination and are therefore largely saturated, remain heavily Python centric, and do not enforce a unified STDIN/STDOUT protocol.

**Multi-programming-language benchmarks.** Several datasets extend code generation evaluation beyond Python. *MBXP* (Athiwaratkun et al., 2022) translates functional-format Python problems (e.g., HumanEval (Chen et al., 2021), MBPP (Austin et al., 2021)) by rewriting function signatures and regenerating unit tests for each language. Even a simple Python assertion like:

```
assert binomial_coeff(5, 2) == 10
```

must be expanded into multi-line Java test code. This translation must be repeated separately for every language and is sensitive to syntax and runtime differences. Concurrent work *MultiPL-E* (Cassano et al., 2023) similarly performs translation of HumanEval and MBPP (including their unit tests)

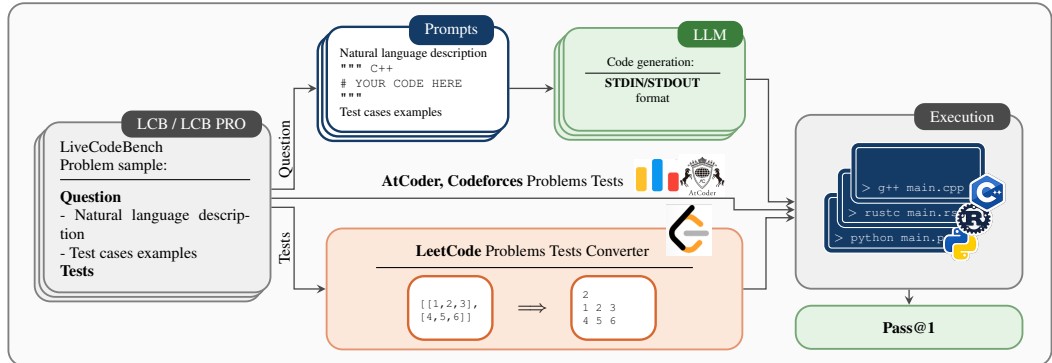

Figure 1: **Multi-LCB overview.** Top: LCB natural-language problem descriptions are wrapped into prompts specifying the target programming language and passed to the LLM for `STDIN/STDOUT` code generation. AtCoder and Codeforces problem tests are passed directly to the execution stage. Bottom: LeetCode problem tests are transformed through a dedicated test converter to produce equivalent `STDIN/STDOUT` inputs. The generated code is compiled or executed in the target programming language and evaluated using Pass@1.

into 19 programming languages. *HumanEval-XL* (Peng et al., 2024) similarly expands HumanEval to additional languages and provides a standardized execution harness while preserving the functional, unit-test format. Multi-LCB avoids this by keeping only the natural-language description and converting hidden tests into a language-agnostic `STDIN/STDOUT` format, for example:

```
Input:
5 2
Output:
10
```

Other projects broaden language coverage in different ways. *Ag-LiveCodeBench-X* (Boruch-Gruszecki et al., 2025) reuses a subset of LiveCodeBench tasks already in STDIN/STDOUT format and adds rarer targets such as Lua, R, Julia, OCaml, and Fortran. *xCodeEval* (Khan et al., 2023) likewise provides a unified multilingual execution framework and resembles our approach, but it draws exclusively from Codeforces problems and is not continuously updated. *McEval* (Chai et al., 2024) and *BigCodeBench* (Zhuo et al., 2024) once offered broad language coverage, but both are static and evaluate different task sets per language, hindering direct cross language comparison.

**Contamination-aware evaluation.** LiveCodeBench (LCB) (Jain et al., 2024) introduced release date filtering and continuous collection of Python problems from three major competitive programming platforms: LeetCode, AtCoder, and Codeforces (see Appendix D.1 for task statistics). By harvesting new tasks and filtering them by post-training release dates, LCB enables live, contamination aware evaluation of LLMs and has become a *de-facto standard* for robust single language (Python) code assessment (Comanici et al., 2025; Yang et al., 2025; Liu et al., 2024). A related effort, *EvoCodeBench* (Li et al., 2024), followed a similar evolving design but was not actively maintained and remained limited to Python. *Multi-LCB* builds directly on this foundation. It reuses the entire LCB (Jain et al., 2024) task pool and inherits its contamination controls.

## 3  BENCHMARK DESIGN

This section describes the approach, used to construct the Multi-LCB benchmark. Figure 1 illustrates the full pipeline. Please note, that although Multi-LCB is built on LCB, the same approach *can be applied to any dataset* with a comparable structure.

**Data Source.** Earlier versions of LCB supported several evaluation scenarios beyond code generation: self-repair, code execution, and test output prediction. But the latest releases (v5-v6) focus exclusively on *code generation*, the most widely benchmarked capability of modern LLMs. In this setting, a model receives a natural language problem statement with sample input/output pairs and must synthesize a program that passes all hidden test cases.

To construct Multi-LCB, we load the desired version of the LCB *code generation* dataset from Hugging Face, retrieving Python problems and their metadata. We convert every release of LCB code generation dataset without modification, preserving all tasks from three competitive-programming platforms: LeetCode, AtCoder, and Codeforces. Each task includes a natural language description, input/output examples, and contest release date for contamination-aware filtering. Test conversion is applied only to LeetCode's functional format tasks to ensure unified STDIN/STDOUT evaluation. Details about platforms and temporal distribution appear in the Appendix D.1.

**Conversion of functional format.** LCB supports two native task formats: **STDIN/STDOUT** (as in AtCoder and Codeforces), where a program reads from standard input and writes to standard output, and **Functional** (as in LeetCode), where a specific function is implemented and invoked by the evaluation system. Directly extending the functional format to a multi-programming language benchmark is challenging. Each LeetCode task provides Python starter code tightly coupled to its own testing harness. Producing equivalent starter code and call signatures for many target languages would require custom templates for every language, leading to an unsustainable and error-prone process. To overcome this limitation, we designed an **automatic conversion pipeline** that rewrites every Functional task into a unified STDIN/STDOUT format. This pipeline consists of two components: (1) prompt adaptation that reformats problem statements and examples for model input, and (2) test conversion that transforms all test cases for automated evaluation.

The pipeline first parses examples from the problem statement and reformats them into STDIN/STDOUT format for inclusion in model prompts. (see Appendix C.1). Separately, it converts all test cases (both public and hidden) from the original format to enable unified automated evaluation. This unification allows a single evaluation harness to handle both the original STDIN/STDOUT problems and the adapted functional tasks across all supported languages. Since the original benchmark is based on Python, tasks involving Python-specific behavior could theoretically appear. However, tasks on LeetCode, AtCoder, and Codeforces are authored by human experts and are intentionally designed to avoid language-specific ambiguities, as these platforms support many programming languages. Consequently, Multi-LCB requires no language-specific rewriting, and the tasks remain inherently language-agnostic. Moreover, in our manual inspection of approximately 500 tasks, we did not find any cases in which language-dependent features introduced inconsistencies. Note that tasks unsuitable for strict input/output grading, such as those admitting multiple valid answers or requiring explicit data structure construction, are already excluded in the official LCB dataset that we load, so Multi-LCB inherits this filtering without any additional intervention. The remaining tasks are grouped by I/O structure: **Scalar**: inputs and outputs are single, scalar values (e.g. integers, floats, booleans, or simple strings); **One-Dimensional**: involve one-dimensional arrays (lists) as input or output; **Two-Dimensional**: include exactly one two-dimensional array (matrix or jagged array) in the I/O. As a result, all functional tasks, including their examples and hidden tests, are consistently converted to STDIN/STDOUT format: lists are space-separated, and for 2D arrays the first line specifies the number of rows, followed by row-wise space-separated values. This conversion applies to both the examples shown to the models and all test cases used for evaluation.

**Code generation.** We adopt a zero-shot prompting strategy that follows the original LiveCodeBench protocol. For each task, the benchmark constructs a prompt with three components:

1. a `system message` instructs the model to act as an expert programmer in the target language (e.g., ``You are an expert Python programmer...'');

2. a `user message` provides the complete natural language problem statement with explicit STDIN/STDOUT specifications and input/output sample cases already provided in the original problem descriptions;

3. a code-block placeholder indicates where the solution must be written:

```
""" python
# YOUR CODE HERE
"""
### Answer: (use the provided format with backticks)
```

The code-block header is set to the target language (e.g., `cpp`, `java`, `python`) to ensure correct syntax highlighting and parsing.

Models are required to output only the complete program source that reads from the standard input and writes to the standard output. High-level zero-shot template prompts for both native AtCoder and CodeForces tasks and adapted LeetCode problems are included in the Appendix C for reference.

**Automatic Testing and Evaluation.** Correctness is assessed against a hidden suite of official test cases provided by the original contests. A program is marked correct only if it passes all tests without runtime errors or timeouts. For quantitative comparison we report Pass@1, the fraction of tasks for which the model's first generated solution passes every public and hidden test.

Together, these stages create a fully automated pipeline: a model receives a problem prompt, emits a candidate solution, the code is securely compiled and executed, and the output is graded against hidden tests – all without human intervention. This process preserves LCB's rigorous contamination controls while enabling direct, language-agnostic evaluation of code generation across the diverse set of languages supported by Multi-LCB. Note, that the same set of tasks is used across evaluations on different programming languages, hence task difference does not hinder the comparison of the multi-language model capabilities.

## 3.1 LANGUAGE SET AND MOTIVATION

This study evaluates multilingual code generation across major programming languages: C++, C#, Python, Java, Rust, Go, TypeScript, JavaScript, Ruby, PHP, Kotlin and Scala. The selection balances three criteria: (1) popularity based on Github, StackOverFlow, RedMonk and TIOBE rankings, (2) stable infrastructure support through package managers like Conda for reproducible execution, and (3) paradigmatic diversity across compilation strategies, type systems, and memory management models. For detailed programming language rankings across multiple sources, as well as the runtime characteristics information, please see Appendix E.

## 4 EXPERIMENT SETUP

Here we describe the experimental configuration used to evaluate LLMs on the Multi-LCB benchmark.

**Models** We evaluate a diverse set of 24 publicly available large language models (LLMs) spanning from 7B to 685B parameters and covering both general-purpose and code-specialized domains. The pool includes instruction-tuned and reasoning-augmented variants from the Qwen3, DeepSeek, OlympicCoder, OpenReasoning, and OpenCoder families, among others. Representative examples include `GPT-OSS-120B* (Medium)`, `Qwen3-235B-A22B-Thinking-2507*`, `DeepSeek-R1-0528*` and `OpenReasoning-Nemotron-32B*`. We intentionally selected models to capture a wide variety of training paradigms (pure code pretraining, mixed-domain training, instruction tuning, reasoning-enhanced fine-tuning). Appendix F.1 lists all checkpoints with their estimated training cut-off dates.

**Hardware and Environment.** All experiments were run on a cluster of 16 NVIDIA H100 80 GB GPUs with CUDA 12.3 and Python 3.11 inside Conda environments. Each programming language is executed inside an isolated sandbox container that bundles its corresponding compiler or interpreter (e.g., GCC 13 for C++, Rust 1.79, OpenJDK 21, .NET 8, CPython 3.11, Node.js 20). The sandbox enforces strict resource limits: 6 s wall-time per test case, 4 GB memory, and no external network access. This ensures deterministic, secure, and language-agnostic execution.

**Inference Protocol.** Following the original LiveCodeBench protocol, we adopt a zero-shot prompting strategy. For each problem, we generate a model-specific number of tokens (set according to its configuration) using nucleus sampling with temperature = 0.2 and top-p = 0.95, applying a triple-backtick stop sequence to capture the complete code block. Models are served with vLLM (Kwon et al., 2023) or SGLang (Zheng et al., 2024) for efficient batched decoding.

**Evaluation Metric** We report Pass@1 (%) averaged on 10 runs as the primary metric, which measures the fraction of problems solved correctly by the first generated solution. A solution is marked correct only if it compiles/interprets successfully and passes all hidden official test cases without runtime errors or timeouts.

Table 1: Performance results on Multi-LCB for the tasks from February 2025 till May 2025. Scores represent the **Pass@1 (%)** metric averaged on 10 runs. Higher is better, **bold** is best, *italic* is the second best. Temperature t=0.2 (* - reasoning mode)

| Model | Python | C++ | Java | Go | JS | TS | C# | Rust | Ruby | PHP | Kotlin | Scala | Avg |
|---|---|---|---|---|---|---|---|---|---|---|---|---|---|
| GPT-OSS-120B* (Medium) | 71.1 ±2.1 | 72.3 ±1.9 | 70.4 ±3.0 | **69.9 ±3.0** | **70.5 ±1.9** | **70.3 ±3.8** | 57.3 ±2.7 | **70.5 ±2.5** | **70.2 ±2.0** | *66.1 ±2.8* | **71.0 ±2.5** | 54.1 ±3.0 | **67.8 ±5.9** |
| Qwen3-235B-A22B-Thk-2507* | **74.0 ±3.7** | **75.8 ±2.4** | **73.9 ±2.0** | 56.7 ±2.0 | 67.0 ±3.5 | 62.5 ±2.9 | **66.5 ±2.2** | 47.7 ±2.8 | 49.4 ±3.2 | **69.0 ±3.7** | 67.7 ±3.0 | 57.6 ±3.0 | 64.0 ±9.4 |
| DeepSeek-R1-0528* | 66.3 ±2.0 | 68.0 ±1.6 | 67.8 ±1.8 | 55.0 ±3.0 | 64.6 ±2.8 | 58.9 ±3.5 | 61.6 ±2.8 | *63.1 ±2.3* | 62.4 ±1.5 | 61.6 ±2.2 | 66.0 ±2.8 | **62.3 ±2.2** | 63.1 ±3.8 |
| GPT-OSS-20B* (Medium) | 63.6 ±2.5 | 65.7 ±4.0 | 62.7 ±2.7 | *59.9 ±3.4* | 61.9 ±3.4 | 61.8 ±2.3 | 52.4 ±2.5 | 61.9 ±2.3 | 61.7 ±2.1 | 60.5 ±2.5 | 62.4 ±2.2 | 43.1 ±2.9 | 59.8 ±6.1 |
| Qwen3-30B-A3B-Thk-2507* | 64.0 ±2.6 | 65.7 ±4.0 | 62.4 ±3.2 | 44.1 ±1.9 | 51.9 ±4.3 | 46.5 ±2.3 | 56.5 ±3.8 | 51.7 ±4.0 | 42.1 ±2.6 | 58.8 ±2.9 | 50.6 ±2.7 | 43.6 ±2.8 | 53.2 ±8.3 |
| GPT-OSS-120B* (Low) | 56.0 ±3.1 | 55.4 ±2.8 | 56.8 ±2.0 | 51.8 ±2.2 | 55.9 ±2.9 | 55.6 ±1.9 | 45.7 ±2.3 | 56.0 ±1.7 | 53.0 ±2.8 | 53.4 ±2.3 | 55.8 ±2.8 | 42.2 ±4.2 | 53.1 ±4.6 |
| Qwen3-235B-A22B* | 58.9 ±2.8 | 58.3 ±2.7 | 55.0 ±4.2 | 48.7 ±3.5 | 50.0 ±2.8 | 48.8 ±3.1 | 51.0 ±4.0 | 40.7 ±3.7 | 46.6 ±2.6 | 48.4 ±3.8 | 47.5 ±3.9 | 33.6 ±3.4 | 48.9 ±7.0 |
| Qwen3-32B* | 57.6 ±4.0 | 55.3 ±3.4 | 56.0 ±4.5 | 42.1 ±2.6 | 49.6 ±2.6 | 49.3 ±3.8 | 49.1 ±4.1 | 40.1 ±4.1 | 52.1 ±2.7 | 50.0 ±3.1 | 46.4 ±2.7 | 35.6 ±3.4 | 48.6 ±6.7 |
| Qwen3-30B-A3B* | 55.0 ±3.6 | 51.5 ±3.2 | 50.6 ±2.6 | 36.9 ±1.8 | 49.9 ±4.0 | 48.2 ±4.9 | 43.9 ±2.8 | 38.4 ±2.9 | 46.8 ±3.1 | 48.0 ±3.0 | 44.3 ±3.7 | 32.2 ±2.7 | 45.5 ±6.7 |
| GPT-OSS-20B* (Low) | 46.2 ±3.0 | 47.9 ±2.4 | 46.3 ±1.8 | 42.6 ±1.4 | 45.1 ±2.0 | 42.7 ±1.9 | 41.2 ±2.1 | 42.0 ±1.7 | 44.7 ±1.6 | 45.8 ±1.6 | 46.3 ±2.3 | 29.2 ±2.7 | 43.3 ±4.9 |
| Qwen3-14B* | 53.5 ±5.3 | 47.2 ±4.1 | 47.2 ±2.8 | 32.4 ±3.9 | 45.0 ±3.0 | 46.0 ±5.2 | 43.3 ±2.8 | 31.5 ±2.7 | 45.3 ±5.0 | 45.5 ±2.9 | 39.2 ±3.1 | 32.4 ±3.0 | 42.4 ±7.0 |
| Qwen3-235B-A22B-Instr-2507 | 43.8 ±2.8 | 42.7 ±2.4 | 45.5 ±2.4 | 35.0 ±1.4 | 26.4 ±1.3 | 19.5 ±2.7 | 44.1 ±1.8 | 39.5 ±1.0 | 41.5 ±1.4 | 42.4 ±2.2 | 41.1 ±1.20 | 28.1 ±1.9 | 37.5 ±8.4 |
| Qwen3-8B* | 46.3 ±5.9 | 39.7 ±5.0 | 36.7 ±5.5 | 25.8 ±4.4 | 36.5 ±4.9 | 38.8 ±4.8 | 36.3 ±4.3 | 20.5 ±4.1 | 39.5 ±5.8 | 36.0 ±2.2 | 24.0 ±3.33 | 27.0 ±2.7 | 33.9 ±7.7 |
| Qwen3-Coder-30B-A3B-Instr | 36.6 ±2.5 | 31.1 ±2.9 | 35.3 ±2.8 | 25.8 ±2.2 | 28.4 ±1.5 | 28.0 ±1.4 | 34.7 ±2.3 | 34.3 ±2.3 | 34.7 ±2.2 | 31.8 ±1.3 | 35.7 ±2.3 | 20.2 ±1.8 | 31.4 ±4.9 |
| Qwen3-30B-A3B-Instr-2507 | 38.9 ±2.5 | 35.6 ±2.2 | 37.2 ±2.0 | 22.4 ±1.9 | 20.8 ±1.8 | 18.2 ±1.7 | 36.5 ±1.1 | 32.1 ±2.7 | 34.7 ±2.2 | 34.8 ±1.9 | 35.9 ±1.9 | 25.7 ±1.6 | 31.1 ±7.2 |
| Qwen2.5-Coder-32B-Instr | 27.5 ±0.8 | 26.9 ±0.7 | 30.5 ±0.9 | 23.9 ±0.7 | 6.3 ±1.2 | 28.8 ±0.6 | 28.5 ±1.3 | 24.7 ±0.6 | 24.6 ±0.8 | 27.3 ±0.6 | 26.6 ±0.8 | 24.5 ±0.6 | 25.0 ±6.2 |
| Seed-Coder-8B-Instr | 22.1 ±0.8 | 23.4 ±0.7 | 26.0 ±1.5 | 22.1 ±1.5 | 23.3 ±2.3 | 23.1 ±1.6 | 27.0 ±0.8 | 21.8 ±1.4 | 21.6 ±0.9 | 20.4 ±1.3 | 23.4 ±1.4 | 21.8 ±1.0 | 23.0 ±1.9 |
| OpenRsn-Nmt-32B* | 64.4 ±3.6 | 44.2 ±5.2 | 40.8 ±3.0 | 11.5 ±4.2 | 10.8 ±6.9 | 10.5 ±5.3 | 29.9 ±3.8 | 2.8 ±1.5 | 18.3 ±3.4 | 15.8 ±3.2 | 17.3 ±3.5 | 6.0 ±1.3 | 22.7 ±18.5 |
| DeepSeek-R1-Distill-Qwen-32B* | 39.4 ±7.3 | 22.2 ±4.6 | 33.2 ±6.5 | 11.9 ±2.8 | 16.2 ±3.9 | 11.6 ±3.4 | 29.3 ±4.3 | 20.2 ±4.6 | 40.1 ±5.9 | 12.9 ±2.6 | 20.5 ±4.7 | 7.0 ±1.4 | 22.0 ±11.2 |
| Devstral-Small-2505* | 23.2 ±1.0 | 22.6 ±0.9 | 22.8 ±0.7 | 16.1 ±2.2 | 22.7 ±1.0 | 24.7 ±1.4 | 24.1 ±1.4 | 19.9 ±1.2 | 19.9 ±2.0 | 20.8 ±1.3 | 21.2 ±1.0 | 17.2 ±1.4 | 21.3 ±2.7 |
| Qwen2.5-Coder-14B-Instr | 22.0 ±0.6 | 21.3 ±0.3 | 23.9 ±0.8 | 19.2 ±0.6 | 22.6 ±0.8 | 17.5 ±0.8 | 23.3 ±0.5 | 16.7 ±0.6 | 22.7 ±0.8 | 22.7 ±0.6 | 18.1 ±0.4 | 20.4 ±0.6 | 20.9 ±2.4 |
| OpenCodeRsn-Nmt-1.1-32B* | 56.0 ±12.4 | 37.3 ±8.0 | 33.1 ±4.2 | 9.9 ±2.6 | 8.2 ±3.7 | 4.9 ±2.0 | 25.5 ±3.4 | 1.1 ±0.6 | 23.4 ±3.1 | 19.3 ±4.3 | 12.3 ±3.2 | 7.0 ±2.2 | 19.8 ±16.1 |
| DeepSeek-R1-Distill-Qwen-14B* | 41.8 ±5.5 | 16.3 ±1.8 | 24.9 ±2.7 | 10.8 ±2.1 | 10.2 ±3.4 | 11.5 ±3.0 | 29.2 ±4.0 | 3.7 ±1.4 | 34.5 ±4.2 | 3.8 ±1.8 | 11.5 ±3.6 | 3.3 ±1.3 | 16.8 ±12.8 |
| Deepseek-Coder-33B-Instr | 17.2 ±0.7 | 16.2 ±0.5 | 18.5 ±0.8 | 12.4 ±0.5 | 8.5 ±1.7 | 7.4 ±2.3 | 17.1 ±0.7 | 2.6 ±0.6 | 15.2 ±0.7 | 16.5 ±0.7 | 16.0 ±0.5 | 12.2 ±1.0 | 13.3 ±4.9 |

## 5 EXPERIMENTS AND RESULTS

We evaluate a suite of frontier large language models on Multi-LCB, spanning 12 programming languages and reporting Pass@1 averaged on 10 runs as the primary metric (Table 1). This section presents a detailed analysis of model performance on latest Dataset v6 (Feb 2025 – May 2025) (Section 5.1), compares findings with single-language LiveCodeBench (LCB) results (Section 5.2), and investigates contamination signals (Section 5.3). For additional performance results at various sampling temperatures, Pass@5 and Pass@10 metrics and other dataset releases (July 2024-May 2025 and the full 1,055-task benchmark) see appendix F.

### 5.1 EXPERIMENTS RESULTS ON MULTI-LCB

We study performance variations in models released more recently. Particularly, we evaluate 24 recent large language models on Multi-LCB, restricting tasks to those released after 2025-02-01 to ensure live, post-cutoff evaluation and minimize any risk of training-data leakage. Model approximate cutoff dates are listed in Appendix F.1 Table 4.

Table 1 summarizes Pass@1 averaged on 10 runs with temperature $t = 0.2$ performance across twelve programming languages on Dataset v6 (Feb 2025 – May 2025), while Figure 2 highlights the results for the 10 best-performing models.

Our results reveal substantial and practically meaningful performance gaps across languages. For example, GPT-OSS-120B* (Medium) outperforms Qwen3-235B-A22B-Thk-2507* on Go, Javascript, Typescript, Rust, Ruby and Kotlin, and DeepSeek-R1-0528* outperforms Qwen3- 235B-A22B-Thk-2507* on Rust, Ruby and Scala, despite Qwen3-235B-A22B-Thk-2507* being consistently stronger on Python. *This is precisely why strong Python ability is not always a reliable proxy for true cross-lingual code generation competence* and evaluation must consider performance in the target languages rather than relying on Python alone.

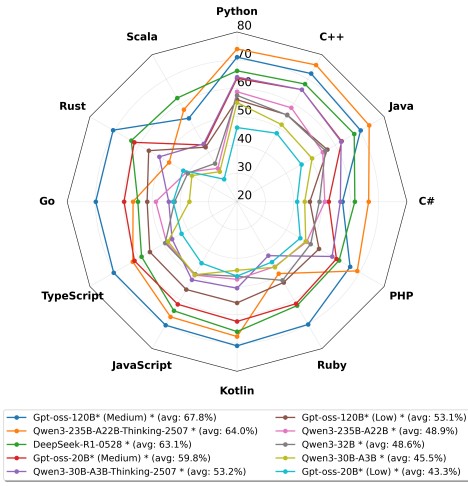

Figure 2: Top-10 models by Pass@1

Figure 3 plots per-model Pass@1 averaged on 10 runs with sampling temperature $t = 0.2$ scores on Python against the cross programming language average on Dataset v6 (Feb 2025 – May 2025).

Almost every point lies above the $x = y$ diagonal, demonstrating a consistent bias toward Python. Models without explicit multi programming languages training, such as `OpenRsn-Nmt-32B*` and `OpenCodeRsn-Nmt-1.1-32B*`, show the starkest gap, exceeding 60% on Python while remaining below 30% across other languages.

Even the largest reasoning-augmented models, including `Qwen3-235B-Thk` and `DeepSeek-R1`, retain a measurable positive bias toward Python, though the disparity is less pronounced.

These results confirm that strong Python ability is not necessarily a reliable proxy for true cross-lingual code generation competence.

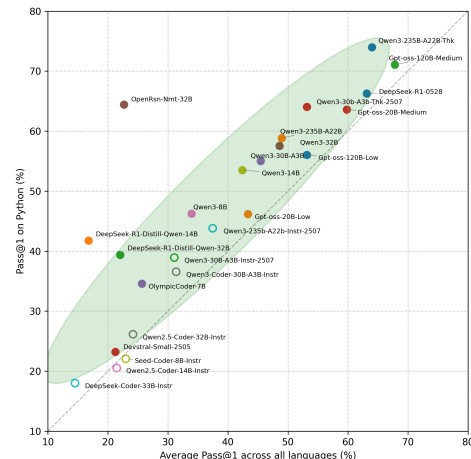

Figure 3: Scatter of Python vs. Average Pass@1

The most strongest models, `GPT-OSS-120B* (Medium)`, `Qwen3-235B-A22B-Thk-2507*` and `DeepSeek-R1-0528*` establish a strong yet far-from-saturated frontier, while the next tier of high-performing models, such as `Qwen3-30B-A3B-Thk-2507*`, illustrates that only a handful of reasoning-augmented variants can exceed the 50% mark. Most of the evaluated models remain below 40%, *underscoring the benchmark's challenge of achieving robust multi programming language code generation correctness.*

Figure 4 plots Pass@1 distribution across 12 languages on with sampling temperature t = 0.2 on Dataset v6 (Feb 2025 – May 2025). Boxes show the interquartile range with the horizontal line marking the median and the red diamond indicating the mean. This reveals a clear difficulty gradient. Python achieves the highest **mean** Pass@1 of 0.482, with Java and C++ close behind at about 0.44. C#, Ruby, PHP, Go, Rust, Kotlin and JavaScript/TypeScript form a middle tier with means near 0.33-0.39, while Scala consistently trail at means below 0.29. These gaps persist across the top-performing models, reflecting structural challenges such as compilation complexity, ownership semantics, and smaller ecosystem resources.

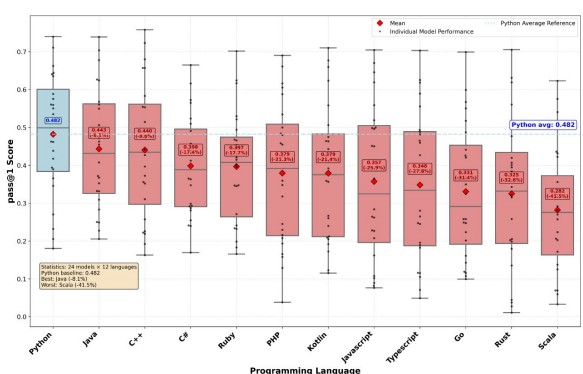

Figure 4: Pass@1 distribution across 12 languages

We observe that Python consistently outperforms other languages on Multi-LCB. This suggests that current LLMs are substantially more trained on Python code, especially for reasoning-mode training, and that cross-language knowledge transfer remains only partial. We suppose that model performance could be improved by increasing training coverage of non-Python programming languages.

## 5.2 COMPARISON WITH LIVECODEBENCH

To verify that our multilingual extensions preserve the fidelity of the original LiveCodeBench (LCB), we compare Pass@1 scores on the Python subset of Multi-LCB against the official results reported

for LCB versions v4-v6. Table 2 reports original leaderboard results (ORIG) and our reproduced scores (OUR), with $\Delta$ representing the absolute difference.

Table 2: Comparison of reasoning/code models on Python across benchmark versions (v4–v6). Original leaderboard values (ORIG, %) are contrasted with our reproduced scores (OUR, %). Difference is computed as $\Delta = \text{OUR} - \text{ORIG}$.

| Model | Benchmark version (range) | ORIG (%) | OUR (%) | $\Delta$ (%) | Source |
|---|---|---|---|---|---|
| Qwen3-235B-A22B-Thinking-2507 | v6 [2502–2505] | 74.1 | 74.0 | -0.1 | Hugging Face |
| DeepSeek R1 0528 | v6 [2502–2505] | 68.7 | 66.3 | -2.4 | LCB leaderboard |
| Qwen3-30B-A3B-Thinking-2507 | v6 [2502–2505] | 66.0 | 64.0 | -2.0 | Hugging Face |
| OpenReasoning-Nemotron-32B | v6 [2502–2505] | 65.6 | 64.4 | -1.2 | LCB leaderboard |
| OpenCodeReasoning-Nemotron-1.1-32B | v6 [2502–2505] | 61.4 | 56.0 | -5.4 | LCB leaderboard |
| Qwen3-30B-A3B* | v6 [2502–2505] | 57.4 | 55.0 | -2.4 | Hugging Face |
| Qwen3-235B-A22B | v6 [2502–2505] | 55.7 | 58.9 | 3.2 | LCB leaderboard |
| Qwen3-235B-A22B-Instruct-2507 | v6 [2502–2505] | 51.8 | 43.8 | -8.0 | Hugging Face |
| Qwen3-32B* | v5 [2410–2502] | 65.7 | 64.3 | -1.4 | Qwen3 Tech report |
| Qwen3-14B* | v5 [2410–2502] | 63.5 | 56.7 | -6.8 | Qwen3 Tech report |
| Qwen3-30B-A3B* | v5 [2410–2502] | 62.6 | 61.0 | -1.6 | Qwen3 Tech report |
| Qwen3-8B* | v5 [2410–2502] | 57.5 | 49.1 | -8.6 | Qwen3 Tech report |
| Seed-Coder-8B-Instruct | v5 [2410–2502] | 24.7 | 19.8 | -4.9 | Hugging Face |
| OpenCodeReasoning-Nemotron-1.1-32B | v4–v5 [2408–2502] | 69.9 | 65.3 | -4.6 | Hugging Face |
| OlympicCoder-32B | v4–v5 [2408–2502] | 54.5 | 52.3 | -2.2 | Hugging Face |
| OlympicCoder-7B | v4–v5 [2408–2502] | 40.7 | 35.6 | -5.1 | Hugging Face |
| Qwen2.5-Coder-32B-Instruct | v4–v5 [2408–2502] | 28.3 | 27.6 | -0.7 | Hugging Face |

Overall, reproduction is strong: differences are typically within a few percentage points, with a mean absolute deviation of only about 3%. For example, `Qwen3-235B-A22B-Thinking-2507` achieves 74.0% Pass@1 in our evaluation versus 74.1% on the original v6 leaderboard ($\Delta = -0.1$), while `DeepSeek-R1-0528` records 66.3% compared to 68.7% ($\Delta = -2.4$). Even for models with larger gaps, such as `Qwen3-235B-A22B-Ins-2507` ($\Delta = -8.0$) or `Qwen3-8B*` ($\Delta = -8.6$), the rank ordering across models remains consistent.

These close alignments confirm that Multi-LCB's multilingual transformations introduce no artificial difficulty for Python tasks. Performance differences instead reflect natural leaderboard variance and underscore that the multilingual benchmark faithfully reproduces the single-language LCB setting, ensuring that any additional challenges arise from genuine cross-language generalization rather than implementation artifacts.

## 5.3 CONTAMINATION ON MULTI-LCB

A core design goal of Multi-LCB is contamination-aware evaluation via release-date filtering. Nevertheless, *time-wise* analysis reveals clear evidence of residual contamination on older (pre-cutoff) problems. Figure 5 shows monthly Pass@1 trends for the top-10 models averaged across all programming languages : scores are systematically higher on earlier months and exhibit step-like drops when the evaluation window crosses model cutoffs, followed by sustained lower performance on post–cutoff problems. Our main comparisons in Section 5 restrict evaluation to tasks released on or after 2025-02-01, ensuring *live, post-cutoff* measurement. Under this setting, performance drops to a level that better reflects true generalization, whereas inflated scores on older windows are explained by pretraining exposure rather than genuine zero-contamination generalization.

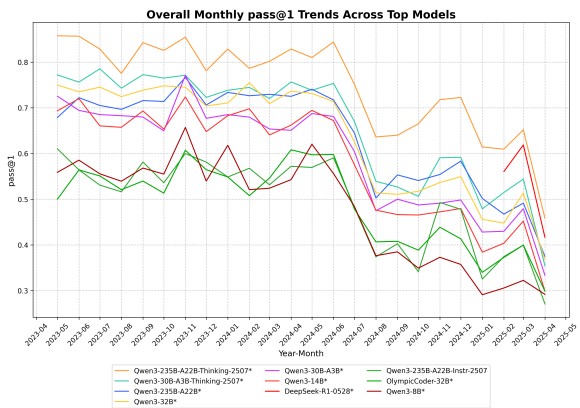

Figure 5: Monthly Pass@1 trends averaged across all programming languages for top-10 models.

## 6 LIMITATIONS AND THREATS TO VALIDITY

**Language Coverage and Selection.** Multi-LCB covers 12 programming languages but does not include some important languages such as Swift, Haskell, R, and others. The language selection is based on popularity rankings in 2025, which may not reflect specialized domains or emerging languages. Additionally, some languages have various dialects and versions that are not accounted for in our evaluation framework.

**Task Complexity and Domain.** While the selected programming languages span different domains (systems programming, web development, data science), the tasks themselves remain rooted in competitive programming. Although algorithmic problem-solving has indirect relevance to industrial coding capabilities, Multi-LCB does not directly assess real-world software engineering scenarios such as API integration, debugging legacy code, or collaborative development workflows.

**Evaluation Protocol Constraints.** The strict STDIN/STDOUT format may introduce performance degradation not only due to algorithmic reasoning limitations but also due to syntax unfamiliarity, difficulty parsing input formats, or failure to follow output specifications. Models may fail tasks due to format compliance issues rather than core problem-solving deficits, potentially confounding our assessment of true multilingual coding competence.

**Model Selection Bias.** Our evaluation focuses exclusively on publicly available models, excluding proprietary systems that may represent the current state-of-the-art. This limitation means our results reflect only a subset of available models and may not accurately represent the real-world leaderboard of multilingual code generation capabilities.

**Construct Validity.** The automatic conversion from functional format to STDIN/STDOUT may alter task complexity differently across programming languages. Some languages may be more naturally suited for certain problem types, potentially creating unequal evaluation conditions that affect cross-language comparisons.

**Internal Validity.** Despite date-based filtering, hidden forms of contamination may persist through similar problem patterns or solution templates present in training data. Additionally, models may exhibit temporal bias based on varying exposure to different programming languages during their training periods.

## 7 FUTURE WORK

Multi-LCB's modular design enables straightforward language expansion. We plan to add Swift, Haskell, R, and Julia by defining their compilation commands and runtime environments. We will evaluate proprietary models (GPT-4, Claude, Gemini) to establish comprehensive multilingual leaderboards reflecting current state-of-the-art performance. The STDIN/STDOUT framework directly supports LCB-Pro (Zheng et al., 2025) and other benchmarks requiring format conversion, enabling broader contamination-aware multilingual evaluation without additional infrastructure changes.

## 8 CONCLUSIONS

We introduced **Multi-LCB**, a contamination aware benchmark for evaluating large language models on multilingual code generation. Multi-LCB provides an extensible framework spanning twelve programming languages and continuously updates with newly released problems. The conversion methodology extends beyond LCB to other Python benchmarks (e. g. LCB Pro (Zheng et al., 2025)), offering a general approach for multilingual code evaluation. By inheriting LiveCodeBench's live evaluation protocol and unified STDIN/STDOUT execution, it enables rigorous, cross programming language assessment and mitigates data contamination that affects static benchmarks. Our experiments expose programming language specific contamination, evidence of Python overfitting, and significant performance gaps across programming languages. We hope Multi-LCB will serve as a durable resource for advancing the evaluation of code-oriented LLMs and guiding future research in multilingual program synthesis.

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

# Appendix

## CONTENTS

## A  LEGAL COMPLIANCE AND LICENSE

The Multi-LCB benchmark contains no personally identifiable information, offensive content, or proprietary code. It is derived entirely from the publicly released LiveCodeBench (LCB) dataset, which itself sources only publicly accessible contest problems, reference solutions, and test cases from **LeetCode**, **AtCoder**, and **Codeforces**. Our redistribution and multi programming language transformation of LCB fall under Fair Use (§107, U.S. Copyright Act): the benchmark is provided solely for non-commercial academic research, reproduces only the material necessary for evaluation, and does not diminish the market value of the original platforms or LCB. Multi-LCB is strictly an evaluation resource, no models are trained on these tasks, and is released under a **CC BY-NC 4.0** license to ensure non-commercial use.

## B  UI OF MULTI-LCB

Figure 6 presents the web interface of **Multi-LCB**, displaying a subset of tasks released between **January 2024** and **December 2024**. A *time-range scroller* at the top allows users to interactively select different time windows to filter tasks and monitor model performance on newly released problems. This interactive design highlights the *live and continuously updated* nature of the benchmark, enabling researchers to track progress as fresh contest tasks are incorporated.

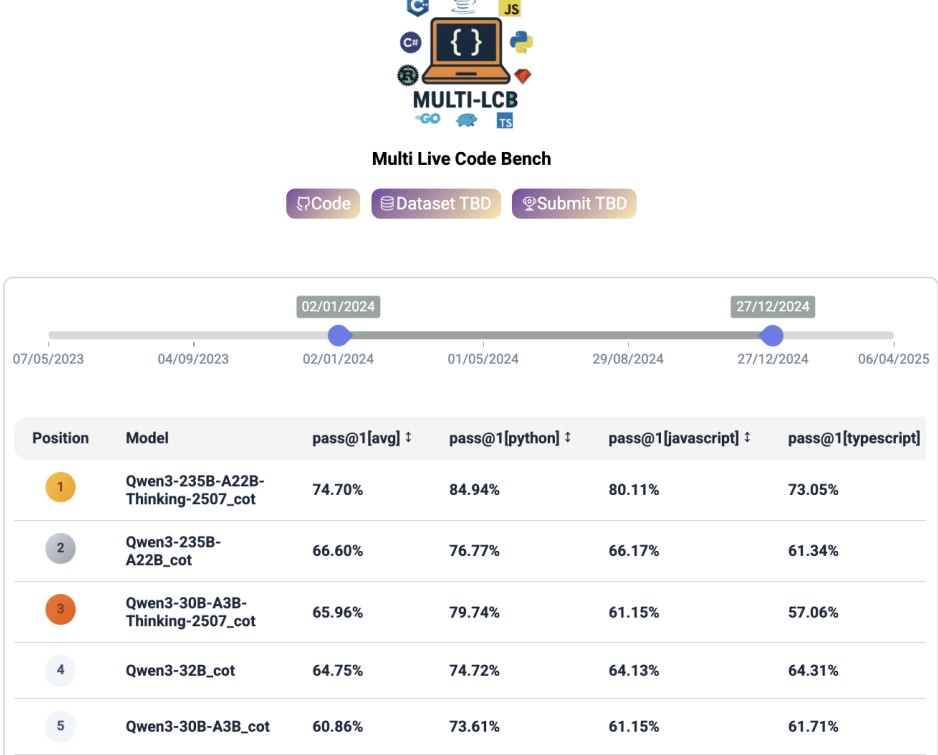

Figure 6: Multi-LCB web interface showing tasks released between January 2024 to December 2024, with an interactive time-range scroller for filtering and visualization.

## C  PROMPT EXAMPLES

This appendix shows example prompts from Multi-LCB. We distinguish the **original problem text** as it appears on the source platform and the **additional instructions** that we add in order to unify everything into the STDIN/STDOUT format. Original parts are placed in blue boxes, while added parts are placed in red boxes.

## C.1 ATCODER/CODEFORCES EXAMPLE (NATIVE STDIN/STDOUT)

**Original**

### Question: Find the number of positive integers not greater than $N$ that have exactly 9 positive divisors.

**Input:** $N$
**Output:** Print the answer.

**Constraints:** $1 \leq N \leq 4 \times 10^{12}$

Sample Input 1:

```
200
```

Sample Output 1:

```
3
```

**Added**

### Format:
Read the inputs from STDIN solve the problem and write the answer to STDOUT (do not directly test on the sample inputs). Enclose your code within delimiters as follows. Ensure that when the python program runs, it reads the inputs, runs the algorithm and writes output to STDOUT.

```
""" python
# YOUR CODE HERE
"""
```

### Answer: (use the provided format with triple quotes)

## C.2 LEETCODE EXAMPLE (ADAPTED INTO STDIN/STDOUT)

**Original**

### Question: You are given an integer array `enemyEnergies` and an integer `currentEnergy`... (original description)

**Example 1:**
**Input:** `enemyEnergies` $= [3, 2, 2]$, `currentEnergy` $= 2$
**Output:** $3$
**Explanation:**
Several operations lead to a maximum of 3 points (see original problem description).

**Example 2:**
**Input:** `enemyEnergies` $= [2]$, `currentEnergy` $= 10$
**Output**: $5$
**Explanation**:
Performing the first operation 5 times on enemy 0 yields the maximum number of points.

**Constraints**:

- $1 \leq$ `enemyEnergies.length` $\leq 10^5$
- $1 \leq$ `enemyEnergies`$[i] \leq 10^9$
- $0 \leq$ `currentEnergy` $\leq 10^9$

---

**Added**

### Format:
Read the inputs from STDIN solve the problem and write the answer to STDOUT (do not directly test on the sample inputs). Enclose your code within delimiters as follows. Ensure that when the python program runs, it reads the inputs, runs the algorithm and writes output to STDOUT.

For 2D arrays, the first line indicates the number of rows, followed by newline-separated rows.

Sample Input 1:

```
3 2 2
2
```

Sample Output 1:

```
3
```

```python
# YOUR CODE HERE
```

### Answer: (use the provided format with triple quotes)

---

For non-Python settings, only the header of the code block is replaced (e.g., `""" cpp`, `""" java`). The rest of the prompt structure remains identical.

## D   TASKS DISTRIBUTION

### D.1   TASK DISTRIBUTION BY DIFFICULTY AND PLATFORM

LiveCodeBench (LCB) continuously aggregates competitive programming problems in Python from three major platforms: **LeetCode**, **AtCoder**, and **Codeforces**. Figure 7 shows the monthly distribution of tasks by difficulty, and Figure 8 presents the monthly distribution by source platform. Together, these figures highlight the steady inflow of new problems and the live, contamination-aware nature of LCB, and, by extension Multi-LCB.

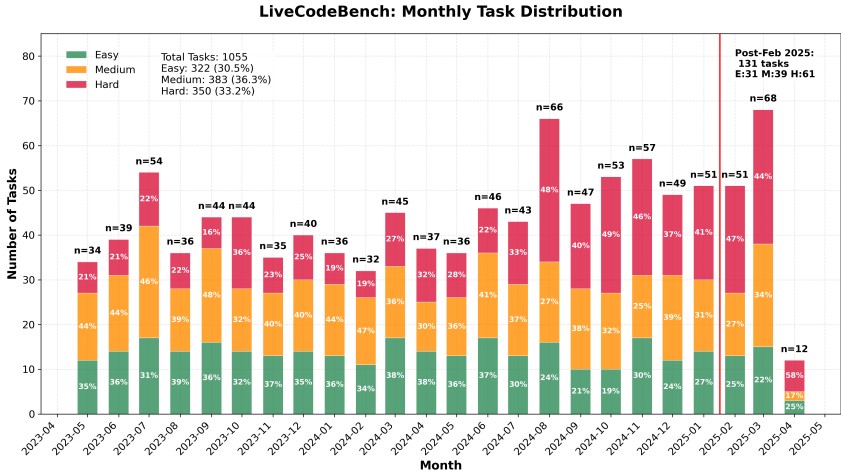

Figure 7: Monthly distribution of Tasks by Difficulty.

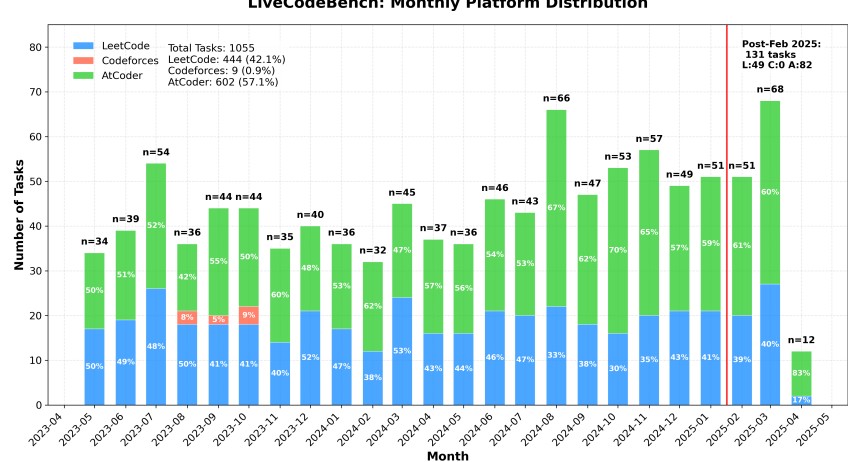

Figure 8: Monthly distribution of LCB tasks by platform.

Each platform hosts frequent contests whose tasks provide a natural language description of a problem, example input/output pairs, and hidden tests, ensuring that solutions must be fully correct to receive credit. Because every contest attracts thousands of participants and receives official editorial review, the problems are inherently vetted for clarity and correctness. Across the full lifetime of the dataset, the platform composition is as follows:

**Codeforces**: Competitive-programming problems known for a wide range of difficulty and algorithmic focus, almost exclusively in `STDIN/STDOUT` format.

**LeetCode**: Interview oriented challenges emphasizing data structures and algorithms, originally in a Functional format.

**AtCoder**: Algorithmically rich competitive programming problems, typically using `STDIN/STDOUT` input/output.

## D.2 Task Distribution by I/O Data Dimensionality (LeetCode Functional Format)

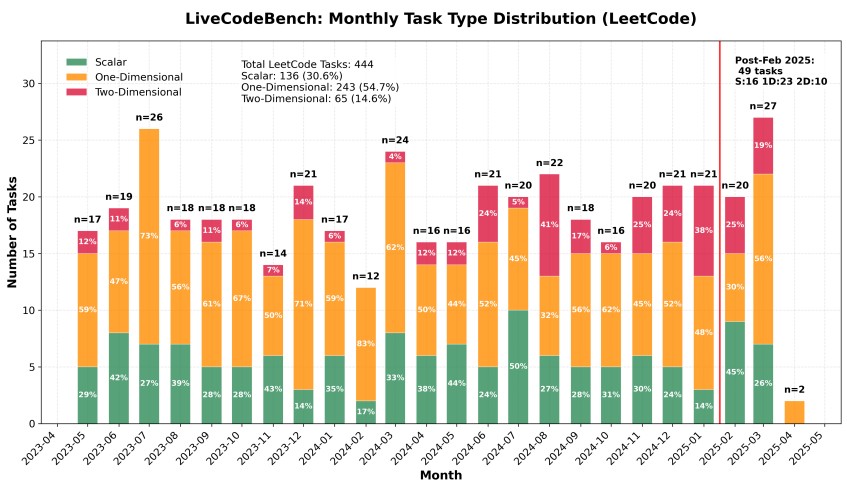

Figure 9: Monthly task distribution by I/O data dimensionality (LeetCode Functional format).

Figure 9 presents the temporal distribution of LeetCode tasks grouped by the I/O data dimensionality of their Functional format. The plot highlights how problems with different input/output structures,

such as scalar values, one-dimensional arrays, and two-dimensional arrays, have entered the benchmark over time, illustrating the variety of functional tasks inherited from LeetCode within the LCB dataset.

# E  PROGRAMMING LANGUAGE RANKINGS AND RUNTIME CHARACTERISTICS

This study evaluates multilingual code generation across major programming languages selected for their 2025 popularity and broad industrial relevance. Programming language rankings across multiple sources presented in Table 3.

Table 3: Programming language rankings across multiple sources (dates in footnotes)

| Language | TIOBE[1] | GitHub[2] | Stack Overflow[3] | RedMonk[4] |
|---|---|---|---|---|
| Python | 1 (26.98%) | 1 | 4 (57%) | 2 |
| C++ | 2 (9.80%) | 5 | 9 (23%) | 7 |
| Java | 4 (8.76%) | 2 | 7 (29%) | 3 |
| C# | 5 (4.87%) | 10 | 8 (27%) | 5 |
| JavaScript | 6 (3.36%) | 4 | 1 (66%) | 1 |
| TypeScript | 37 (0.28%) | 6 | 6 (43%) | 6 |
| Go | 7 (2.04%) | 3 | 13 ($\sim$2%) | 12 |
| Rust | 18 (1.01%) | 13 | 14 ($\sim$2%) | 19 |
| Ruby | 23 (0.76%) | 8 | 18 ($\sim$1.5%) | 9 |
| PHP | 14 (1.28%) | 7 | 12 ($\sim$15%) | 4 |
| Kotlin | 20 (0.90%) | 15 | 15 ($\sim$3%) | 14 |
| Scala | 34 (0.41%) | 14 | 29 ($\sim$1%) | 14 |

These languages span a wide range of paradigms and runtime characteristics, capturing the diversity of real-world software development:

- **Compilation model:**
  - Compiled/JIT — C++, Rust, Go, Java, C#, Scala, Kotlin
  - Interpreted — Python, Ruby, PHP
  - Transpiled — TypeScript $\rightarrow$ JavaScript
- **Type system:**
  - Static — C++, Rust, Go, Java, C#, Scala, Kotlin, TypeScript
  - Dynamic — Python, JavaScript, Ruby, PHP
- **Memory management:**
  - RAII/manual — C++
  - Ownership/borrowing — Rust
  - Garbage collection — Java, C#, Go, Scala, Kotlin, PHP, Ruby, JavaScript/TypeScript
- **Runtime platforms:**
  - Native — C++, Rust, Go
  - JVM — Java, Scala, Kotlin
  - .NET CLR — C#
  - Interpreters/VMs — Python, Ruby, PHP
  - JavaScript engines — JavaScript, TypeScript
- **Domain ecosystems:**
  - Systems/performance — C++, Rust, Go
  - Enterprise/JVM and .NET — Java, C#, Scala, Kotlin
  - Web/backend and scripting — JavaScript, TypeScript, PHP, Ruby
  - Data/AI glue — Python

# F EXPERIMENTS

## F.1 MODELS OVERVIEW

We provide details for all models included in our study in Table 4.

Table 4: Overview of Large Language Models (* denotes reasoning mode)

| Model | Short Name | Approximate Cutoff Date | Link |
|---|---|---|---|
| openai/gpt-oss-120b | Gpt-oss-120B* (Medium/Low) | 08/05/2025 | gpt-oss |
| openai/gpt-oss-20b | Gpt-oss-20B* (Medium/Low) | 08/05/2025 | gpt-oss |
| Qwen/Qwen3-235B-A22B-Thinking-2507 | Qwen3-235B-A22B-Thk* | 10/31/2024 | qwen |
| deepseek-ai/DeepSeek-R1-0528 | DeepSeek-R1-0528* | 11/29/2024 | deepseek-ai |
| Qwen/Qwen3-30B-A3B-Instruct-2507 | Qwen3-30b-A3b-Thk-2507* | 10/31/2024 | qwen |
| Qwen/Qwen3-32B | Qwen3-32B* | 10/31/2024 | qwen |
| Qwen/Qwen3-235B-A22B | Qwen3-235B-A22B* | 10/31/2024 | qwen |
| Qwen/Qwen3-30B-A3B | Qwen3-30B-A3B* | 10/31/2024 | qwen |
| Qwen/Qwen3-14B | Qwen3-14B* | 10/31/2024 | qwen |
| open-r1/OlympicCoder-32B | OlympicCoder-32B* | - | open-r1 |
| Qwen/Qwen3-235B-A22B-Instruct-2507 | Qwen3-235b-A22b-Instr-2507 | 10/31/2024 | qwen |
| Qwen/Qwen3-8B | Qwen3-8B* | 10/31/2024 | qwen |
| Qwen/Qwen3-Coder-30B-A3B-Instruct | Qwen3-Coder-30B-A3B-Instr | - | qwen |
| Qwen/Qwen3-30B-A3B-Instruct-2507 | Qwen3-30B-A3B-Instr-2507 | 10/31/2024 | qwen |
| open-r1/OlympicCoder-7B | OlympicCoder-7B* | - | open-r1 |
| Qwen/Qwen2.5-Coder-32B-Instruct | Qwen2.5-Coder-32B-Instr | 03/23/2024 | qwen |
| nvidia/OpenCodeReasoning-Nemotron-1.1-32B | OpenRsn-Nmt-32B* | - | nvidia |
| ByteDance-Seed/Seed-Coder-8B-Instruct | Seed-Coder-8B-Instr | - | bytedance-seed |
| Qwen/Qwen2.5-Coder-14B-Instruct | Qwen2.5-Coder-14B-Instr | 03/23/2024 | qwen |
| mistralai/Devstral-Small-2505 | Devstral-Small-2505 | 11/22/2024 | mistralai |
| nvidia/OpenReasoning-Nemotron-32B | OpenRsn-Nmt-32B | - | nvidia |
| deepseek-ai/deepseek-coder-33b-instruct | DeepSeek-Coder-33B-Instr | 08/30/2023 | deepseek-ai |

## F.2 PERFORMANCE ON THE MULTI-LCB (FEB-MAY 2025 SUBSET) ACROSS SAMPLING TEMPERATURES

### F.2.1 PASS@1 AVERAGED OVER 10 RUNS PERFORMANCE AT VARIOUS SAMPLING TEMPERATURES

Table 5 report Pass@1 scores averaged over 10 runs at sampling temperature $t = 0.6$. Each score indicates the percentage of problems solved correctly on the first attempt, with higher values reflecting better performance.

Table 5: Performance results at temperature $t = 0.6$ Scores represent the **Pass@1 (%)** metric averaged on 10 runs. Higher is better, **bold** is best, *italic* is the second best. (* - reasoning mode)

| Model | Python | C++ | Java | Go | JS | TS | C# | Rust | Ruby | PHP | Kotlin | Scala | Avg |
|---|---|---|---|---|---|---|---|---|---|---|---|---|---|
| Gpt-oss-120B* (Medium) | 69.9 ±1.8 | 72.6 ±2.1 | 70.0 ±1.9 | **69.9** ±2.4 | 70.5 ±2.7 | **71.9** ±2.0 | 59.8 ±2.9 | **70.1** ±2.4 | **69.6** ±3.1 | 67.3 ±1.8 | **70.2** ±2.8 | 54.1 ±3.6 | **68.0** ±5.5 |
| Qwen3-235B-A22B-Thk-2507* | **74.0** ±2.5 | **75.3** ±2.6 | **74.8** ±2.9 | 57.7 ±2.4 | 68.6 ±2.5 | 63.4 ±2.1 | **65.8** ±2.6 | 51.5 ±3.2 | 48.9 ±2.4 | **67.5** ±2.5 | 68.6 ±2.8 | 59.2 ±2.8 | 63.7 ±8.6 |
| DeepSeek-R1-0528* | 66.6 ±2.6 | 68.4 ±2.8 | 67.2 ±2.3 | 54.1 ±2.4 | 64.9 ±2.8 | 58.6 ±4.6 | 62.1 ±2.2 | 62.5 ±3.8 | 62.4 ±2.0 | 61.3 ±1.9 | 66.4 ±3.0 | **61.2** ±3.5 | 63.0 ±4.1 |
| Gpt-oss-20B* (Medium) | 62.3 ±2.7 | 65.5 ±3.3 | 62.5 ±2.0 | 59.2 ±0.4 | 63.6 ±2.9 | 63.9 ±2.0 | 50.2 ±2.3 | 61.5 ±2.1 | 61.4 ±2.8 | 60.6 ±3.1 | 62.1 ±2.6 | 42.4 ±3.3 | 59.6 ±6.6 |
| Qwen3-30B-A3B-Thk-2507* | 65.2 ±3.0 | 66.0 ±3.6 | 63.9 ±2.9 | 44.5 ±2.0 | 53.4 ±1.9 | 50.6 ±3.0 | 56.2 ±3.0 | 51.2 ±3.2 | 43.1 ±3.5 | 57.0 ±3.6 | 52.2 ±3.3 | 40.5 ±2.4 | 53.6 ±8.5 |
| Gpt-oss-120B* (Low) | 57.6 ±2.5 | 56.6 ±2.4 | 57.2 ±3.0 | 53.6 ±2.6 | 54.8 ±2.1 | 54.6 ±2.3 | 46.4 ±1.4 | 55.8 ±2.3 | 53.8 ±3.2 | 53.4 ±2.5 | 54.9 ±2.6 | 40.8 ±3.2 | 53.3 ±4.9 |
| Qwen3-235B-A22B* | 58.2 ±1.7 | 58.6 ±3.1 | 56.1 ±2.6 | 48.6 ±3.6 | 49.9 ±2.9 | 46.6 ±2.5 | 51.5 ±2.8 | 43.2 ±2.9 | 48.4 ±1.8 | 48.9 ±3.0 | 47.7 ±4.0 | 34.4 ±3.5 | 49.3 ±6.7 |
| Qwen3-32B* | 58.6 ±2.7 | 56.2 ±2.5 | 54.2 ±2.7 | 42.5 ±4.0 | 50.5 ±3.5 | 50.8 ±3.2 | 51.1 ±2.4 | 39.1 ±2.1 | 52.0 ±2.3 | 51.9 ±1.8 | 45.2 ±2.6 | 38.1 ±2.1 | 49.2 ±6.5 |
| Qwen3-30B-A3B* | 55.3 ±3.2 | 53.5 ±3.4 | 51.0 ±3.4 | 37.1 ±3.1 | 50.1 ±1.8 | 49.9 ±2.9 | 42.9 ±2.8 | 38.4 ±3.0 | 47.3 ±2.6 | 48.2 ±2.0 | 43.9 ±3.1 | 33.7 ±1.7 | 46.0 ±6.8 |
| Qwen3-14B* | 55.9 ±3.5 | 49.9 ±2.2 | 50.6 ±3.2 | 34.6 ±1.8 | 48.1 ±3.1 | 47.7 ±3.8 | 44.8 ±1.5 | 32.0 ±3.3 | 46.2 ±2.9 | 44.5 ±1.3 | 39.5 ±5.3 | 31.2 ±2.7 | 43.7 ±7.8 |
| Gpt-oss-20B* (Low) | 45.7 ±2.0 | 47.5 ±2.1 | 45.6 ±2.4 | 41.8 ±1.9 | 43.5 ±1.8 | 44.1 ±2.4 | 39.9 ±2.9 | 43.1 ±2.8 | 44.0 ±3.0 | 44.7 ±2.7 | 45.0 ±1.6 | 32.5 ±4.1 | 43.1 ±3.9 |
| Qwen3-235B-A22B-Instr-2507 | 44.9 ±2.6 | 42.8 ±2.7 | 45.5 ±2.9 | 36.1 ±2.2 | 27.9 ±3.0 | 21.8 ±1.8 | 43.7 ±2.5 | 40.2 ±2.3 | 41.5 ±2.9 | 42.6 ±2.5 | 41.5 ±1.5 | 28.8 ±2.2 | 38.1 ±7.8 |
| Qwen3-8B* | 50.5 ±2.5 | 43.7 ±2.8 | 42.5 ±1.8 | 29.2 ±2.7 | 41.3 ±2.5 | 41.8 ±2.6 | 39.8 ±1.7 | 22.4 ±2.0 | 42.5 ±3.2 | 38.7 ±2.6 | 25.4 ±2.6 | 29.5 ±2.5 | 37.3 ±8.5 |
| Qwen3-30B-A3B-Instr-2507 | 41.3 ±2.5 | 36.7 ±1.5 | 37.1 ±2.5 | 23.4 ±3.6 | 21.4 ±1.3 | 20.2 ±3.5 | 35.3 ±2.5 | 31.8 ±2.5 | 33.7 ±1.4 | 35.3 ±1.4 | 36.1 ±1.9 | 26.4 ±2.1 | 31.6 ±6.9 |
| Qwen3-Coder-30B-A3B-Instr | 36.0 ±2.0 | 33.1 ±1.8 | 35.5 ±3.1 | 25.2 ±1.8 | 28.6 ±1.5 | 26.3 ±2.1 | 34.5 ±2.2 | 33.7 ±2.2 | 33.8 ±1.7 | 31.5 ±1.2 | 35.0 ±2.1 | 20.9 ±1.7 | 31.2 ±4.8 |
| DeepSeek-R1-Distill-Qwen-32B* | 45.9 ±2.8 | 25.8 ±1.6 | 38.8 ±2.9 | 12.9 ±3.3 | 20.4 ±3.4 | 15.9 ±1.9 | 34.2 ±3.2 | 21.2 ±2.2 | 43.0 ±2.3 | 13.8 ±1.5 | 21.1 ±2.4 | 8.7 ±1.3 | 25.1 ±12.5 |
| Qwen2.5-Coder-32B-Instr | 27.4 ±2.6 | 25.3 ±2.4 | 27.6 ±2.7 | 25.0 ±2.1 | 8.1 ±1.9 | 25.0 ±1.3 | 28.6 ±2.1 | 24.2 ±2.3 | 23.5 ±2.1 | 25.3 ±2.5 | 26.6 ±2.1 | 23.5 ±1.8 | 24.2 ±5.3 |
| OpenRsn-Nmt-32B* | 66.0 ±2.9 | 44.8 ±4.7 | 41.3 ±1.7 | 10.8 ±3.8 | 12.2 ±7.2 | 11.0 ±6.6 | 31.5 ±2.0 | 3.1 ±2.6 | 17.6 ±4.7 | 13.9 ±2.9 | 17.2 ±2.9 | 7.4 ±1.9 | 23.1 ±18.9 |
| Seed-Coder-8B-Instr | 22.6 ±1.2 | 22.9 ±1.7 | 24.4 ±1.7 | 19.2 ±1.9 | 23.4 ±1.6 | 22.8 ±1.8 | 22.8 ±1.6 | 21.8 ±1.3 | 21.5 ±2.6 | 19.9 ±2.0 | 23.5 ±1.9 | 21.5 ±0.7 | 22.2 ±1.5 |
| Qwen2.5-Coder-14B-Instr | 22.3 ±1.8 | 21.8 ±1.7 | 24.6 ±1.4 | 18.2 ±1.6 | 22.5 ±2.5 | 19.8 ±1.9 | 23.7 ±2.3 | 17.6 ±1.3 | 22.4 ±1.7 | 21.0 ±2.7 | 22.3 ±0.7 | 19.8 ±2.3 | 21.3 ±2.1 |
| OpenCodeRsn-Nmt-1.1-32B* | 62.8 ±3.6 | 41.2 ±2.6 | 31.8 ±4.8 | 8.6 ±2.1 | 9.9 ±5.6 | 6.8 ±3.5 | 25.0 ±2.9 | 1.2 ±0.8 | 25.7 ±3.4 | 21.1 ±3.3 | 13.6 ±2.4 | 7.1 ±3.2 | 21.2 ±17.6 |
| Devstral-Small-2505* | 22.1 ±1.6 | 22.0 ±1.1 | 22.9 ±0.9 | 16.8 ±2.4 | 22.3 ±1.3 | 24.1 ±1.5 | 22.3 ±1.6 | 19.9 ±1.2 | 20.2 ±2.5 | 21.1 ±2.1 | 21.6 ±1.8 | 16.3 ±1.3 | 20.9 ±2.4 |
| DeepSeek-R1-Distill-Qwen-14B* | 45.7 ±3.2 | 18.5 ±2.3 | 24.6 ±4.1 | 8.3 ±1.3 | 9.8 ±1.7 | 10.5 ±2.3 | 30.5 ±3.2 | 3.7 ±1.5 | 36.6 ±3.5 | 4.0 ±1.9 | 13.6 ±3.4 | 3.5 ±1.0 | 17.4 ±14.0 |
| DeepSeek-Coder-33B-Instr | 18.6 ±1.5 | 18.2 ±0.9 | 20.8 ±1.3 | 14.0 ±1.4 | 10.8 ±3.6 | 9.0 ±3.8 | 18.6 ±1.8 | 2.9 ±1.5 | 17.0 ±2.0 | 16.2 ±1.5 | 17.0 ±1.7 | 12.1 ±2.5 | 14.6 ±5.1 |

Table 6 reports Pass@1 scores averaged over 10 runs at sampling temperature $t = 1.0$. Each score indicates the percentage of problems solved correctly on the first attempt, with higher values reflecting better performance.

Table 6: Performance results at temperature $t = 1.0$. Scores represent the **Pass@1 (%)** metric averaged on 10 runs. Higher is better, **bold** is best, *italic* is the second best. (* - reasoning mode)

| Model | Python | C++ | Java | Go | JS | TS | C# | Rust | Ruby | PHP | Kotlin | Scala | Avg |
|---|---|---|---|---|---|---|---|---|---|---|---|---|---|
| Gpt-oss-120B* (Medium) | 69.1±2.0 | 72.3±1.9 | 70.0±3.5 | 67.9±2.8 | 71.8±3.1 | 70.5±2.1 | 58.8±4.8 | 70.5±3.1 | 69.9±1.9 | 68.1±2.2 | 71.2±1.8 | 51.1±3.4 | 67.6±6.3 |
| Qwen3-235B-A22B-Thk-2507* | 73.7±2.6 | 75.0±2.7 | 73.7±3.2 | 57.0±3.2 | 69.0±3.5 | 63.5±2.2 | 67.4±2.5 | 54.2±3.5 | 49.2±2.4 | 68.7±1.8 | 68.6±3.4 | 58.5±2.8 | 64.9±8.4 |
| Gpt-oss-20B* (Medium) | 64.2±2.9 | 65.0±2.1 | 63.1±3.2 | 58.9±2.5 | 62.1±3.1 | 61.5±2.5 | 52.6±2.8 | 59.8±2.1 | 61.7±3.0 | 61.5±3.3 | 62.9±3.4 | 40.4±2.3 | 59.5±6.8 |
| DeepSeek-R1-0528* | 59.8±2.7 | 62.3±2.4 | 62.4±1.1 | 50.5±4.6 | 59.9±1.7 | 55.3±2.7 | 58.4±2.5 | 57.4±2.3 | 57.3±2.6 | 57.4±1.4 | 57.7±2.2 | 57.4±2.8 | 58.0±3.2 |
| Gpt-oss-120B* (Low) | 58.2±3.2 | 56.6±3.0 | 55.3±3.3 | 53.0±2.4 | 54.4±2.2 | 54.7±3.6 | 47.3±2.7 | 54.6±2.5 | 53.7±1.8 | 53.1±1.8 | 56.0±2.7 | 40.1±2.8 | 53.1±4.9 |
| Qwen3-30B-A3B-Thk-2507* | 63.9±3.9 | 65.9±3.2 | 63.8±3.7 | 45.2±4.1 | 49.5±3.2 | 45.0±3.9 | 57.6±3.5 | 51.2±3.3 | 42.4±3.4 | 57.3±2.9 | 51.6±3.6 | 40.5±1.5 | 52.8±8.8 |
| Qwen3-32B* | 59.0±3.2 | 56.8±2.6 | 55.8±2.4 | 42.7±2.5 | 51.8±2.0 | 51.7±3.1 | 50.6±3.2 | 38.8±2.5 | 51.0±2.8 | 51.7±4.8 | 46.5±3.7 | 35.7±3.6 | 49.3±7.1 |
| Qwen3-235B-A22B* | 59.2±1.7 | 58.5±2.1 | 56.6±2.2 | 48.0±3.4 | 51.5±2.6 | 48.6±2.3 | 50.0±2.8 | 43.4±2.3 | 47.3±3.0 | 47.9±1.8 | 46.6±2.8 | 34.2±2.6 | 49.3±6.9 |
| Qwen3-30B-A3B* | 57.4±2.7 | 52.2±2.3 | 52.1±2.5 | 36.7±3.2 | 50.1±3.2 | 50.4±2.5 | 44.7±2.0 | 39.4±2.7 | 47.0±2.3 | 48.9±4.6 | 44.9±3.2 | 33.8±1.3 | 46.5±6.9 |
| Qwen3-14B* | 55.3±2.1 | 50.6±2.0 | 50.5±2.2 | 34.7±4.4 | 50.5±1.9 | 50.2±1.8 | 45.4±1.2 | 31.5±2.0 | 47.2±2.5 | 46.3±3.0 | 37.8±1.9 | 32.4±2.5 | 44.4±8.1 |
| Gpt-oss-20B* (Low) | 47.3±2.4 | 45.2±2.1 | 42.9±2.9 | 41.4±2.9 | 42.6±2.2 | 42.1±3.1 | 38.5±2.9 | 43.0±2.4 | 43.4±2.2 | 44.3±2.0 | 42.9±2.2 | 31.4±2.3 | 42.1±4.0 |
| Qwen3-235B-A22B-Instr-2507 | 44.3±1.6 | 42.8±3.6 | 45.7±3.2 | 36.7±2.0 | 27.3±2.5 | 23.9±4.5 | 43.5±2.7 | 38.5±3.0 | 42.4±2.1 | 43.4±3.1 | 42.9±2.1 | 31.2±2.7 | 38.5±7.3 |
| Qwen3-8B* | 50.8±3.1 | 44.5±2.3 | 42.7±3.3 | 30.2±4.3 | 41.6±3.1 | 42.0±3.0 | 40.8±2.4 | 35.9±2.4 | 42.8±1.8 | 35.9±2.4 | 27.7±2.8 | 29.9±2.6 | 37.8±8.1 |
| Qwen3-30B-A3B-Instr-2507 | 40.5±2.0 | 35.6±1.5 | 37.4±1.8 | 26.0±1.9 | 20.8±1.5 | 20.6±1.8 | 36.6±2.4 | 31.3±2.1 | 34.5±2.9 | 34.7±2.1 | 34.8±1.7 | 25.9±3.2 | 31.6±6.6 |
| Qwen3-Coder-30B-A3B-Instr | 36.6±2.1 | 32.8±1.9 | 35.0±2.8 | 25.4±1.9 | 26.3±3.2 | 27.1±3.2 | 34.3±2.1 | 34.3±2.1 | 34.5±2.9 | 29.5±2.2 | 34.7±1.8 | 19.2±2.0 | 30.6±5.1 |
| DeepSeek-R1-Distill-Qwen-32B* | 47.6±2.3 | 24.9±2.4 | 38.7±2.2 | 14.7±4.0 | 19.6±2.0 | 15.9±3.6 | 33.4±4.4 | 22.7±3.3 | 44.3±3.5 | 13.7±2.0 | 23.1±2.4 | 9.2±1.3 | 25.6±12.6 |
| Qwen2.5-Coder-32B-Instr | 26.9±2.2 | 25.2±1.4 | 27.5±2.4 | 24.2±1.8 | 10.0±2.5 | 25.3±1.7 | 28.1±1.7 | 24.7±1.1 | 24.6±2.5 | 23.7±2.4 | 25.4±2.9 | 23.1±2.5 | 24.1±4.7 |
| OpenRsn-Nmt-32B* | 66.1±3.0 | 44.1±2.1 | 38.4±3.1 | 11.6±3.0 | 11.4±6.2 | 8.2±4.7 | 29.7±1.9 | 2.4±2.2 | 17.6±3.8 | 14.9±1.7 | 16.0±2.3 | 6.6±2.4 | 22.3±18.8 |
| Seed-Coder-8B-Instr | 21.2±2.4 | 21.8±2.3 | 21.9±2.2 | 18.2±1.1 | 20.9±2.3 | 19.8±2.3 | 20.6±2.0 | 20.2±3.0 | 21.7±1.9 | 19.1±2.8 | 21.8±1.7 | 21.2±2.3 | 20.7±1.2 |
| Qwen2.5-Coder-14B-Instr | 22.1±2.2 | 22.5±1.8 | 23.0±1.5 | 17.5±2.1 | 21.4±1.9 | 18.5±2.6 | 22.9±1.9 | 16.8±1.7 | 21.7±1.5 | 19.3±2.6 | 22.6±1.7 | 18.8±1.8 | 20.6±2.3 |
| OpenCodeRsn-Nmt-1.1-32B* | 63.5±2.8 | 41.7±3.6 | 30.5±3.7 | 8.2±4.6 | 8.5±6.1 | 5.5±5.0 | 22.7±3.5 | 1.9±0.7 | 23.2±3.2 | 17.7±3.4 | 12.7±1.9 | 5.7±2.6 | 20.2±18.0 |
| Devstral-Small-2505* | 23.3±2.0 | 21.2±2.7 | 22.4±1.3 | 13.6±2.4 | 20.8±1.8 | 20.5±2.2 | 20.8±1.6 | 18.9±1.9 | 20.2±1.8 | 21.4±1.4 | 15.2±1.9 | | 19.8±2.8 |
| DeepSeek-R1-Distill-Qwen-14B* | 45.0±2.8 | 18.1±2.6 | 22.7±3.8 | 9.2±2.0 | 9.5±2.6 | 10.0±1.5 | 28.2±1.9 | 4.5±1.4 | 34.8±3.0 | 3.8±1.0 | 12.8±3.4 | 3.8±1.8 | 16.9±13.3 |
| Deepseek-Coder-33B-Instr | 16.9±1.6 | 17.9±1.7 | 18.9±1.4 | 12.9±1.7 | 10.7±3.1 | 8.7±2.5 | 16.8±2.4 | 2.3±1.3 | 14.4±1.4 | 12.8±1.3 | 16.1±1.5 | 7.8±1.7 | 13.0±4.9 |

### F.2.2 PASS@5 PERFORMANCE AT DIFFERENT SAMPLING TEMPERATURES

Table 7, Table 8 and Table 9 reports Pass@5 scores at sampling temperatures $t = 0.2$, $t = 0.6$ and $t = 1.0$ respectively. Each score indicates the percentage of problems solved correctly on the 5th attempt, with higher values reflecting better performance.

Table 7: Performance results at temperature $t = 0.2$. Scores represent the **Pass@5 (%)** metric. Higher is better, **bold** is best, *italic* is the second best. (* - reasoning mode)

| Model | Python | C++ | Java | Go | JS | TS | C# | Rust | Ruby | PHP | Kotlin | Scala | Avg |
|---|---|---|---|---|---|---|---|---|---|---|---|---|---|
| Gpt-oss-120B* (Medium) | 83.7 | 83.7 | 85.1 | 83.8 | 85.2 | 82.6 | 77.6 | 85.0 | 85.2 | 80.4 | 85.0 | 78.7 | 83.0 |
| Qwen3-235B-A22B-Thk-2507* | 83.6 | 86.2 | 85.8 | 78.8 | 80.0 | 80.4 | 81.8 | 75.3 | 65.0 | 81.1 | 84.5 | 79.1 | 80.1 |
| DeepSeek-R1-0528* | 78.3 | 79.7 | 79.9 | 72.9 | 77.3 | 78.2 | 77.5 | 77.9 | 75.7 | 77.0 | 79.1 | 78.4 | 77.7 |
| Gpt-oss-20B* (Medium) | 77.3 | 80.4 | 78.1 | 78.9 | 77.7 | 78.8 | 73.1 | 76.0 | 75.9 | 74.9 | 79.3 | 66.6 | 76.4 |
| Qwen3-30B-A3B-Thk-2507* | 77.5 | 79.5 | 77.4 | 65.4 | 73.4 | 72.4 | 71.9 | 72.5 | 57.3 | 73.9 | 67.6 | 62.0 | 70.9 |
| Gpt-oss-120B* (Low) | 69.1 | 69.4 | 70.3 | 65.9 | 70.3 | 70.5 | 62.6 | 68.2 | 65.5 | 65.8 | 69.1 | 63.6 | 67.5 |
| Qwen3-235B-A22B* | 69.2 | 70.4 | 70.1 | 66.4 | 69.0 | 69.5 | 66.9 | 66.8 | 58.1 | 66.5 | 65.8 | 55.1 | 66.2 |
| Qwen3-32B* | 68.8 | 68.7 | 70.5 | 61.9 | 67.3 | 67.3 | 65.4 | 61.8 | 64.7 | 63.8 | 67.6 | 57.7 | 65.5 |
| Qwen3-30B-A3B* | 66.9 | 64.5 | 63.2 | 56.3 | 62.8 | 62.1 | 59.1 | 54.1 | 58.0 | 61.5 | 59.8 | 47.0 | 59.6 |
| Qwen3-14B* | 66.4 | 59.3 | 61.7 | 54.4 | 60.4 | 62.9 | 57.7 | 48.0 | 59.7 | 58.9 | 60.9 | 47.0 | 58.1 |
| Gpt-oss-20B* (Low) | 59.6 | 61.1 | 59.3 | 55.8 | 56.2 | 54.2 | 53.8 | 54.6 | 55.4 | 57.8 | 59.8 | 48.5 | 56.3 |
| Qwen3-8B* | 56.0 | 51.8 | 51.0 | 44.6 | 52.0 | 52.9 | 49.3 | 36.3 | 51.2 | 48.3 | 38.7 | 39.0 | 47.6 |
| Qwen3-235B-A22B-Instr-2507 | 53.0 | 53.9 | 58.5 | 45.4 | 36.8 | 31.6 | 52.7 | 50.1 | 48.0 | 52.1 | 49.7 | 39.1 | 47.6 |
| OpenRsn-Nmt-32B* | 78.4 | 69.2 | 66.2 | 30.1 | 33.8 | 30.1 | 54.8 | 10.9 | 40.2 | 42.1 | 39.5 | 20.2 | 43.0 |
| OlympicCoder-7B* | 49.6 | 49.1 | 45.9 | 38.9 | 44.0 | 43.6 | 43.3 | 29.1 | 44.5 | 39.1 | 41.9 | 32.2 | 41.8 |
| Qwen3-30B-A3B-Instr-2507 | 49.0 | 44.9 | 46.9 | 31.0 | 32.4 | 27.5 | 47.0 | 42.2 | 43.6 | 42.9 | 45.5 | 34.4 | 40.6 |
| Qwen3-Coder-30B-A3B-Instr | 43.0 | 39.5 | 42.1 | 34.0 | 36.9 | 34.3 | 43.2 | 34.1 | 42.3 | 38.3 | 43.4 | 29.0 | 38.8 |
| DeepSeek-R1-Distill-Qwen-32B* | 51.4 | 35.5 | 47.1 | 27.8 | 39.3 | 32.2 | 43.7 | 38.7 | 49.2 | 33.5 | 40.8 | 19.6 | 38.2 |
| OpenCodeRsn-Nmt-1.1-32B* | 74.5 | 59.8 | 58.7 | 24.5 | 26.6 | 18.2 | 47.7 | 3.8 | 44.4 | 42.5 | 29.7 | 21.3 | 37.7 |
| Qwen2.5-Coder-32B-Instr | 33.0 | 31.3 | 36.1 | 31.6 | 13.4 | 31.2 | 34.5 | 31.3 | 30.6 | 31.0 | 33.0 | 28.9 | 30.5 |
| DeepSeek-R1-Distill-Qwen-14B* | 51.4 | 30.3 | 42.0 | 21.0 | 28.0 | 29.6 | 44.1 | 10.9 | 47.2 | 11.1 | 28.8 | 10.5 | 29.6 |
| Seed-Coder-8B-Instr | 27.8 | 26.0 | 31.2 | 26.6 | 29.3 | 27.5 | 31.7 | 26.6 | 26.6 | 26.5 | 26.7 | 26.2 | 27.7 |
| Devstral-Small-2505* | 27.3 | 26.9 | 25.3 | 24.0 | 27.0 | 29.2 | 27.0 | 24.8 | 26.3 | 26.8 | 25.7 | 22.1 | 26.0 |
| Qwen2.5-Coder-14B-Instr | 22.9 | 25.9 | 28.7 | 25.8 | 26.5 | 24.5 | 28.0 | 22.4 | 29.0 | 26.6 | 25.4 | 24.6 | 25.8 |
| DeepSeek-Coder-33B-Instr | 21.5 | 22.3 | 24.1 | 19.0 | 14.7 | 13.6 | 22.9 | 8.8 | 23.2 | 23.5 | 22.7 | 18.8 | 19.6 |

Table 8: Performance results at temperature $t = 0.6$. Scores represent the **Pass@5 (%)** metric. Higher is better, **bold** is best, *italic* is the second best. (* - reasoning mode)

| Model | Python | C++ | Java | Go | JS | TS | C# | Rust | Ruby | PHP | Kotlin | Scala | Avg |
|---|---|---|---|---|---|---|---|---|---|---|---|---|---|
| Gpt-oss-120B* (Medium) | 84.7 | 84.9 | 84.8 | 85.1 | 84.1 | 85.2 | 80.7 | 84.1 | 83.3 | 82.1 | 83.7 | 80.2 | 83.6 |
| Qwen3-235B-A22B-Thk-2507* | 85.0 | 85.7 | 86.4 | 78.0 | 80.8 | 81.3 | 81.4 | 78.4 | 68.4 | 79.3 | 82.7 | 79.6 | 80.6 |
| Gpt-oss-20B* (Medium) | 79.2 | 81.5 | 80.1 | 79.5 | 80.7 | 82.5 | 75.5 | 79.8 | 79.1 | 77.9 | 79.8 | 65.8 | 78.4 |
| DeepSeek-R1-0528* | 79.6 | 79.1 | 78.8 | 72.4 | 77.5 | 76.0 | 78.1 | 79.1 | 76.3 | 75.9 | 79.5 | 77.6 | 77.5 |
| Qwen3-30B-A3B-Thk-2507* | 78.3 | 79.3 | 78.5 | 67.6 | 75.2 | 73.0 | 72.2 | 69.9 | 62.4 | 71.8 | 70.3 | 59.1 | 71.5 |
| Gpt-oss-120B* (Low) | 70.5 | 68.6 | 70.5 | 68.5 | 68.1 | 68.7 | 63.4 | 71.5 | 69.6 | 65.7 | 68.1 | 65.0 | 68.2 |
| Qwen3-235B-A22B* | 71.1 | 69.4 | 70.7 | 66.2 | 70.3 | 67.7 | 67.1 | 67.0 | 60.6 | 67.0 | 66.9 | 57.1 | 66.8 |
| Qwen3-32B* | 70.1 | 68.7 | 69.5 | 65.3 | 67.4 | 69.8 | 66.0 | 61.1 | 65.8 | 67.1 | 68.8 | 58.5 | 66.5 |
| Qwen3-14B* | 68.4 | 63.9 | 63.6 | 57.0 | 63.6 | 62.8 | 60.4 | 50.1 | 60.4 | 58.7 | 61.3 | 47.6 | 59.8 |
| Qwen3-30B-A3B* | 64.4 | 65.0 | 64.8 | 56.9 | 63.9 | 62.4 | 58.1 | 53.7 | 59.9 | 60.1 | 59.1 | 48.4 | 59.7 |
| Gpt-oss-20B* (Low) | 59.9 | 61.1 | 59.1 | 57.3 | 56.6 | 56.8 | 54.6 | 58.1 | 58.4 | 55.7 | 58.4 | 53.3 | 57.4 |
| Qwen3-8B* | 61.0 | 55.4 | 57.4 | 51.0 | 56.2 | 56.6 | 51.8 | 39.3 | 55.4 | 49.3 | 41.4 | 43.5 | 51.5 |
| Qwen3-235B-A22B-Instr-2507 | 54.6 | 54.4 | 58.9 | 50.3 | 44.2 | 39.5 | 53.5 | 51.0 | 49.0 | 55.2 | 51.2 | 43.0 | 50.4 |
| Qwen3-30B-A3B-Instr-2507 | 50.3 | 47.1 | 49.0 | 39.0 | 36.8 | 38.5 | 48.4 | 44.6 | 42.8 | 45.5 | 47.9 | 38.0 | 44.0 |
| OpenRsn-Nmt-32B* | 77.8 | 71.6 | 67.4 | 28.3 | 35.3 | 36.3 | 57.1 | 10.7 | 41.2 | 39.8 | 36.9 | 23.5 | 43.8 |
| DeepSeek-R1-Distill-Qwen-32B* | 58.3 | 42.8 | 53.4 | 31.8 | 45.1 | 36.2 | 54.4 | 41.4 | 53.4 | 40.2 | 43.8 | 24.4 | 43.8 |
| OlympicCoder-7B* | 51.5 | 49.5 | 44.8 | 39.7 | 46.3 | 43.2 | 43.4 | 31.2 | 40.0 | 40.0 | 38.0 | 29.4 | 41.4 |
| OpenCodeRsn-Nmt-1.1-32B* | 75.6 | 63.4 | 60.4 | 26.1 | 32.7 | 25.4 | 49.2 | 5.6 | 46.0 | 46.6 | 32.3 | 22.7 | 40.5 |
| Qwen3-Coder-30B-A3B-Instr | 42.4 | 40.3 | 41.3 | 34.0 | 38.9 | 37.7 | 42.4 | 41.7 | 41.2 | 43.0 | 43.5 | 30.1 | 39.7 |
| Qwen2.5-Coder-32B-Instr | 34.3 | 33.0 | 36.3 | 33.5 | 17.8 | 34.0 | 37.7 | 32.2 | 32.9 | 34.5 | 35.9 | 32.3 | 32.9 |
| DeepSeek-R1-Distill-Qwen-14B* | 55.5 | 35.1 | 39.8 | 19.4 | 28.1 | 28.0 | 46.0 | 13.2 | 50.6 | 15.6 | 31.6 | 11.2 | 31.2 |
| Seed-Coder-8B-Instr | 32.7 | 29.4 | 30.3 | 28.3 | 32.0 | 31.2 | 31.1 | 32.1 | 31.1 | 28.8 | 31.3 | 27.8 | 30.5 |
| Devstral-Small-2505* | 30.8 | 29.6 | 29.1 | 28.5 | 30.8 | 33.3 | 29.6 | 29.1 | 29.5 | 29.5 | 29.4 | 25.2 | 29.5 |
| Qwen2.5-Coder-14B-Instr | 29.0 | 27.9 | 29.9 | 26.9 | 28.9 | 27.8 | 31.6 | 25.8 | 29.4 | 28.3 | 30.0 | 27.3 | 28.6 |
| DeepSeek-Coder-33B-Instr | 26.7 | 25.6 | 27.0 | 21.8 | 20.8 | 22.4 | 26.8 | 7.4 | 26.2 | 24.9 | 23.5 | 21.8 | 22.9 |

Table 9: Performance results at temperature $t = 1.0$. Scores represent the **Pass@5 (%)** metric. Higher is better, **bold** is best, *italic* is the second best. (* - reasoning mode)

| Model | Python | C++ | Java | Go | JS | TS | C# | Rust | Ruby | PHP | Kotlin | Scala | Avg |
|---|---|---|---|---|---|---|---|---|---|---|---|---|---|
| Gpt-oss-120B* (Medium) | 82.8 | 83.8 | 85.9 | 84.1 | 84.6 | 83.9 | 78.0 | 83.5 | 83.6 | 84.0 | 85.2 | 78.1 | 83.1 |
| Qwen3-235B-A22B-Thk-2507* | 85.2 | 85.2 | 85.9 | 77.4 | 81.6 | 82.4 | 82.4 | 82.0 | 67.8 | 79.8 | 84.0 | 79.0 | 81.1 |
| Gpt-oss-20B (Medium)* | 79.4 | 80.5 | 81.0 | 77.5 | 79.2 | 80.5 | 75.2 | 77.5 | 78.2 | 77.4 | 82.6 | 66.9 | 78.0 |
| DeepSeek-R1-0528* | 72.3 | 74.3 | 73.9 | 67.6 | 72.5 | 71.8 | 73.2 | 70.8 | 69.4 | 72.3 | 68.7 | 71.7 | 71.5 |
| Qwen3-30B-A3B-Thk-2507* | 76.7 | 79.9 | 77.8 | 68.5 | 72.9 | 73.3 | 73.1 | 70.0 | 60.0 | 71.9 | 68.5 | 60.5 | 71.1 |
| Gpt-oss-120B* (Low) | 71.8 | 71.7 | 71.0 | 67.8 | 68.7 | 69.7 | 65.3 | 68.7 | 67.7 | 65.1 | 69.6 | 63.4 | 68.4 |
| Qwen-3-32B* | 73.4 | 70.1 | 70.3 | 64.1 | 70.9 | 69.4 | 66.2 | 61.8 | 65.3 | 65.5 | 67.5 | 58.9 | 67.0 |
| Qwen-3-235B-A22B* | 70.0 | 71.4 | 70.0 | 65.8 | 69.7 | 70.5 | 66.0 | 66.6 | 61.7 | 65.4 | 66.3 | 54.9 | 66.5 |
| Qwen-3-14B* | 68.4 | 63.5 | 66.1 | 57.9 | 67.5 | 64.6 | 59.5 | 50.7 | 60.3 | 61.2 | 60.4 | 50.7 | 60.9 |
| Qwen-3-30B-A3B* | 66.3 | 63.7 | 64.8 | 59.1 | 64.7 | 63.8 | 59.0 | 57.7 | 57.7 | 62.6 | 61.4 | 49.8 | 60.9 |
| Gpt-oss-20B* (Low) | 60.0 | 59.3 | 59.7 | 54.9 | 55.6 | 55.6 | 55.6 | 58.6 | 58.0 | 56.8 | 57.7 | 51.4 | 56.9 |
| Qwen-3-8B* | 63.1 | 56.7 | 58.7 | 52.8 | 59.3 | 57.5 | 56.3 | 42.6 | 56.8 | 50.9 | 44.8 | 43.1 | 53.5 |
| Qwen3-235B-A22B-Instr-2507 | 54.8 | 56.1 | 59.6 | 51.1 | 44.1 | 46.1 | 53.7 | 49.4 | 52.3 | 55.9 | 52.9 | 44.9 | 51.8 |
| DeepSeek-R1-Distill-Qwen-32B* | 59.6 | 47.0 | 55.4 | 35.5 | 45.6 | 43.0 | 53.2 | 44.9 | 56.6 | 39.1 | 47.7 | 25.6 | 46.1 |
| Qwen3-30B-A3B-Instr-2507 | 52.0 | 48.8 | 49.4 | 40.2 | 38.1 | 39.3 | 47.5 | 44.9 | 45.5 | 48.8 | 47.0 | 39.7 | 45.1 |
| OpenRsn-Nmt-32B* | 78.8 | 69.6 | 64.1 | 29.3 | 35.2 | 27.2 | 53.3 | 9.5 | 41.9 | 42.9 | 39.7 | 22.4 | 42.8 |
| OlympicCoder-7B* | 47.6 | 47.1 | 46.7 | 36.7 | 43.1 | 42.2 | 43.3 | 28.2 | 42.3 | 39.1 | 37.8 | 29.2 | 40.3 |
| Qwen3-Coder-30B-A3B-Instr | 44.5 | 40.9 | 43.3 | 34.4 | 37.6 | 39.3 | 43.0 | 42.1 | 40.0 | 40.0 | 44.3 | 32.6 | 40.2 |
| OpenCodeRsn-Nmt-1.1-32B* | 77.3 | 66.1 | 60.6 | 24.4 | 29.3 | 21.1 | 44.8 | 8.3 | 44.1 | 44.5 | 31.5 | 19.4 | 39.3 |
| Qwen2.5-Coder-32B-Instr | 34.7 | 34.0 | 39.8 | 33.6 | 26.1 | 36.7 | 36.6 | 33.6 | 36.3 | 34.4 | 35.6 | 33.1 | 34.5 |
| DeepSeek-R1-Distill-Qwen-14B* | 56.4 | 35.2 | 41.9 | 21.5 | 29.7 | 31.0 | 44.1 | 14.7 | 50.0 | 15.5 | 32.5 | 13.6 | 32.2 |
| Seed-Coder-8B-Instr | 32.0 | 29.3 | 30.6 | 28.0 | 30.7 | 29.6 | 31.7 | 30.8 | 31.9 | 31.8 | 30.6 | 31.9 | 30.7 |
| Qwen2.5-Coder-14B-Instr | 29.7 | 30.2 | 30.8 | 28.0 | 29.8 | 29.0 | 31.7 | 28.6 | 31.8 | 29.6 | 31.0 | 28.5 | 29.9 |
| Devstral-Small-2505* | 33.4 | 28.5 | 30.1 | 26.2 | 32.8 | 30.6 | 30.8 | 27.7 | 28.1 | 30.7 | 30.4 | 25.8 | 29.6 |
| Deepseek-Coder-33B-Instr | 25.7 | 27.2 | 27.8 | 23.3 | 22.3 | 21.3 | 26.0 | 8.8 | 24.4 | 20.5 | 23.8 | 17.9 | 22.4 |

### F.2.3   PASS@10 PERFORMANCE AT DIFFERENT SAMPLING TEMPERATURES

Table 10, Table 11 and Table 12 reports Pass@10 scores at sampling temperatures $t = 0.2$, $t = 0.6$ and $t = 1.0$ respectively. Each score indicates the percentage of problems solved correctly on the 10th attempt, with higher values reflecting better performance.

Table 10: Performance results at temperature $t = 0.2$. Scores represent the **Pass@10 (%)** metric. Higher is better, **bold** is best, *italic* is the second best. (* - Rsn mode)

| Model | Python | C++ | Java | Go | JS | TS | C# | Rust | Ruby | PHP | Kotlin | Scala | Avg |
|---|---|---|---|---|---|---|---|---|---|---|---|---|---|
| Gpt-oss-120B* (Medium) | 87.0 | 86.3 | 87.8 | 86.3 | 87.0 | 84.7 | 84.0 | 87.8 | 88.6 | 84.0 | 87.0 | 84.0 | 86.2 |
| Qwen3-235B-A22B-Thinking-2507* | 87.0 | 88.6 | 88.6 | 84.0 | 84.0 | 85.5 | 85.5 | 82.4 | 71.8 | 84.7 | 87.8 | 84.0 | 84.5 |
| Gpt-oss-20B* (Medium) | 80.2 | 84.0 | 82.4 | 84.0 | 81.7 | 84.7 | 77.9 | 80.9 | 81.7 | 79.4 | 83.2 | 76.3 | 81.4 |
| DeepSeek-R1-0528* | 80.9 | 82.4 | 83.2 | 78.6 | 80.2 | 83.2 | 80.9 | 80.9 | 80.9 | 80.9 | 81.7 | 81.7 | 81.3 |
| Qwen3-30B-A3B-Thinking-2507* | 80.9 | 82.4 | 79.4 | 68.7 | 77.9 | 77.9 | 74.8 | 77.9 | 62.6 | 79.4 | 73.3 | 67.2 | 75.2 |
| Gpt-oss-120B* (Low) | 73.3 | 72.5 | 74.1 | 70.2 | 75.6 | 74.8 | 68.7 | 71.8 | 68.7 | 70.2 | 72.5 | 65.7 | 71.7 |
| Qwen3-235B-A22B* | 72.5 | 74.8 | 73.3 | 69.5 | 71.8 | 74.8 | 70.2 | 72.5 | 61.1 | 72.5 | 71.0 | 61.8 | 70.5 |
| Qwen3-32B* | 72.5 | 73.3 | 74.1 | 67.2 | 72.5 | 73.3 | 69.5 | 67.9 | 67.9 | 68.7 | 73.3 | 63.4 | 70.3 |
| Qwen3-30B-A3B* | 69.5 | 67.2 | 66.4 | 61.1 | 65.7 | 64.9 | 63.4 | 58.8 | 62.6 | 64.9 | 63.4 | 51.9 | 63.3 |
| Qwen3-14B* | 71.0 | 63.4 | 66.4 | 60.3 | 64.9 | 67.9 | 63.4 | 54.2 | 64.9 | 63.4 | 65.7 | 51.2 | 63.0 |
| Gpt-oss-20B* (Low) | 64.1 | 64.9 | 63.4 | 60.3 | 60.3 | 58.8 | 57.3 | 58.8 | 59.5 | 61.1 | 64.9 | 55.7 | 60.8 |
| OpenRsn-Nmt-32B* | 84.0 | 76.3 | 74.1 | 39.7 | 45.8 | 40.5 | 65.7 | 18.3 | 39.6 | 55.0 | 49.6 | 30.5 | 52.4 |
| Qwen3-8B* | 60.3 | 56.5 | 55.7 | 51.2 | 56.5 | 56.5 | 52.7 | 41.2 | 56.5 | 51.9 | 43.5 | 42.8 | 52.1 |
| Qwen3-235B-A22B-Instr-2507 | 55.7 | 58.0 | 63.4 | 48.1 | 41.2 | 37.4 | 55.0 | 52.7 | 51.2 | 55.7 | 52.7 | 42.8 | 51.2 |
| OlympicCoder-7B* | 53.4 | 53.4 | 51.2 | 44.3 | 48.9 | 47.3 | 48.1 | 37.4 | 50.4 | 45.0 | 45.8 | 38.9 | 47.0 |
| OpenCodeRsn-Nmt-1.1-32B* | 78.6 | 67.9 | 65.7 | 32.1 | 37.4 | 26.7 | 58.0 | 5.3 | 54.2 | 45.0 | 45.8 | 28.2 | 45.4 |
| Qwen3-30B-A3B-Instr-2507 | 51.9 | 48.9 | 51.2 | 33.6 | 37.4 | 32.8 | 51.9 | 46.6 | 45.8 | 46.6 | 49.6 | 37.4 | 44.5 |
| DeepSeek-R1-Distill-Qwen-32B* | 55.0 | 38.9 | 48.9 | 35.1 | 48.1 | 42.8 | 46.6 | 44.3 | 52.7 | 41.2 | 47.3 | 26.7 | 44.0 |
| Qwen3-Coder-30B-A3B-Instr | 45.0 | 42.0 | 43.5 | 36.6 | 38.9 | 36.6 | 45.8 | 41.2 | 43.5 | 39.7 | 45.8 | 33.6 | 41.0 |
| DeepSeek-R1-Distill-Qwen-14B* | 54.2 | 35.9 | 45.8 | 26.0 | 35.9 | 38.2 | 48.9 | 15.3 | 51.9 | 15.3 | 35.9 | 14.5 | 34.8 |
| Qwen2.5-Coder-32B-Instr | 36.6 | 32.8 | 40.5 | 35.1 | 14.5 | 32.8 | 35.9 | 34.4 | 32.8 | 33.6 | 36.6 | 31.3 | 33.1 |
| Seed-Coder-8B-Instr | 29.8 | 26.7 | 32.8 | 27.5 | 32.1 | 29.0 | 33.6 | 28.2 | 28.2 | 28.2 | 27.5 | 28.2 | 29.3 |
| Devstral-Small-2505* | 29.0 | 27.5 | 26.0 | 26.7 | 28.2 | 30.5 | 27.5 | 27.5 | 29.8 | 28.2 | 27.5 | 24.4 | 27.7 |
| Qwen2.5-Coder-14B-Instr | 23.7 | 27.5 | 29.8 | 27.5 | 27.5 | 26.0 | 29.8 | 24.4 | 30.5 | 29.0 | 27.5 | 25.2 | 27.4 |
| DeepSeek-Coder-33B-Instr | 22.9 | 23.7 | 24.4 | 20.6 | 17.6 | 16.0 | 25.2 | 10.7 | 25.2 | 26.0 | 23.7 | 21.4 | 21.4 |

Table 11: Performance results at temperature $t = 0.6$. Scores represent the **Pass@10 (%)** metric. Higher is better, **bold** is best, *italic* is the second best. (* - Rsn mode)

| Model | Python | C++ | Java | Go | JS | TS | C# | Rust | Ruby | PHP | Kotlin | Scala | Avg |
|---|---|---|---|---|---|---|---|---|---|---|---|---|---|
| Gpt-oss-120B* (Medium) | 86.7 | 87.8 | 84.7 | 87.8 | 87.8 | 87.8 | 87.0 | 86.3 | 86.3 | 85.5 | 86.3 | 85.5 | 87.8 |
| Qwen3-235B-A22B-Thinking-2507* | 84.5 | 87.0 | 86.3 | 87.8 | 84.0 | 90.1 | 84.7 | 84.7 | 82.4 | 74.1 | 84.0 | 84.7 | 84.0 |
| Gpt-oss-20B* (Medium) | 83.0 | 83.2 | 83.2 | 84.0 | 83.2 | 83.2 | 84.7 | 81.7 | 85.5 | 84.0 | 73.3 | 73.3 | 87.0 |
| DeepSeek-R1-0528* | 81.2 | 83.2 | 81.7 | 80.9 | 79.4 | 81.7 | 81.7 | 83.2 | 80.9 | 80.2 | 83.2 | 80.2 | 78.6 |
| Qwen3-30B-A3B-Thinking-2507* | 76.1 | 80.9 | 77.1 | 82.4 | 73.3 | 81.7 | 79.4 | 77.1 | 75.6 | 69.5 | 74.8 | 65.7 | 75.6 |
| Gpt-oss-120B* (Low) | 72.7 | 74.1 | 69.5 | 72.5 | 72.5 | 74.1 | 71.8 | 71.8 | 69.5 | 74.8 | 75.6 | 72.5 | 74.1 |
| Qwen3-32B* | 71.7 | 75.6 | 71.0 | 73.3 | 71.8 | 74.8 | 72.5 | 74.8 | 72.5 | 70.2 | 64.9 | 64.4 | 74.8 |
| Qwen3-235B-A22B* | 71.3 | 75.6 | 70.2 | 71.8 | 71.0 | 74.8 | 76.3 | 72.5 | 71.0 | 64.1 | 73.3 | 64.1 | 71.0 |
| Qwen3-14B* | 65.0 | 72.5 | 65.7 | 68.7 | 64.1 | 66.4 | 66.4 | 67.2 | 65.7 | 64.9 | 56.5 | 55.0 | 67.2 |
| Qwen3-30B-A3B* | 63.6 | 66.4 | 62.6 | 66.4 | 62.6 | 69.5 | 67.9 | 61.8 | 63.4 | 64.1 | 58.8 | 53.4 | 65.7 |
| Gpt-oss-20B* (Low) | 61.9 | 63.4 | 59.5 | 64.9 | 61.8 | 61.8 | 61.8 | 61.8 | 61.1 | 62.6 | 62.6 | 59.5 | 61.8 |
| Qwen3-8B* | 56.2 | 64.1 | 55.7 | 60.3 | 57.3 | 62.6 | 60.3 | 47.3 | 51.9 | 60.3 | 44.3 | 48.9 | 61.1 |
| Qwen3-235B-A22B-Instr-2507 | 54.6 | 58.0 | 56.5 | 58.8 | 54.2 | 64.1 | 50.4 | 55.0 | 59.5 | 51.2 | 54.2 | 47.3 | 46.6 |
| OpenRsn-Nmt-32B* | 52.5 | 80.9 | 67.9 | 79.4 | 38.2 | 74.1 | 47.3 | 44.3 | 51.9 | 50.4 | 15.3 | 32.1 | 48.9 |
| DeepSeek-R1-Distill-Qwen-32B* | 50.5 | 63.4 | 59.5 | 48.1 | 40.5 | 55.0 | 54.2 | 51.2 | 50.4 | 57.3 | 48.1 | 33.6 | 44.3 |
| OpenCodeRsn-Nmt-1.1-32B* | 49.1 | 78.6 | 59.5 | 69.5 | 37.4 | 68.7 | 46.6 | 41.2 | 55.0 | 51.2 | 10.7 | 31.3 | 39.7 |
| Qwen3-30B-A3B-Instr-2507 | 48.4 | 52.7 | 52.7 | 51.2 | 45.0 | 54.2 | 42.8 | 51.9 | 48.9 | 45.8 | 48.9 | 41.2 | 45.0 |
| OlympicCoder-7B* | 46.6 | 56.5 | 47.3 | 54.2 | 45.0 | 49.6 | 50.4 | 44.3 | 45.0 | 43.5 | 39.7 | 35.1 | 48.1 |
| Qwen3-Coder-30B-A3B-Instr | 42.1 | 44.3 | 45.0 | 42.8 | 37.4 | 42.0 | 41.2 | 45.8 | 45.8 | 43.5 | 43.5 | 32.8 | 41.2 |
| DeepSeek-R1-Distill-Qwen-14B* | 37.3 | 59.5 | 51.9 | 41.2 | 24.4 | 42.8 | 36.6 | 39.7 | 24.4 | 55.0 | 19.9 | 16.0 | 35.9 |
| Qwen2.5-Coder-32B-Instr | 36.3 | 37.4 | 40.5 | 35.1 | 36.6 | 39.7 | 21.4 | 38.9 | 37.4 | 38.9 | 35.1 | 36.6 | 37.4 |
| Seed-Coder-8B-Instr | 33.5 | 36.6 | 32.1 | 32.8 | 32.1 | 31.3 | 35.9 | 35.1 | 31.3 | 35.1 | 35.9 | 29.8 | 33.6 |
| Devstral-Small-2505* | 32.5 | 35.9 | 32.8 | 31.3 | 32.1 | 31.3 | 34.4 | 32.1 | 31.3 | 31.3 | 33.6 | 28.2 | 35.1 |
| Qwen2.5-Coder-14B-Instr | 30.9 | 30.5 | 34.4 | 30.5 | 29.0 | 31.3 | 30.5 | 32.1 | 31.3 | 32.1 | 28.2 | 30.5 | 30.5 |
| DeepSeek-Coder-33B-Instr | 25.7 | 29.8 | 29.0 | 27.5 | 25.2 | 29.0 | 24.4 | 26.0 | 28.2 | 28.2 | 9.9 | 23.7 | 27.5 |

Table 12: Performance results at temperature $t = 1.0$. Scores represent the **Pass@10 (%)** metric. Higher is better, **bold** is best, *italic* is the second best. (* - Rsn mode)

| Model | Python | C++ | Java | Go | JS | TS | C# | Rust | Ruby | PHP | Kotlin | Scala | Avg |
|---|---|---|---|---|---|---|---|---|---|---|---|---|---|
| Gpt-oss-120B* (Medium) | **86.3** | 85.5 | 80.9 | *87.0* | **87.0** | 89.3 | **87.8** | 87.0 | **87.0** | **86.3** | **86.3** | **84.0** | **87.0** |
| Qwen3-235B-A22B-Thk-2507* | *84.8* | **90.1** | **85.5** | **87.8** | 80.2 | **90.1** | 85.5 | 87.0 | 82.4 | 74.1 | 85.5 | *84.0* | 85.5 |
| Gpt-oss-20B* (Medium) | 82.6 | 82.4 | *81.7* | 84.7 | 82.4 | 85.5 | 84.7 | 86.3 | 80.2 | *81.7* | 80.9 | 75.6 | 84.7 |
| DeepSeek-R1-0528* | 75.9 | 77.9 | 77.9 | 79.4 | 74.8 | 77.9 | 76.3 | 71.8 | 76.3 | 73.3 | 73.3 | 76.3 | 75.6 |
| Qwen3-30B-A3B-Thk-2507* | 75.4 | 78.6 | 75.6 | 83.2 | 74.1 | 81.7 | 77.1 | 73.3 | 74.8 | 65.7 | 74.1 | 68.7 | 77.9 |
| Gpt-oss-120B* (Low) | 73.1 | 74.8 | 71.0 | 76.3 | 72.5 | 75.6 | 73.3 | 74.8 | 68.7 | 71.8 | 73.3 | 69.5 | 75.6 |
| Qwen3-32B* | 71.9 | 76.3 | 71.0 | 74.8 | 71.0 | 74.8 | 76.3 | 71.0 | 71.0 | 70.2 | 67.2 | 64.9 | 73.3 |
| Qwen3-235B-A22B* | 70.8 | 72.5 | 71.0 | 74.8 | 71.0 | 74.1 | 74.1 | 71.0 | 68.7 | 66.4 | 71.0 | 60.3 | 74.8 |
| Qwen3-14B* | 65.7 | 72.5 | 64.9 | 67.2 | 64.1 | 71.0 | 71.8 | 65.7 | 66.4 | 64.1 | 56.5 | 55.7 | 68.7 |
| Qwen3-30B-A3B* | 64.5 | 68.7 | 61.1 | 65.7 | 64.1 | 67.2 | 68.7 | 64.9 | 66.4 | 61.8 | 62.6 | 54.2 | 68.7 |
| Gpt-oss-20B* (Low) | 61.2 | 64.1 | 60.3 | 63.4 | 60.3 | 64.9 | 58.8 | 61.1 | 61.1 | 61.1 | 64.1 | 55.7 | 59.5 |
| Qwen3-8B* | 58.6 | 67.2 | 60.3 | 60.3 | 59.5 | 64.9 | 64.1 | 51.2 | 56.5 | 61.1 | 48.9 | 47.3 | 61.8 |
| Qwen3-235B-A22B-Instr-2507 | 55.8 | 58.0 | 56.5 | 60.3 | 55.7 | 64.1 | 48.9 | 55.7 | 59.5 | 56.5 | 51.9 | 49.6 | 52.7 |
| DeepSeek-R1-Distill-Qwen-32B* | 53.2 | 64.9 | 58.0 | 55.0 | 44.3 | 59.5 | 54.2 | 53.4 | 48.9 | 60.3 | 51.9 | 33.6 | 54.2 |
| OpenRsn-Nmt-32B* | 51.8 | 82.4 | 61.1 | 77.1 | 38.2 | 71.8 | 46.6 | 51.9 | 52.7 | 52.7 | 15.3 | 32.8 | 38.9 |
| Qwen3-30B-A3B-Instr-2507 | 49.7 | 55.7 | 51.2 | 54.2 | 43.5 | 54.2 | 44.3 | 51.9 | 53.4 | 48.9 | 49.6 | 43.5 | 45.8 |
| OpenCodeRsn-Nmt-1.1-32B* | 47.5 | 81.7 | 51.2 | 71.8 | 35.1 | 71.0 | 41.2 | 38.2 | 54.2 | 53.4 | 13.7 | 26.7 | 31.3 |
| OlympicCoder-7B* | 46.1 | 52.7 | 50.4 | 51.9 | 42.8 | 51.9 | 47.3 | 42.8 | 46.6 | 48.1 | 35.1 | 35.9 | 48.1 |
| Qwen3-Coder-30B-A3B-Instr | 42.8 | 48.1 | 45.8 | 44.3 | 37.4 | 45.0 | 39.7 | 45.8 | 43.5 | 41.2 | 44.3 | 36.6 | 41.2 |
| DeepSeek-R1-Distill-Qwen-14B* | 39.5 | 60.3 | 48.9 | 42.0 | 27.5 | 48.9 | 42.8 | 41.2 | 24.4 | 55.0 | 21.4 | 20.6 | 41.2 |
| Qwen2.5-Coder-32B-Instr | 38.2 | 38.2 | 39.7 | 38.2 | 35.9 | 43.5 | 33.6 | 38.2 | 37.4 | 40.5 | 35.9 | 37.4 | 40.5 |
| Seed-Coder-8B-Instr | 33.8 | 35.1 | 35.1 | 31.3 | 30.5 | 32.8 | 33.6 | 32.8 | 35.1 | 35.1 | 35.1 | 35.1 | 33.6 |
| Qwen2.5-Coder-14B-Instr | 33.2 | 32.8 | 34.4 | 33.6 | 31.3 | 34.4 | 32.8 | 34.4 | 33.6 | 35.1 | 32.1 | 32.8 | 31.3 |
| Devstral-Small-2505* | 33.1 | 38.2 | 34.4 | 30.5 | 30.5 | 33.6 | 36.6 | 34.4 | 34.4 | 32.1 | 29.8 | 29.0 | 33.6 |
| DeepSeek-Coder-33B-Instr | 26.0 | 29.0 | 29.0 | 31.3 | 27.5 | 31.3 | 26.0 | 26.7 | 23.7 | 27.5 | 13.7 | 20.6 | 25.2 |

## F.3 PERFORMANCE ON THE MULTI-LCB (JUL 2024-MAY 2025 SUBSET)

Table 13 reports Pass@1 scores at sampling temperature $t = 0.2$ for all evaluated models on the Multi-LCB subset containing tasks from July 2024 to May 2025. Each score reflects the percentage of problems solved correctly on the first attempt, with higher values indicating better performance.

Table 13: Performance results on Multi-LCB tasks from July 2024 till May 2025. Scores represent the **Pass@1 (%)** metric (higher is better). (* - reasoning mode)

| Model | Python | C++ | Java | Go | JS | TS | C# | Rust | Ruby | PHP | Kotlin | Scala | Avg |
|---|---|---|---|---|---|---|---|---|---|---|---|---|---|
| Qwen3-235B-A22B-Thk-2507* | 76.7 | 78.3 | 78.3 | 59.6 | 71.4 | 62.2 | 69.6 | 50.9 | 54.1 | 70.0 | 64.6 | 56.9 | 66.0 |
| Qwen3-30B-A3B-Thk-2507* | 69.4 | 68.2 | 66.6 | 44.5 | 52.1 | 48.5 | 59.8 | 52.5 | 45.1 | 59.0 | 46.7 | 41.0 | 54.4 |
| Qwen3-235B-A22B* | 65.8 | 63.4 | 59.4 | 49.7 | 56.3 | 51.9 | 57.5 | 45.1 | 51.9 | 52.7 | 48.7 | 32.8 | 52.9 |
| Qwen3-32B* | 63.4 | 63.6 | 59.6 | 43.7 | 51.9 | 54.7 | 53.5 | 41.4 | 53.9 | 52.5 | 42.3 | 35.6 | 51.3 |
| Qwen3-30B-A3B* | 60.8 | 56.5 | 56.3 | 37.0 | 49.9 | 53.1 | 50.5 | 43.1 | 50.5 | 50.7 | 39.2 | 30.4 | 48.2 |
| Qwen3-14B* | 57.5 | 54.7 | 51.7 | 38.4 | 50.7 | 49.9 | 48.5 | 33.4 | 50.1 | 46.9 | 37.6 | 30.2 | 45.8 |
| Qwen3-235B-A22B-Instr-2507 | 49.5 | 47.9 | 47.9 | 38.6 | 29.4 | 22.7 | 46.5 | 42.9 | 41.6 | 43.7 | 42.5 | 29.8 | 40.2 |
| OlympicCoder-32B* | 51.3 | 51.9 | 47.9 | 34.6 | 36.0 | 34.4 | 42.9 | 34.0 | 45.3 | 43.3 | 34.6 | 25.6 | 40.1 |
| Qwen3-8B* | 49.9 | 44.3 | 42.5 | 27.2 | 39.8 | 41.7 | 38.0 | 24.7 | 42.3 | 33.6 | 18.1 | 24.9 | 35.6 |
| Qwen3-30B-A3B-Instr-2507 | 40.6 | 36.4 | 37.8 | 22.1 | 23.7 | 22.5 | 37.4 | 32.4 | 35.0 | 36.8 | 26.8 | 19.7 | 31.0 |
| Qwen3-Coder-30B-A3B-Instr | 35.2 | 30.0 | 33.0 | 21.1 | 27.2 | 26.0 | 34.0 | 31.9 | 34.2 | 31.0 | 33.6 | 18.3 | 25.7 |
| OlympicCoder-7B* | 36.0 | 35.6 | 31.4 | 24.5 | 25.6 | 24.1 | 28.9 | 13.3 | 26.6 | 24.7 | 24.7 | 13.3 | 25.7 |
| OpenRsn-Nemotron-32B* | 69.8 | 49.5 | 41.9 | 12.9 | 13.7 | 10.5 | 33.0 | 1.8 | 20.7 | 20.3 | 16.5 | 7.8 | 24.9 |
| Qwen2.5-Coder-32B-Instr | 27.4 | 27.4 | 30.2 | 24.7 | 6.0 | 28.0 | 27.8 | 25.4 | 24.1 | 27.0 | 26.8 | 22.7 | 24.8 |
| Seed-Coder-8B-Instr | 20.7 | 21.7 | 21.9 | 17.7 | 22.1 | 21.7 | 22.5 | 22.3 | 21.5 | 21.5 | 19.3 | 21.2 | 21.2 |
| Qwen2.5-Coder-14B-Instr | 21.3 | 21.1 | 23.5 | 20.5 | 23.1 | 18.9 | 22.5 | 17.3 | 21.7 | 22.7 | 21.3 | 18.9 | 21.1 |
| Devstral-Small-2505 | 25.4 | 20.9 | 23.7 | 19.1 | 22.9 | 22.7 | 21.1 | 19.5 | 19.1 | 21.3 | 20.7 | 15.1 | 21.0 |
| OpenCodeRsn-Nemotron-1.1-32B* | 62.8 | 38.0 | 30.6 | 8.5 | 8.9 | 8.0 | 27.0 | 1.4 | 23.5 | 24.7 | 12.5 | 5.6 | 21.0 |
| DeepSeek-Coder-33B-Instr | 17.5 | 15.9 | 18.9 | 12.1 | 8.7 | 5.8 | 17.7 | 2.8 | 14.7 | 15.7 | 16.9 | 12.9 | 13.3 |

## F.4 PERFORMANCE ON THE COMPLETE MULTI-LCB BENCHMARK

Table 14: Performance results on Multi-LCB (n=1055 per language). Scores represent the **Pass@1 (%)** metric (higher is better). (* - reasoning mode)

| Model | Python | C++ | Java | Go | JS | TS | C# | Rust | Ruby | PHP | Kotlin | Scala | Avg |
|---|---|---|---|---|---|---|---|---|---|---|---|---|---|
| Qwen3-235B-A22B-Thk-2507* | 85.6 | 86.6 | 85.9 | 60.0 | 80.8 | 73.7 | 79.1 | 58.8 | 65.0 | 78.8 | 75.7 | 66.1 | 74.7 |
| Qwen3-30B-A3B-Thk-2507* | 80.6 | 80.2 | 78.5 | 50.2 | 62.5 | 57.7 | 73.5 | 63.2 | 58.7 | 70.0 | 60.5 | 50.6 | 65.5 |
| Qwen3-235B-A22B* | 77.5 | 72.0 | 72.0 | 60.2 | 64.6 | 63.3 | 68.5 | 56.2 | 65.2 | 62.4 | 53.7 | 40.0 | 62.9 |
| Qwen3-32B* | 77.5 | 70.2 | 73.3 | 51.3 | 64.6 | 64.3 | 67.1 | 51.4 | 68.4 | 64.0 | 54.5 | 48.1 | 62.9 |
| Qwen3-30B-A3B* | 74.8 | 67.9 | 68.2 | 43.9 | 62.5 | 63.2 | 61.4 | 52.8 | 64.3 | 60.8 | 47.1 | 39.9 | 58.9 |
| Qwen3-14B* | 73.4 | 67.2 | 65.9 | 47.8 | 63.5 | 62.9 | 60.7 | 41.7 | 64.8 | 60.0 | 41.7 | 39.0 | 57.4 |
| Qwen3-235B-A22B-Instr-2507 | 59.8 | 59.4 | 59.3 | 45.0 | 34.3 | 27.0 | 56.4 | 53.5 | 53.5 | 54.9 | 46.3 | 35.4 | 48.7 |
| OlympicCoder-32B* | 61.4 | 57.3 | 58.7 | 41.0 | 44.8 | 41.7 | 52.9 | 41.1 | 55.0 | 53.1 | 41.0 | 37.2 | 48.3 |
| Qwen3-8B* | 65.6 | 54.7 | 54.5 | 36.3 | 51.4 | 52.9 | 51.8 | 31.9 | 55.6 | 47.5 | 23.6 | 34.4 | 46.7 |
| Qwen3-30B-A3B-Instr-2507 | 52.5 | 48.4 | 51.5 | 31.6 | 33.6 | 30.8 | 49.2 | 43.2 | 46.4 | 47.6 | 33.8 | 28.1 | 41.4 |
| Qwen3-Coder-30B-A3B-Instr | 47.8 | 41.3 | 43.5 | 26.1 | 36.6 | 34.1 | 46.1 | 42.3 | 44.6 | 41.7 | 33.5 | 22.3 | 38.3 |
| Qwen2.5-Coder-32B-Instr | 40.5 | 36.8 | 41.6 | 33.6 | 5.7 | 38.7 | 39.1 | 36.4 | 38.0 | 38.0 | 33.0 | 30.3 | 34.1 |
| OlympicCoder-7B* | 45.0 | 43.0 | 40.7 | 30.0 | 32.8 | 31.9 | 36.3 | 18.6 | 34.5 | 28.5 | 28.5 | 19.8 | 32.7 |
| Qwen2.5-Coder-14B-Instr | 33.6 | 28.0 | 35.1 | 28.3 | 32.2 | 23.0 | 30.9 | 31.6 | 26.6 | 26.3 | 26.6 | 26.3 | 29.5 |
| OpenRsn-Nmt-32B* | 80.2 | 56.9 | 50.5 | 13.8 | 16.5 | 12.5 | 39.9 | 4.3 | 25.1 | 21.5 | 17.4 | 10.1 | 29.1 |
| Seed-Coder-8B-Instr | 28.3 | 27.8 | 28.8 | 22.7 | 30.1 | 29.0 | 29.9 | 25.6 | 28.5 | 27.0 | 27.7 | 23.8 | 27.4 |
| OpenCodeRsn-Nmt-1.1-32B* | 72.4 | 47.2 | 39.1 | 8.5 | 11.8 | 10.0 | 30.9 | 2.1 | 30.3 | 28.3 | 13.7 | 7.8 | 25.2 |
| Devstral-Small-2505 | 30.3 | 23.9 | 27.9 | 21.1 | 27.5 | 26.8 | 25.7 | 24.4 | 26.4 | 26.0 | 24.0 | 17.9 | 24.9 |
| DeepSeek-Coder-33B-Instr | 22.4 | 18.9 | 22.7 | 15.5 | 12.0 | 8.5 | 22.3 | 3.1 | 17.9 | 18.8 | 18.8 | 15.4 | 16.4 |

Table 14 reports Pass@1 scores at sampling temperature $t = 0.2$, for all 19 evaluated models on the complete Multi-LCB benchmark, which contains 1,055 tasks per programming language. Each

score reflects the percentage of problems solved correctly on the first attempt, with higher values indicating better performance. Models marked with an asterisk (*) are reasoning-enhanced variants.

## G    COMPUTATION TIME

Table 15 reports the average compilation and execution time required to evaluate one full Multi-LCB run (1,050 tasks per language) across 90 parallel CPUs. On average, each language requires about 8 min 50 s ($\approx$ 530 s) per model, with a total wall-clock time of roughly 106 hours when aggregated over all twelve languages.

Execution cost varies noticeably by language. Ruby shows the highest mean time at 17 min 37 s, followed by Go and Python, each exceeding 11 minutes on average. In contrast, Kotlin, PHP, and JavaScript complete evaluation in under 4 minutes. These differences primarily reflect compilation overheads and runtime performance of each language's toolchain, and they guide resource planning for future large-scale model evaluations.

Table 15: Evaluation times (compilation + execution on tests + matching) across programming languages. Runs were executed in parallel on 90 CPUs over 1050 tasks (v1–v6). Averages and standard deviations are computed across the measured models.

| Language | Avg. Time (mm:ss) | Std. Dev. (mm:ss) | Avg. Time (s) | Std. Dev. (s) |
|---|---|---|---|---|
| C# | 9:10 | 3:08 | 550.15 | 188.83 |
| C++ | 10:25 | 2:54 | 625.72 | 174.17 |
| Go | 12:36 | 2:36 | 756.13 | 156.71 |
| Java | 10:44 | 2:55 | 644.07 | 175.96 |
| JavaScript | 3:44 | 1:05 | 224.73 | 65.14 |
| Kotlin | 3:14 | 1:46 | 194.87 | 106.23 |
| PHP | 3:29 | 0:49 | 209.82 | 49.78 |
| Python | 11:38 | 2:56 | 698.71 | 176.07 |
| Ruby | 17:37 | 3:27 | 1057.42 | 207.45 |
| Rust | 7:41 | 3:50 | 461.04 | 230.24 |
| Scala | 7:25 | 1:16 | 445.24 | 76.41 |
| TypeScript | 8:17 | 0:55 | 497.59 | 55.37 |
| **Average** | 8:50 | 4:12 | 530.46 | 252.51 |
| **Total (sum)** | 106:05 | — | 6365.48 | — |

## H    LANGUAGES AND COMPILER VERSIONS

All experiments were conducted in a controlled environment using the following language runtimes and compiler versions to ensure consistency and reproducibility across all tasks in Multi-LCB:

- **C++**: gcc 14.3.0
- **Java**: OpenJDK 8.0.412
- **Python**: 3.12.11
- **Rust**: 1.88.0
- **Go**: 1.22.12
- **Ruby**: 3.3.6
- **JavaScript (Node.js)**: 20.19.4
- **TypeScript (Deno)**: 2.3.4
- **C# (Mono)**: 6.12.0.199
- **Compilers (general)**: 1.11.0
- **PHP**: 8.1.0
- **Kotlin**: 2.2.0
- **Scala**: 2.11.8
- **pip**: 25.2

These versions were used consistently for compilation, execution, and evaluation to guarantee reproducibility of all Multi-LCB results.

# I PLATFORM ANALYSIS

Figures 10, 11, and 12 show performance comparison between LeetCode and AtCoder platforms across different programming languages. Models demonstrate varying capabilities depending on the platform, with some excelling on LeetCode's interview-style problems while others perform better on AtCoder's competitive programming tasks.

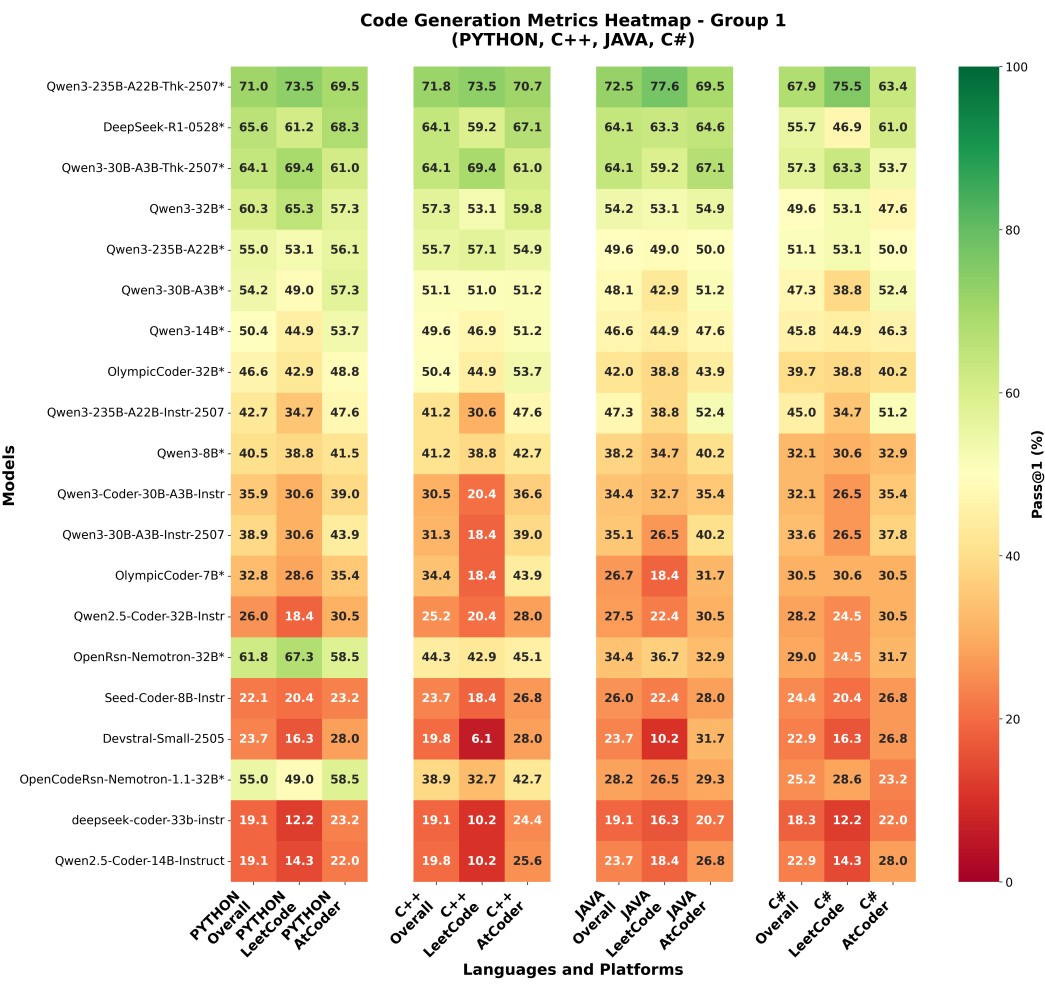

Figure 10: Code generation performance heatmap by platform for Python, C++, Java, and C#. Shows overall performance and platform-specific results (LeetCode vs AtCoder) across different models. Values represent Pass@1 scores (%).

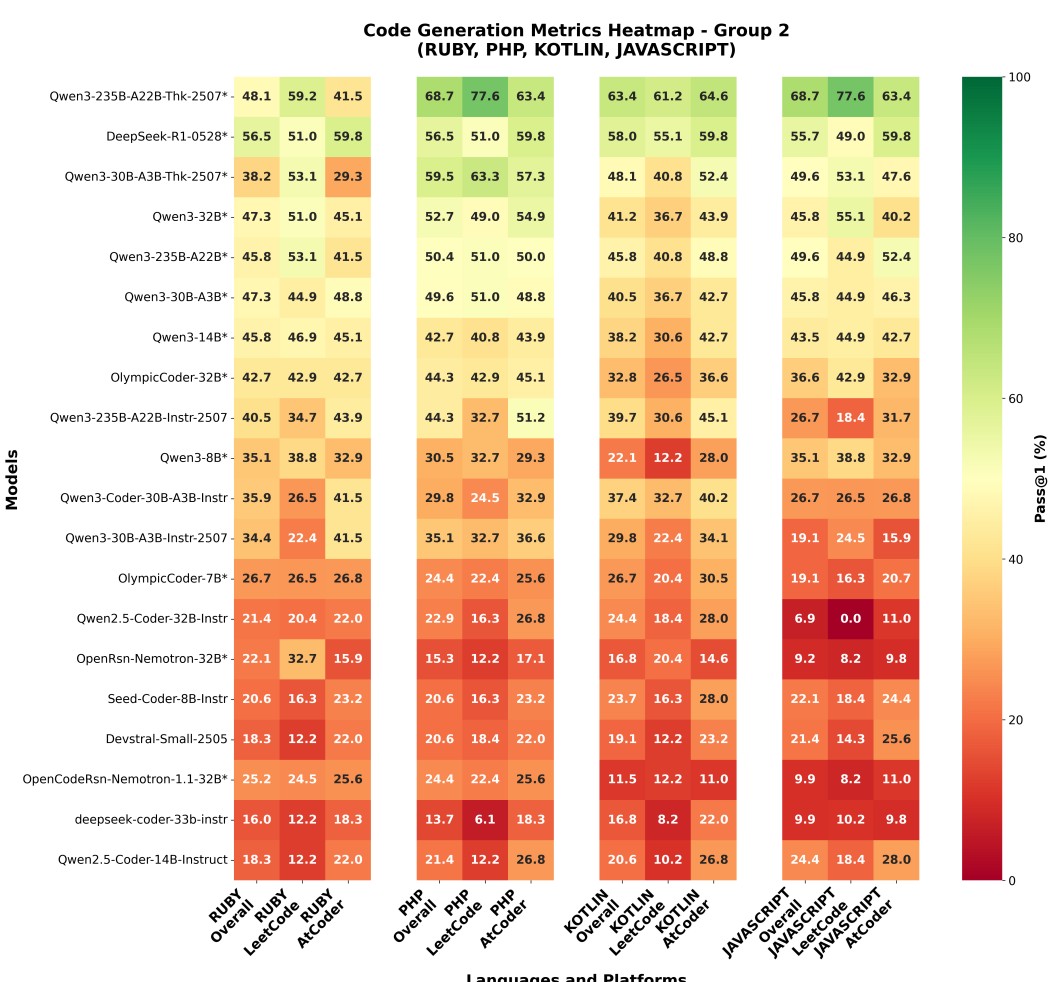

Figure 11: Code generation performance heatmap by platform for Ruby, PHP, Kotlin, and JavaScript. Shows overall performance and platform-specific results (LeetCode vs AtCoder) across different models. Values represent Pass@1 scores (%).

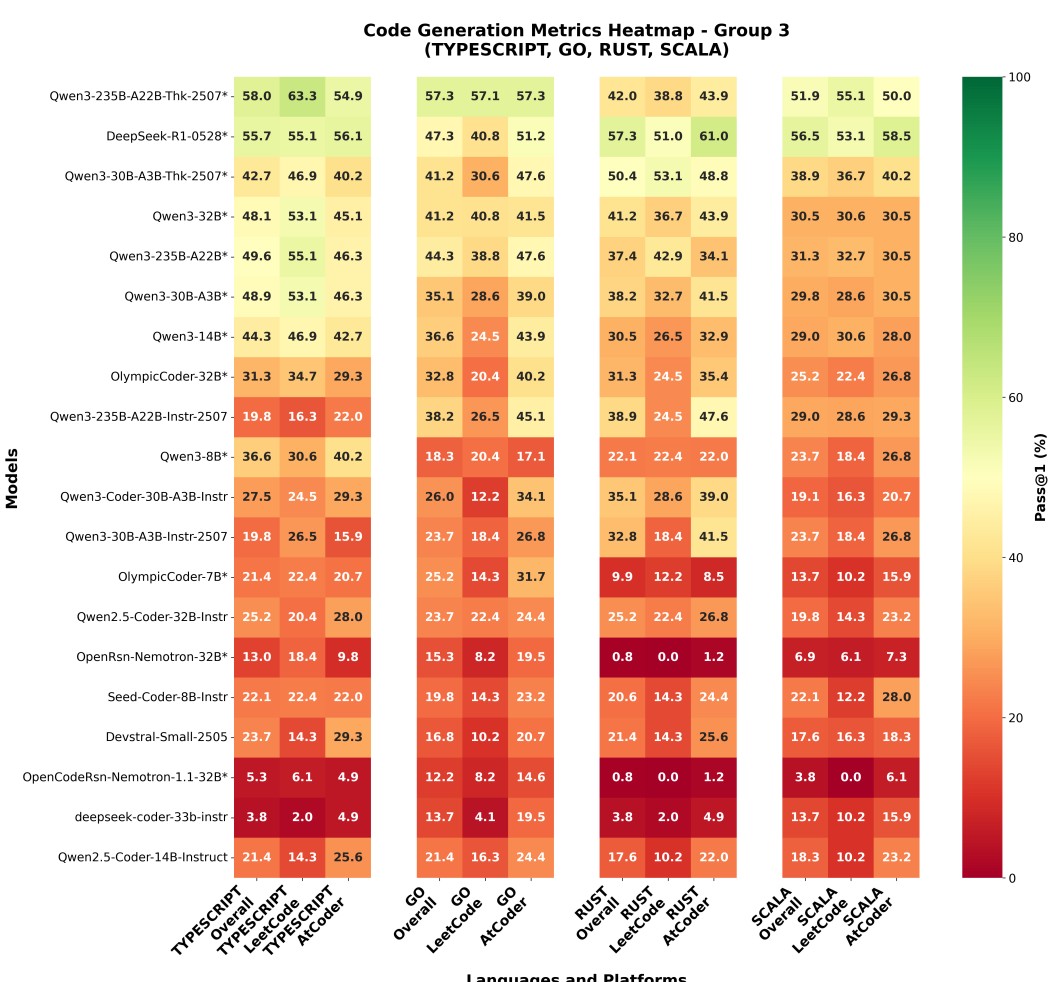

Figure 12: Code generation performance heatmap by platform for TypeScript, Go, Rust, and Scala. Shows overall performance and platform-specific results (LeetCode vs AtCoder) across different models. Values represent Pass@1 scores (%).

## J  DIFFICULTY ANALYSIS

Figures 13, 14, and 15 present performance breakdown by difficulty levels (Easy, Medium, Hard) across programming languages. The results reveal significant performance degradation as problem complexity increases, with Hard problems showing the largest performance gaps between models.

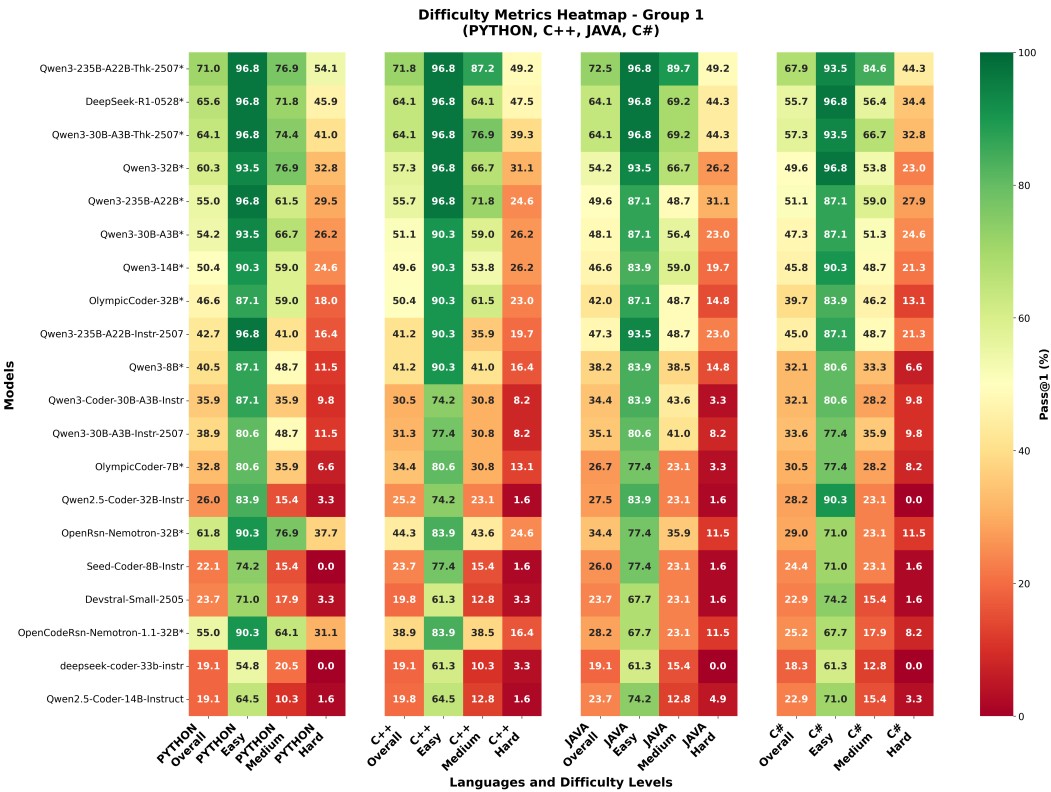

Figure 13: Code generation performance heatmap by difficulty level for Python, C++, Java, and C#. Shows overall performance and difficulty-specific results (Easy, Medium, Hard) across different models. Values represent Pass@1 scores (%).

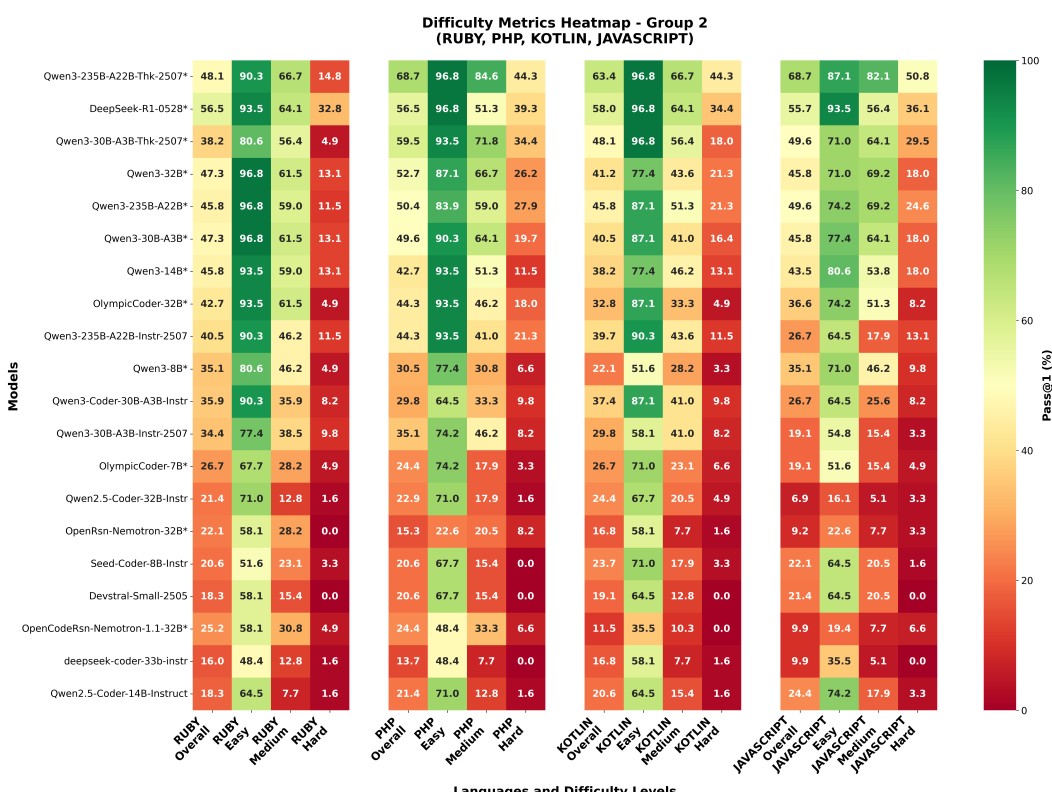

Figure 14: Code generation performance heatmap by difficulty level for Ruby, PHP, Kotlin, and JavaScript. Shows overall performance and difficulty-specific results (Easy, Medium, Hard) across different models. Values represent Pass@1 scores (%).

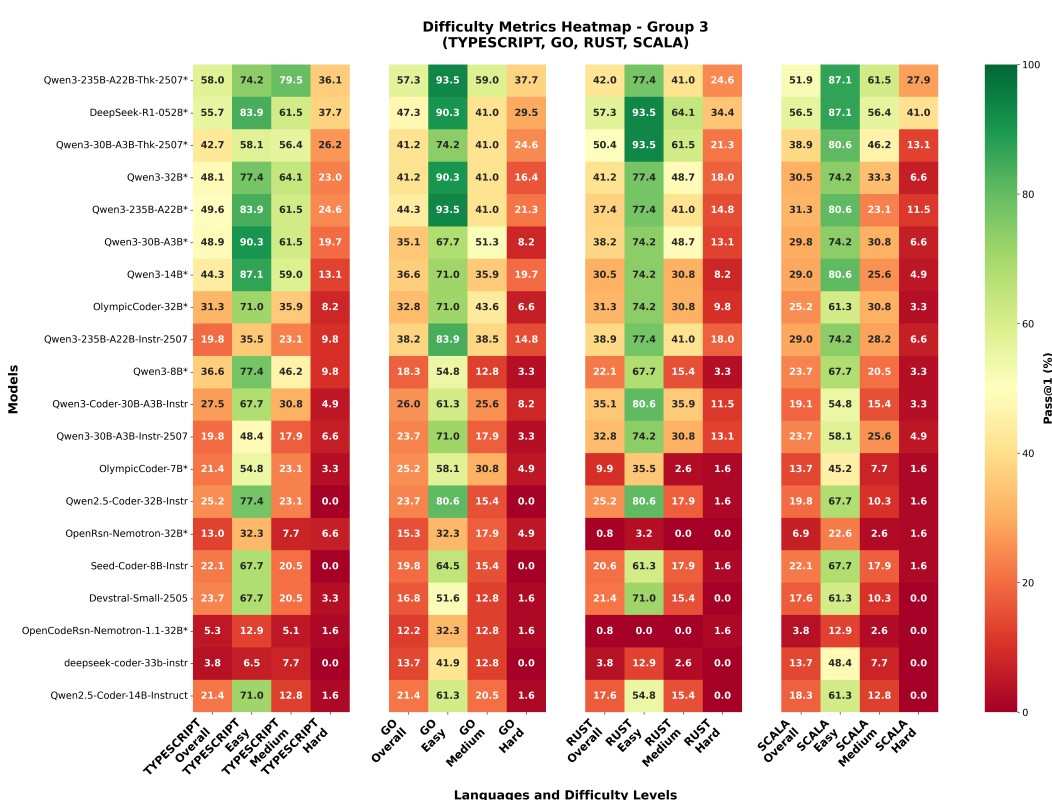

Figure 15: Code generation performance heatmap by difficulty level for TypeScript, Go, Rust, and Scala. Shows overall performance and difficulty-specific results (Easy, Medium, Hard) across different models. Values represent Pass@1 scores (%).

# K  TEMPORAL ANALYSIS

Figures 16, 17, 18, 19, 20, 21, 22, 23, 24, 25, 26, and 27 illustrate monthly performance trends from 2023 to 2025 across different programming languages. A notable declining trend is observed across all models and languages, with top-performing models dropping from approximately 80% to 60% Pass@1 scores over time. This consistent degradation pattern appears universally across programming languages, suggesting systematic factors rather than language-specific issues. The decline may be attributed to two primary factors: (1) data contamination effects, where models perform better on older, potentially seen problems, and (2) increasing problem complexity over time as benchmark creators develop more challenging tasks to maintain discriminative power.

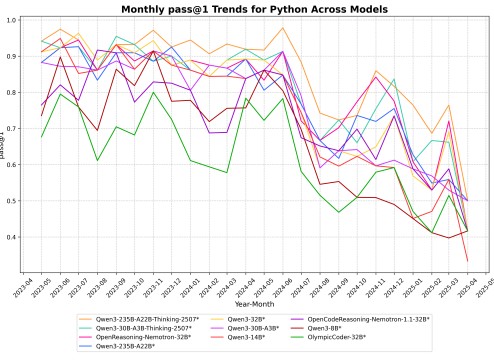

Figure 16: Monthly Pass@1 trends for Python.

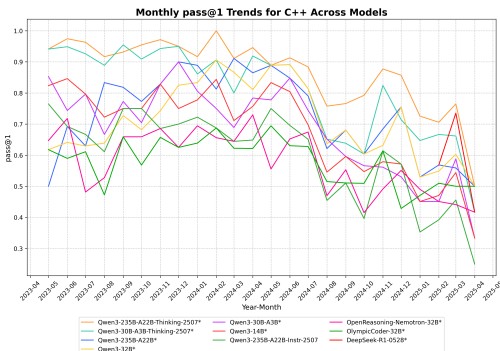

Figure 17: Monthly Pass@1 trends for C++.

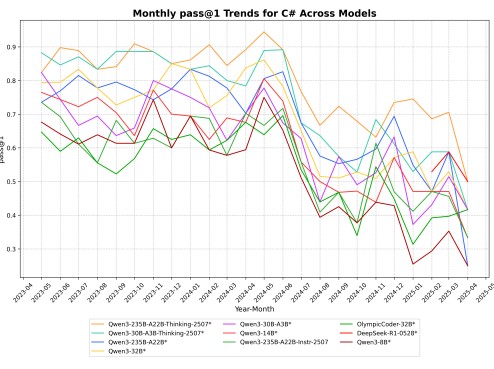

Figure 18: Monthly Pass@1 trends for C#.

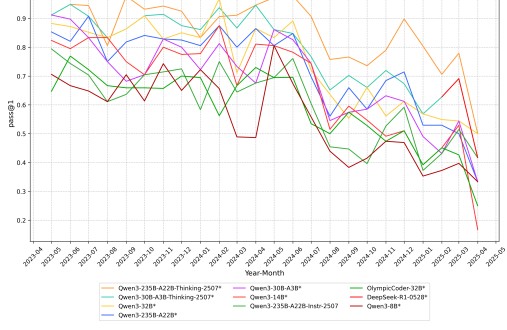

Figure 19: Monthly Pass@1 trends for Java.

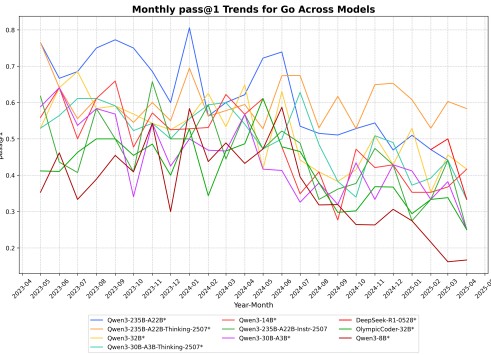

Figure 20: Monthly Pass@1 trends for Go.

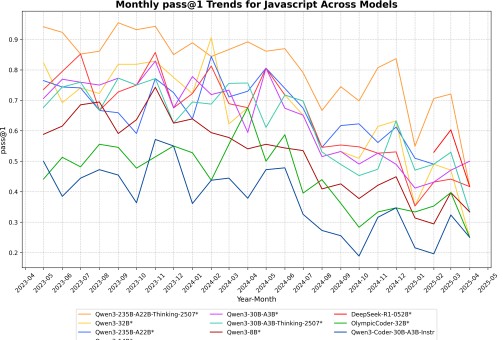

Figure 21: Monthly Pass@1 trends for JavaScript.

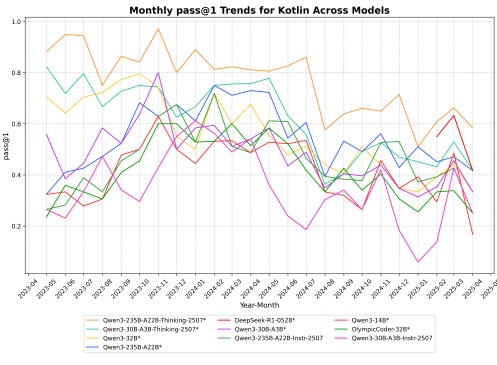

Figure 22: Monthly Pass@1 trends for Kotlin.

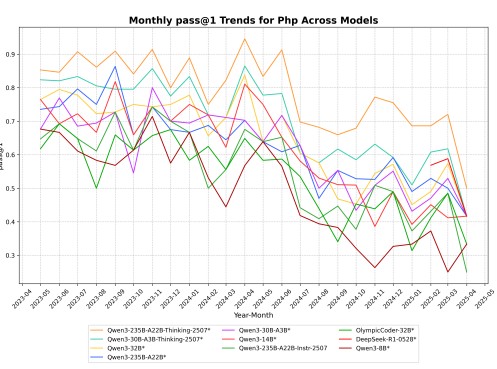

Figure 23: Monthly Pass@1 trends for PHP.

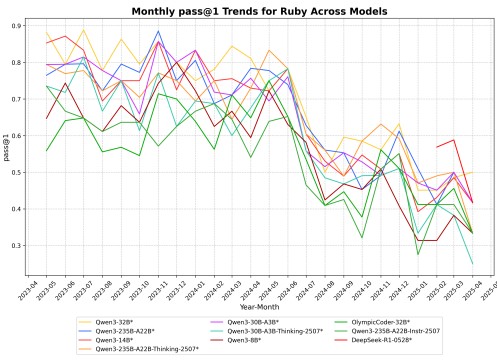

Figure 24: Monthly Pass@1 trends for Ruby.

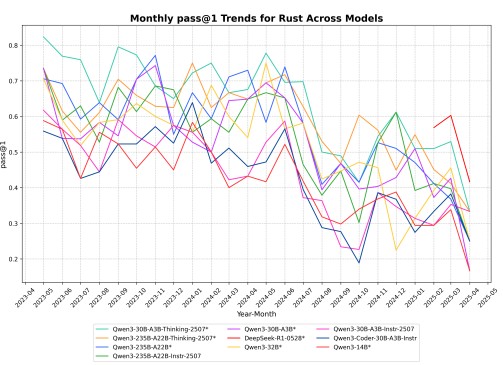

Figure 25: Monthly Pass@1 trends for Rust.

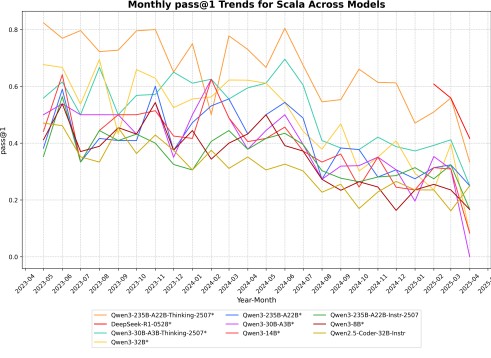

Figure 26: Monthly Pass@1 trends for Scala.

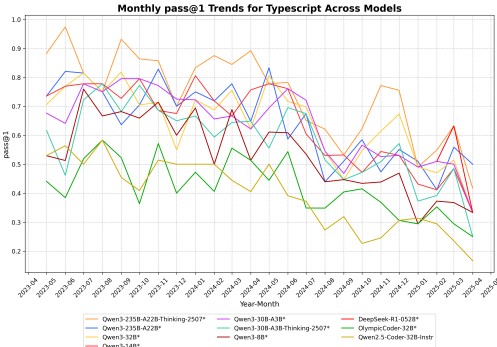

Figure 27: Monthly Pass@1 trends for Type-Script.

## L  LANGUAGES ERRORS TYPE

Figures 28, 29, 30, 31, 32, 33, 34, 35, 36, 37, 38, 39, 40, 41, 42 and 43 illustrate detailed error breakdowns across different programming languages and models. Several consistent patterns emerge:

1. Wrong-answer (WA) errors dominate across almost all languages and models. For every model, WA is the largest source of failure in both Python and non-Python languages, indicating that the primary bottleneck remains algorithmic correctness rather than compilation or parsing.

2. Compiled languages show substantially more compiler- and type-related errors. Languages such as C++, Java, Rust, and Go exhibit significantly higher rates of compilation errors (e.g., missing imports, type mismatches, incorrect signatures) compared to Python. This pattern is consistent across all models and reflects the challenge of generating syntactically valid and type-correct code when strict compilation pipelines are enforced.

3. Runtime exceptions increase in languages that require explicit input parsing. In languages like Java, C#, and Go, runtime errors (e.g., NullPointerException, IndexError, ValueError) are far more frequent than in Python. This supports the hypothesis that the STDIN/STDOUT format, while uniform across languages, exposes weaknesses in model robustness to input handling and data conversion.

4. Timeout and resource-related failures appear more often in slower languages and for reasoning-tuned models. Java, Rust, and Go show noticeably more TimeoutExpired cases, likely because models occasionally generate inefficient implementations. Reasoning-heavy models (e.g., R1-0528, Nemotron-32B) are more prone to long-running solutions when they attempt more complex multi-step logic.

5. Empty-code and trivial-syntax errors are rare but nonzero. These errors appear mostly in smaller models (e.g., 7B-14B) and are nearly absent for 30B+ models. This indicates that larger models rarely fail at the initial code-structuring stage, with most errors occurring deeper in the execution pipeline.

6. Cross-model consistency in error profiles. Despite architectural and training differences, the overall error distributions are remarkably stable across models, demonstrating that: Python remains the least error-prone language, Compiled languages introduce predictable error modes, and Languages with verbose input/output handling (Java, C#, Go) amplify runtime failures.

7. Error distributions reinforce the observed performance gaps. The breakdowns offer a mechanistic explanation for the Pass@1 disparities reported in the main results. For example, models underperforming in Rust and C++ do so not because they fail to produce solutions, but because syntactic and type-level correctness is significantly harder to achieve in those languages.

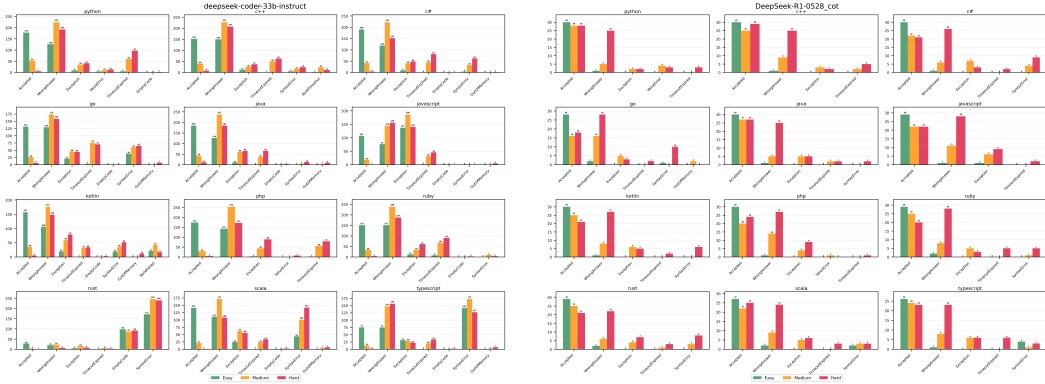

Figure 28: deepseek-coder-33b-instruct     Figure 29: DeepSeek-R1-0528*

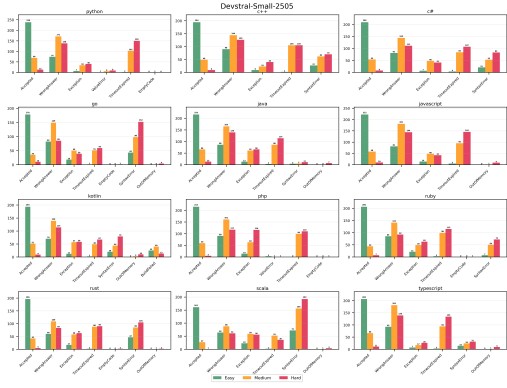

Figure 30: Devstral-Small-2505

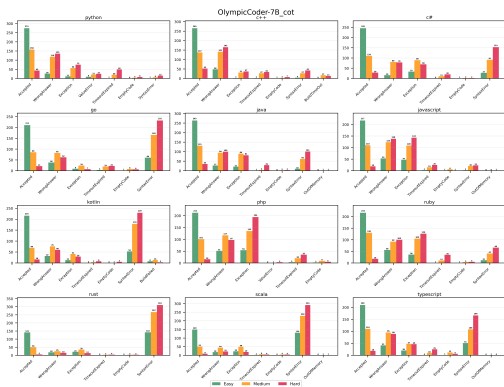

Figure 31: OlympicCoder-7B*

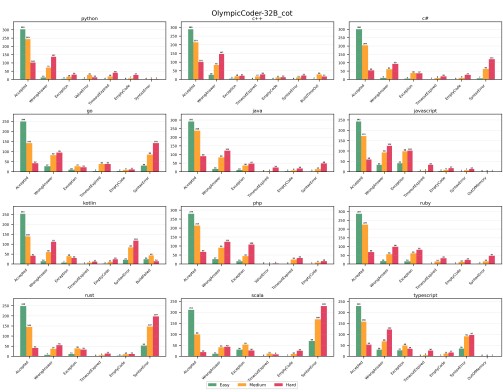

Figure 32: OlympicCoder-32B*

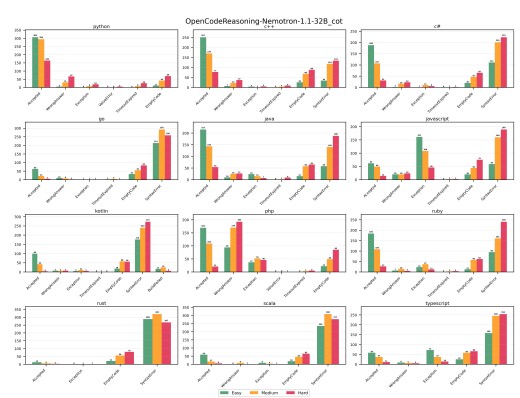

Figure 33: OpenCodeReasoning-Nemotron-1.1-32B*

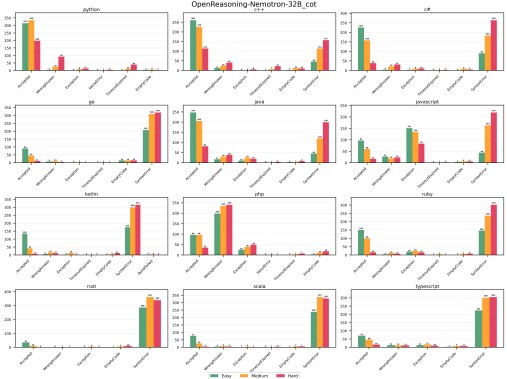

Figure 34: OpenReasoning-Nemotron-32B*

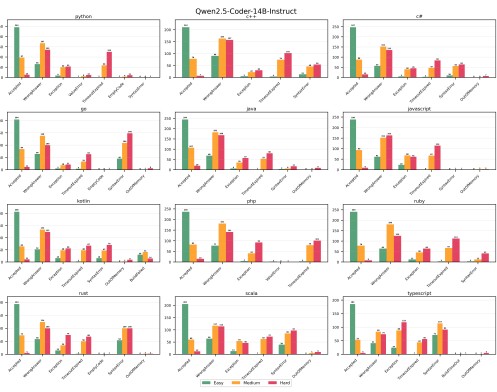

Figure 35: Qwen2.5-Coder-14B-Instruct

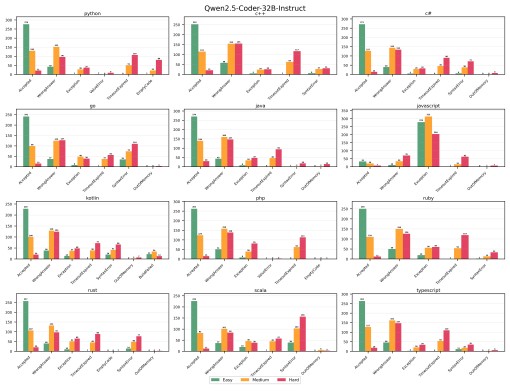

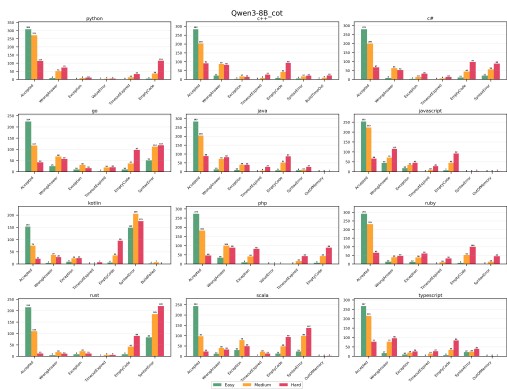

Figure 36: Qwen2.5-Coder-32B-Instruct

Figure 37: Qwen3-8B*

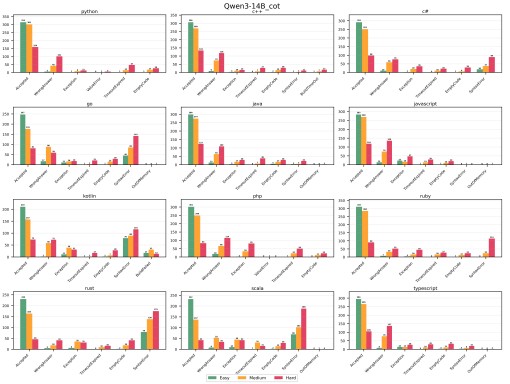

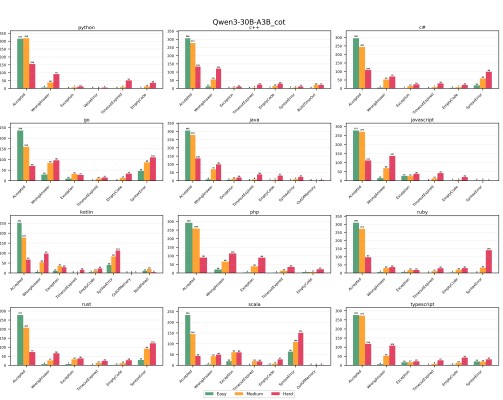

Figure 38: Qwen3-14B*

Figure 39: Qwen3-30B-A3B*

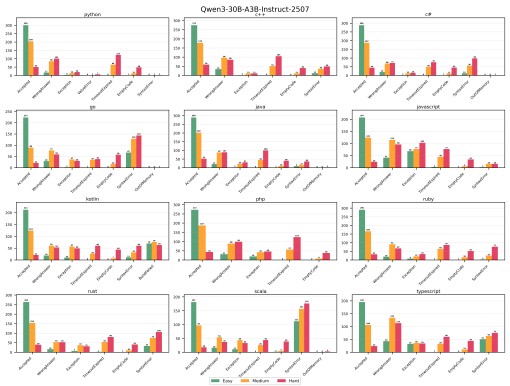

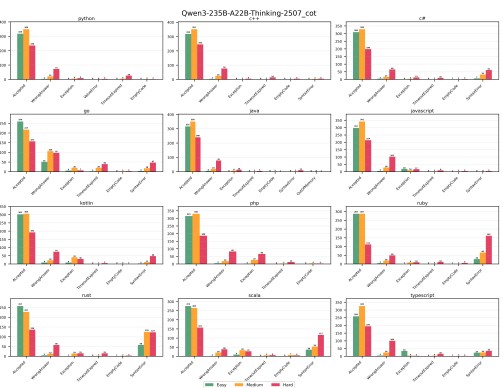

Figure 40: Qwen3-30B-A3B-Instruct-2507

Figure 41: Qwen3-235B-A22B-Thinking-2507*

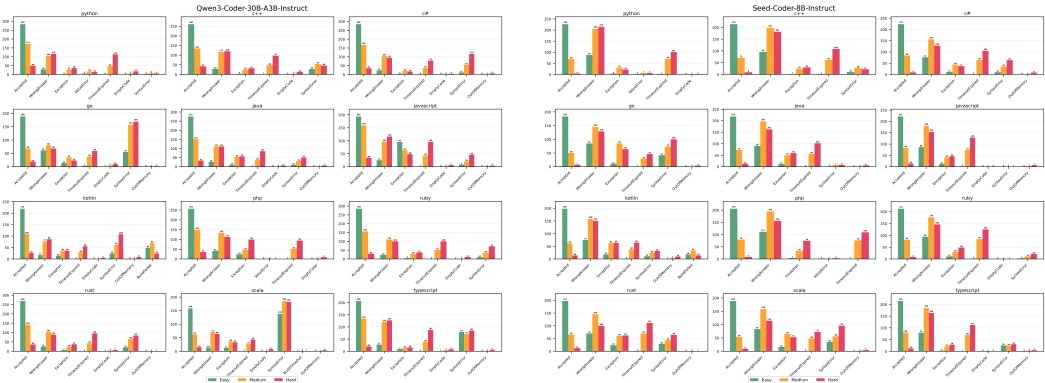

Figure 42: Qwen3-Coder-30B-A3B-Instruct

Figure 43: Seed-Coder-8B-Instruct

