# OpenReview forum: "Multi-LCB: Extending LiveCodeBench to Multiple Programming Languages"
_ICLR.cc/2026/Conference — ICLR 2026 Poster_

### Official Review · Reviewer_EauN · 2025-10-26

**Soundness:** 3
**Presentation:** 3
**Contribution:** 3
**Rating:** 4
**Confidence:** 3

**Summary:**

This paper extends LiveCodeBench (LCB) from Python to twelve programming languages and proposes Multi-LCB, a contamination-aware, continuously updating benchmark for multilingual code generation.
The authors convert LCB tasks into a unified STDIN/STDOUT evaluation pipeline and evaluate 20 recent LLMs.
Their findings include (i) Python overfitting; (ii) language-specific contamination signals via post-cutoff time slicing; and (iii) cross-language performance disparities.
The paper argues that Multi-LCB preserves LCB’s contamination controls while enabling side-by-side, same-task comparisons across languages.

**Strengths:**

1. **Meaningful benchmark**, which can be directly used to evaluate and compare LLMs' code generation capability across different PLs
2. **Large-scale experiments with insightful findings**. Cover diverse programming languages and evaluate 20 LLMs, providing findings to reveal real multilingual gaps in coding tasks.
3. **Additional results and Open-sourced code**. Provide additional results and implementation details in the appendix and also release the scripts and datasets in an anonymous repository.

**Weaknesses:**

1. **Limited Measurement of the Generalization on Different PL**. This paper aims to answer the question of whether LLMs can generalize across diverse programming languages (in abstract).
When evaluating a model's capability in a specific language, it's crucial to assess its capability in utilizing that language's unique features and built-in libraries  (e.g., Rust ownership/lifetimes, Go concurrency primitives, JS/TS async patterns, use of standard libraries). However, since all tasks were converted from Python tasks, one concern is that these tasks may be language-specific and thus fail to effectively measure the model's performance in using syntax and features related to other languages. Consequently, it cannot effectively measure the LLMs' capability in using different programming languages. This may limit the contribution of this benchmark.
2. **Insufficient handling of the differences between the syntax/features of different languages**. The outputs of certain tasks may depend on language-specific syntax/features (e.g., the measurement of Unicode length is different in Python, JS, and TS (code points VS code units), and modulo with negative inputs may have different results across languages). If such samples are directly translated from a Python dataset, the evaluation may unfairly penalize other languages and understate LLMs' competence on them.
3. **Contamination detection across languages**. Beyond filtering by the release date of the original Python problem, the paper does not provide other contamination detection methods. One of my concerns is that the rewritten/converted samples may have textual or semantical overlap with samples in the prior dataset, leading to data contamination and inflating scores of certain models.

**Questions:**

1.  This paper seems to only measure LLMs' capability to use different programming languages ​​to solve the same algorithm problem, which may not comprehensively reflect  LLMs' capability to use syntax and features unique to other languages, thereby providing a one-sided measurement.
Please discuss and clarify the contribution of this paper and what specific research questions this paper wants to answer.
2. How does the author filter or process these samples related to the unique features of the programming language? What are their effects on the assessment of models' capability on different PLs in your experiments?
3. Besides the release-date filtering, what contamination detection methods are used on the converted samples in different languages?

---

> ### Author Response · Authors · 2025-11-15
> **Response to Reviewer EauN**
>
> We thank Reviewer EauN for the detailed and constructive feedback, and we appreciate the opportunity to clarify several points. We address all comments below.
>
> > W1. “However, since all tasks were converted from Python tasks, one concern is that these tasks may be language-specific and thus fail to effectively measure the model's performance in using syntax and features related to other languages.”
>
> >W2. “The outputs of certain tasks may depend on language-specific syntax/features (e.g., the measurement of Unicode length is different in Python, JS, and TS (code points VS code units), and modulo with negative inputs may have different results across languages). If such samples are directly translated from a Python dataset, the evaluation may unfairly penalize other languages and understate LLMs' competence on them.”
>
> >Q1. This paper seems to only measure LLMs' capability to use different programming languages ​​to solve the same algorithm problem, which may not comprehensively reflect LLMs' capability to use syntax and features unique to other languages, thereby providing a one-sided measurement. Please discuss and clarify the contribution of this paper and what specific research questions this paper wants to answer.
>
> >Q2. How does the author filter or process these samples related to the unique features of the programming language?
>
> Prior multi programming language benchmarks translated functional-format Python problems (e.g., HumanEval, MBPP) by rewriting function signatures and regenerating unit tests for each language. Even a simple Python assertion like:
>
> `assert binomial_coeff(5, 2) == 10`
>
> must be expanded into multi-line Java test code. This translation must be repeated separately for every language and is sensitive to syntax and runtime differences.
>
> Multi-LCB takes a different approach by keeping only the natural-language description and converting hidden tests into a language-agnostic STDIN/STDOUT format, for example:
>
> `Input:`
>
> `5 2`
>
> `Output:`
>
> `10`
>
> This provides several methodological advantages:
>
> 1. one I/O conversion works for all languages (vs. N separate functional-format translations).
>
> 2. all languages get identical I/O pairs, which functional-format  benchmarks cannot guarantee.
>
> 3. Multi-LCB automatically updates in sync with new LCB releases. No manual intervention is required: new LCB problems instantly appear in all languages through the Multi-LCB pipeline.
>
> > Q2 What are their effects on the assessment of models' capability on different PLs in your experiments?
>
> Multi-LCB reveals important and previously unobserved findings:
>
> 1. Persistent cross-language accuracy gaps (see section 5.1)
>
> 2. Evidence of Python overfitting and evaluation contamination, visible through temporal performance drops (see section 5.2)
>
> 3. Consistent error patterns across languages (see Appendix L for detailed error breakdown), showing that strict typing and input-parsing requirements systematically challenge models. Our results (e.g., consistent error distributions across languages and models) confirm that the STDIN/STDOUT format provides uniform task difficulty, without introducing language-dependent biases.
>
> 4. Diverging model rankings across languages, demonstrating that Python-only benchmarks are not reliable proxies for general programming ability.
>
> >W3. Contamination detection across languages. Beyond filtering by the release date of the original Python problem, the paper does not provide other contamination detection methods. One of my concerns is that the rewritten/converted samples may have textual or semantical overlap with samples in the prior dataset, leading to data contamination and inflating scores of certain models.
>
> >Q3. Besides the release-date filtering, what contamination detection methods are used on the converted samples in different languages?
>
> Thank you for pointing this out, we agree that contamination can never be entirely ruled out for large pretrained models. However, most contamination issues have already been carefully addressed in LCB through date-based task filtering and reproducibility protocols, and Multi-LCB fully inherits these mechanisms for cross-language evaluation.

---

> > ### Comment · Reviewer_EauN · 2025-11-16
> >
> > Thanks for the authors' prompt reply. However, the authors do not directly answer most of my questions.
> >
> > 1. The response highlights the benchmark design: translating the original Python problem into problems for different PLs by rewriting signatures and using language-agnostic STDIN/STDOUT format. This has already been clearly stated in the paper. However, W1 and Q1 are requesting the authors to clarify the contribution of their benchmark (e.g., the position of this paper, what this benchmark can measure), not the `methodological advantages`. The concern is that different PLs have different features and design philosophies. Since all cases in this benchmark are translated from a Python benchmark, it may not effectively cover the unique features of each PL, thus failing to comprehensively evaluate the LLMs' capability to use different PLs. (see Weakness 1)
> > 2. W2 and Q2 mention functional differences caused by the features of different PLs, which are not `previously unobserved findings`. For example, for the same Unicode emoji, Python, JS, and TS could have different measurements of its length. This is because JS and TS implement strings internally as a sequence of UTF-16 code units (e.g., the result of `"👍".length` is 2), whereas Python uses Unicode code points (e.g., `len("👍")` results in 1). Since the benchmark is translated from Python, the functional difference in features could cause false positives in the experiments, thereby underestimating the correctness of other PLs. Q2 requests the authors to elaborate on how they handle or filter such samples related to language-specific features. If they do not handle them in the benchmark, the authors should report the proportion of these samples in the benchmark and discuss their potential impact on the results. This is directly related to the validity and reliability of the benchmark's measurement results.

---

> > > ### Author Response · Authors · 2025-11-16
> > > **Response to Official Comment by Reviewer EauN**
> > >
> > > We thank the reviewer for the prompt response, we really appreciate the opportunity to clarify all raised concerns.
> > >
> > > >*translating the original Python problem into problems for different PLs by rewriting signatures*
> > >
> > > >*Since all cases in this benchmark are translated from a Python benchmark*
> > >
> > > We would like to clarify that Multi-LCB does not rewrite Python signatures and does not translate Python code into other languages. The benchmark uses only:
> > >
> > > (i) the natural-language problem description (no Python semantics, no Python function signatures), and
> > >
> > > (ii) the hidden I/O tests from LCB (again, no Python semantics, no Python function signatures, and no Python unit-test scaffolding).
> > >
> > > Regarding the broader question of what this benchmark measures:
> > >
> > > >*The concern is that different PLs have different features and design philosophies*
> > >
> > > Our benchmark, like LCB for Python, evaluates the ability to solve algorithmic problems, with the same level of comprehensiveness as competitive programming contests. The ability to solve algorithmic tasks is essential in any programming language. The problem statements are written in natural language with no constraints, so any features, advantages, or design philosophy of the particular language can be fully utilized when solving a task.
> > >
> > > >*comprehensively evaluate the LLMs' capability to use different PLs*
> > >
> > > A comprehensive evaluation of an LLM’s ability to solve not only algorithmic problems but arbitrary tasks in a given programming language is extremely difficult. None of the existing code-generation benchmarks attempt this, and it is unlikely to be achievable in the foreseeable future. Such full-spectrum evaluation is outside the scope of our work.
> > >
> > > > W2 and Q2 mention functional differences caused by the features of different PLs, which are not previously unobserved findings.
> > >
> > > >  Q2 requests the authors to elaborate on how they handle or filter such samples related to language-specific features.
> > >
> > > This issue is not relevant to our benchmark. These are algorithmic problems from programming contests and LeetCode. The task formulations, as presented in the prompt, are constructed to have exactly one intended interpretation (without ambiguities that depend on language-specific features).
> > >
> > > > For example, for the same Unicode emoji, Python, JS, and TS could have different measurements of its length. This is because JS and TS implement strings internally as a sequence of UTF-16 code units (e.g., the result of `"👍".length` is 2), whereas Python uses Unicode code points (e.g., `len("👍")` results in 1).
> > >
> > > Regarding the string-length case mentioned above, the task would be formulated as:
> > >
> > > `“count the number of characters (not bytes) in the input string.” `
> > >
> > > For Python, the correct solution would be `len(input)`.
> > >
> > > For JavaScript, the correct solution would be `[...input].length`, whereas `input.length` would be an incorrect solution.
> > >
> > > > W1 and Q1 are requesting the authors to clarify the contribution of their benchmark
> > >
> > > Our benchmark is the only one that enables analyzing differences in model performance across programming languages on a large volume of contamination-free tasks, and it uses completely identical tasks for all languages, which minimizes the factor of different task difficulty.

---

> > > > ### Comment · Reviewer_EauN · 2025-11-16
> > > >
> > > > Thank you to the authors for their detailed response. This response has effectively addressed some of my questions.
> > > > I admit my use of 'rewriting signature' is ambiguous. I am actually referring to the rewriting/reformatting of the problem statement in Lines 176-179.
> > > >
> > > > Regarding the answer to Q2W2, I still have questions.
> > > > The authors claim to use I/O tests from LCB and use natural language to describe the problems.
> > > > However, natural language can be ambiguous.
> > > > I am concerned about encountering scenarios like the following:
> > > >
> > > > The natural language problem description is: `count the length of the input string`. Since the original benchmark is Python, it defaults to counting the UTF-8 length, so for input `👍`, the expected output is 1.
> > > > However, in JS and TS (where strings are internally represented using UTF-16 encoding), when the same question is asked and is not explicitly requested to use code points, the result might be 2.
> > > > In this case, a result of 2 does not mean the model produces an incorrect result, but rather it may stem from the PL features.
> > > >
> > > > I would like to ask if similar situations (related to the unique PL features) exist in the MultiLCB benchmark?
> > > > Additionally, the authors mentioned `the task would be formulated as: `count the number of characters (not bytes) in the input string.`'
> > > > Does this imply that the automatic conversion pipeline in this paper already has checks related to PL features and automatically formulates and rewrites the problem statement/description for such cases?

---

> > > > > ### Author Response · Authors · 2025-11-16
> > > > > **Response to Official Comment by Reviewer EauN**
> > > > >
> > > > > We thank the reviewer for the clarification.
> > > > >
> > > > > Multi-LCB (as well as LiveCodeBench) uses problems collected from programming contest platforms such as LeetCode, AtCoder, and Codeforces. These problem statements are written by human experts, and the platforms accept solutions in many programming languages (e.g., LeetCode supports 18 languages); therefore, it is expected that problem authors intentionally design their problem definitions in a way that avoids language-specific ambiguities (including cases like Unicode length or encoding-dependent behavior).
> > > > >
> > > > > As a result, Multi-LCB does not introduce any additional rewriting or PL-specific adjustments, the original problems are inherently language-agnostic. During pipeline testing, we also manually inspected around 500 tasks from the LCB releases used in Multi-LCB and did not encounter any cases of ambiguities arising from language-specific features.

---

> > > > > > ### Comment · Reviewer_EauN · 2025-11-20
> > > > > >
> > > > > > Thanks for the authors' response.
> > > > > >
> > > > > > The authors have resolved most of my concerns through manual inspection and explanations. I will raise the score.
> > > > > > In addition, I strongly recommend that the authors include the manual inspection results, corresponding explanations, and cases like `binomial_coeff` in the revised paper (even if only in the appendix) to further enhance the presentation and avoid reader misunderstandings.

---

> > > > > > > ### Author Response · Authors · 2025-11-20
> > > > > > > **Response to Official Comment by Reviewer EauN**
> > > > > > >
> > > > > > > Thank you very much for your thoughtful follow-up and for raising the score! We fully agree with your recommendation to include the manual inspection results, corresponding explanations, and illustrative examples, and we will add these details. Specifically, we will include the `binomial_coeff` example in the main text (Section 3. Benchmark Design). Your support is greatly appreciated.

---

> > > > > > > > ### Author Response · Authors · 2025-11-27
> > > > > > > > **Follow-Up on Our Previous Response to Official Comment by Reviewer EauN**
> > > > > > > >
> > > > > > > > A quick note to confirm that we have incorporated all of your suggestions into the revised manuscript.
> > > > > > > >
> > > > > > > > > I strongly recommend that the authors include the manual inspection results, corresponding explanations, and cases like `binomial_coeff` in the revised paper (even if only in the appendix).
> > > > > > > >
> > > > > > > > We added the manual inspection results and corresponding explanations to Section 3 (Benchmark Design), lines 183-190.
> > > > > > > >
> > > > > > > > We also added the `binomial_coeff` example to the Section 2 (Related Work), as an example of how MBXP and similar works rewrite Python tasks (lines 100-106) and a differentiation from Multi-LCB’s language-agnostic STDIN/STDOUT conversion (lines 129-135).

---

### Official Review · Reviewer_52p9 · 2025-10-27

**Soundness:** 3
**Presentation:** 3
**Contribution:** 2
**Rating:** 6
**Confidence:** 4

**Summary:**

This project extends the LiveCodeBench (LCB) benchmark to support multiple programming languages. Specifically, the authors translated the original Python-based problems into 11 additional languages, resulting in a total of 12 supported languages. They also developed an automated mechanism to track and incorporate future LCB updates, ensuring long-term maintainability of the benchmark. Using the new Multi-LCB dataset, the authors conducted a comprehensive evaluation of 20 large language models (LLMs), revealing that most models exhibit substantially lower performance on languages other than Python due to overfitting, and uneven distribution of pertaining corpora in terms of programming language diversity. The authors publicly release their benchmark extension, including prompts, source code, and configurations under MIT license to facility reproduction and future research.

**Strengths:**

* The work extends a state-of-the-art (SOTA) benchmark while preserving the core strengths of the original LiveCodeBench. The design also emphasizes long-term maintainability through automated update tracking.
* The evaluation framework is relevant for practitioners, addressing key technical challenges such as automatically converting code problems and hidden test cases into the appropriate format, as well as supporting sandboxed execution.
* The paper provides valuable insights into benchmark contamination by using release date metadata to perform experiments, highlighting an important issue in LLM evaluation.
* The evaluation includes a sanity check by comparing Python results on both the original LCB and the proposed Multi-LCB, demonstrating methodological rigor.

**Weaknesses:**

* The work does not introduce new task types. Since certain programming languages have inherent advantages for specific problems, adding novel or language-agnostic tasks could have strengthened the contribution’s originality.
* While the overall presentation is solid, there are opportunities to enhance transparency in specific methodological aspects (see Questions section).
* It is unclear whether the proposed benchmark covers all existing LCB releases or only the latest one.
* The benchmark appears less challenging for LLMs on widely used programming languages — strong performance is reported for C++, Java, PHP, C#, and JavaScript.
* The evaluation omits GPT-OSS and leading proprietary models, limiting the completeness of the comparative analysis.

**Questions:**

1. My understanding is that LCB releases are non-overlapping and cumulative, each representing a temporal evaluation slice. Did you convert tasks from all releases or only from the latest one? The paper notes that your pipeline supports the conversion of multiple versions, but it is not explicit whether all versions were actually converted.
2. Lines 205–215 mention adopting a zero-shot strategy, yet Step 2 refers to the inclusion of illustrative examples. Could you clarify how these examples fit within a zero-shot setup?
3. Please elaborate on the infrastructure or practical constraints that prevented inclusion of the Swift programming language.
4. I recommend adding evaluations for GPT-OSS as well as leading closed-source models to strengthen the study’s benchmarking scope.
5. In your view, which factor contributes more to improving cross-language performance — multi-language training or enhanced reasoning capabilities?
6. Please explain how the evaluation framework supports long-term maintainability through automated update tracking.

---

> ### Author Response · Authors · 2025-11-15
> **Response to Reviewer 52p9. Part 1**
>
> We thank Reviewer 52p9 for the thoughtful and positive evaluation, and we appreciate highlighting the strengths of our work. We address all raised concerns below.
>
> >W1. The work does not introduce new task types. Since certain programming languages have inherent advantages for specific problems, adding novel or language-agnostic tasks could have strengthened the contribution’s originality.
>
> We respectfully disagree. Multi-LCB *does* introduce language-agnostic task types by transforming LCB problems into a universal STDIN/STDOUT format.
>
> Prior multi programming language benchmarks translated functional-format Python problems (e.g., HumanEval, MBPP) by rewriting function signatures and regenerating unit tests for each language. Even a simple Python assertion like:
>
> `assert binomial_coeff(5, 2) == 10`
>
> must be expanded into multi-line Java test code. This translation must be repeated separately for every language and is sensitive to syntax and runtime differences.
>
> Multi-LCB takes a different approach by keeping only the natural-language description and converting hidden tests into a language-agnostic STDIN/STDOUT format, for example:
>
> `Input:`
>
> `5 2`
>
> `Output:`
>
> `10`
>
> This provides several methodological advantages:
>
> 1. one I/O conversion works for all languages (vs. N separate functional-format translations).
>
> 2. all languages get identical I/O pairs, which functional-format  benchmarks cannot guarantee.
>
> 3. Multi-LCB automatically updates in sync with new LCB releases. No manual intervention is required: new LCB problems instantly appear in all languages through the Multi-LCB pipeline.
>
> >W3. It is unclear whether the proposed benchmark covers all existing LCB releases or only the latest one.
>
> > Q1 My understanding is that LCB releases are non-overlapping and cumulative, each representing a temporal evaluation slice. Did you convert tasks from all releases or only from the latest one? The paper notes that your pipeline supports the conversion of multiple versions, but it is not explicit whether all versions were actually converted.
>
> Our benchmark covers *all existing or future LCB releases*, not just the latest one. We convert every release, and the appendix reports results across multiple temporal slices. Appendix F includes Pass@1 for February-May 2025, July 2024-May 2025, and the full 1,055-task Multi-LCB benchmark. Appendix K provides monthly performance trends from 2023 to 2025 (Figures 16-27).
>
> > W4. The benchmark appears less challenging for LLMs on widely used programming languages — strong performance is reported for C++, Java, PHP, C#, and JavaScript.
>
> We would appreciate clarification on this point. Although some widely used languages show higher Pass@1, Appendix L (Figures 28-43) provides detailed error breakdowns for all models and languages, including Wrong Answer, Compilation Error, Runtime Exception, Timeout, and Other, allowing us to analyze the sources of difficulty beyond Pass@1.
>
> >W5. The evaluation omits GPT-OSS and leading proprietary models, limiting the completeness of the comparative analysis.
>
> >Q4. I recommend adding evaluations for GPT-OSS as well as leading closed-source models to strengthen the study’s benchmarking scope.
>
> While we agree that including leading proprietary models would provide additional context, running our full Multi-LCB evaluation on commercial APIs is prohibitively expensive.
>
> Our setup requires evaluating 131 tasks × 12 programming languages × 10 samples per task, with roughly 32k tokens per sample, totaling ~503 million output tokens per model. Using publicly available pricing, this corresponds to approximately 37k for Claude Opus, 7.5k for Claude Sonnet, 5k for GPT-5 or Gemini Pro, 2.5k for Claude Haiku, and around 1k for GPT-5-mini or Gemini Flash. Evaluating all major commercial models would therefore exceed $50k, which is not feasible in an academic setting. Proprietary model providers explicitly report LCB results in their technical reports, meaning that high-quality performance numbers for commercial models generally become publicly available.
>
> For these reasons, we focus on open, reproducible models and leave large-scale proprietary-model benchmarking to future work. However, to improve completeness, we will additionally include experimental results for GPT-OSS in the revised version.

---

> > ### Author Response · Authors · 2025-11-15
> > **Response to Reviewer 52p9. Part 2**
> >
> > > Q2. Lines 205–215 mention adopting a zero-shot strategy, yet Step 2 refers to the inclusion of illustrative examples. Could you clarify how these examples fit within a zero-shot setup?
> >
> > Thank you for pointing this out, we clarified the wording: the illustrative examples we include are the input/output samples already provided in the original problem descriptions, not examples of solutions or solution strategies. This is standard practice in code benchmarks and fully compatible with zero-shot evaluation. The model sees only the task statement and these sample I/O pairs and never any example solutions, so the zero-shot setup remains intact.
> >
> > > Q3. Please elaborate on the infrastructure or practical constraints that prevented inclusion of the Swift programming language.
> >
> > Thank you for the suggestion, Swift support has now been added to the Multi-LCB pipeline, and experiments are currently ongoing. We will include full Swift results in the revised version of the paper.
> >
> > > Q5. In your view, which factor contributes more to improving cross-language performance — multi-language training or enhanced reasoning capabilities?
> >
> > This is an important but inherently difficult question to disentangle experimentally. Based on our observations, we hypothesize that multi-language training plays the dominant role in improving cross-language performance. Languages with limited representation in pre-training corpora (e.g., Ruby, Scala, Rust) consistently lag behind Python even for models with strong general reasoning abilities. At the same time, scaling up reasoning-centric fine-tuning (e.g., CoT-tuned variants) tends to improve success rates within a language but does not eliminate cross-language gaps.
> >
> > Our results therefore suggest that reasoning helps, but it does not compensate for insufficient multilingual code exposure, and that robust cross-language generalization is most strongly driven by diverse, balanced multilingual training data. Further controlled experiments would be valuable, but this trend is consistent across the models evaluated in Multi-LCB.
> >
> > > Q6. Please explain how the evaluation framework supports long-term maintainability through automated update tracking.
> >
> > Multi-LCB is built for long-term maintainability. Our update-tracking scripts automatically detect new LCB releases, convert the newly added problems into the STDIN/STDOUT format, and recompute all evaluation metrics with no manual intervention. This keeps Multi-LCB fully synchronized with LCB as it evolves and ensures consistent, up-to-date multilingual benchmarking.

---

> > > ### Comment · Reviewer_52p9 · 2025-11-22
> > >
> > > > Q2
> > >
> > > Please make sure wording clarification is added to the paper.
> > >
> > > > Q3
> > >
> > > Great!
> > >
> > > > Q5
> > >
> > > I am not expecting additional experiments here; I was primarily seeking your viewpoint. Thank you for sharing your insights.
> > >
> > > > Q6
> > >
> > > Thanks for the clarification.

---

> > > > ### Author Response · Authors · 2025-11-22
> > > > **Response to Official Comment by Reviewer 52p9**
> > > >
> > > > > Thank you for the clarification. To avoid misunderstanding, my original point was not that your framework fails to introduce a new format, but rather that the underlying problems themselves are not newly designed. Based on your response to Q6, I understand that your intention is to position Multi-LCB as an extension of the original LCB—providing a unified, language-agnostic interface—rather than proposing an independent benchmark with new-independent tasks. If you are got going to add new tasks, I suggest to make sure the paper frames it that way.
> > > >
> > > > Thank you for the clarification. We would like to emphasize that our approach is fully open and can be directly applied to other benchmarks, for example, to LCB Pro (Zheng et al., 2025), which contains more challenging problems. From our side, we are not able to include LCB Pro at this moment because several tasks remain non-public, but LCB Pro can easily incorporate our multi-programming-language STDIN/STDOUT pipeline.
> > > >
> > > > > I'm not asking to evaluate on proprietary models. GPT-OSS-120B and GPT-OSS-20B are open source models, you can load them locally with vLLM. At least try to include results for GPT-OSS-20B.
> > > >
> > > > Thank you for the clarification. We have now added both GPT-OSS-120B and GPT-OSS-20B to our evaluation in the Low and Medium thinking modes, for sampling temperatures 0.2, 0.6, and 1.0 (Table 1, Table 5, and Table 6).

---

> > > > > ### Author Response · Authors · 2025-11-27
> > > > > **Follow-Up on Our Previous Response to Official Comment by Reviewer 52p9**
> > > > >
> > > > > A quick note to confirm that we have incorporated all of your suggestions into the revised manuscript.
> > > > >
> > > > > >> Q3. Please elaborate on the infrastructure or practical constraints that prevented inclusion of the Swift programming language.
> > > > >
> > > > > > Thank you for the suggestion, Swift support has now been added to the Multi-LCB pipeline, and experiments are currently ongoing. We will include full Swift results in the revised version of the paper.
> > > > >
> > > > > The experiments are still ongoing. Below, we provide preliminary Swift, R, and Lua Pass@1 (\%) averaged over 10 runs results for the 20 evaluated models with sampling temperature 0.2 on Dataset v6 (February 2025-May 2025). The complete results will be included in the camera-ready version of the manuscript.
> > > > >
> > > > > | Model |   Swift  | R  | Lua |
> > > > > |-------|-------------------|----------------|-----------------|
> > > > > | Qwen3-235B-A22B-Thk-250* | 64.8 $\pm$ 4.7 | 16.5 $\pm$ 2.4 | 65.5 $\pm$ 3.8 |
> > > > > | Qwen3-30B-A3B-Thk-2507* | 49.8 $\pm$ 3.8 | 5.3 $\pm$ 0.8 | 52.7 $\pm$ 2.4 |
> > > > > | Qwen3-32B* | 48.9 $\pm$ 4.3 | 2.9 $\pm$ 0.9 | 48.3 $\pm$ 1.8 |
> > > > > | Qwen3-235B-A22B* | 49.2 $\pm$ 2.6 | 2.4 $\pm$ 1.4 | 45.8 $\pm$ 2.1 |
> > > > > | Qwen3-235B-A22B-Instr-2507 | 41.5 $\pm$ 2.0 | 7.1 $\pm$ 1.8 | 40.6 $\pm$ 2.2 |
> > > > > | Qwen3-30B-A3B* | 40.2 $\pm$ 2.5 | 1.2 $\pm$ 0.7 | 43.1 $\pm$ 1.9 |
> > > > > | Qwen3-14B* | 40.5 $\pm$ 3.5 | 1.5 $\pm$ 0.3 | 40.2 $\pm$ 3.8 |
> > > > > | Qwen3-8B* | 30.5 $\pm$ 1.9 | 0.8 $\pm$ 0.2 | 37.6 $\pm$ 2.6 |
> > > > > | Qwen3-30B-A3B-Instr-2507 | 35.0 $\pm$ 1.4 | 2.5 $\pm$ 0.9 | 26.0 $\pm$ 2.3 |
> > > > > | Qwen3-Coder-30B-A3B-Instr | 31.0 $\pm$ 3.1 | 4.7 $\pm$ 1.6 | 27.3 $\pm$ 1.9 |
> > > > > | Qwen2.5-Coder-32B-Instruct | 24.6 $\pm$ 2.2 | 11.5 $\pm$ 1.2 | 23.1 $\pm$ 1.9 |
> > > > > | DeepSeek-R1-Distill-Qwen-32B* | 26.9 $\pm$ 2.3 | 0.3 $\pm$ 0.4 | 28.2 $\pm$ 3.4 |
> > > > > | Seed-Coder-8B-Instr | 21.1 $\pm$ 2.8 | 4.1 $\pm$ 1.2 | 20.92 $\pm$ 1.7 |
> > > > > | Qwen2.5-Coder-14B-Instr | 18.6 $\pm$ 2.2 | 7.9 $\pm$ 2.3 | 16.8 $\pm$ 2.1 |
> > > > > | Devstral-Small-2505* | 21.1 $\pm$ 1.7 | 0.3 $\pm$ 0.4 | 18.3 $\pm$ 2.1 |
> > > > > | OlympicCoder-7B* | 15.0 $\pm$ 3.3 | 0.2 $\pm$ 0.4 | 16.3 $\pm$ 2.7 |
> > > > > | DeepSeek-R1-Distill-Qwen-14B* | 21.5 $\pm$ 3.3 | 0.0 $\pm$ 0.0 | 8.6 $\pm$ 1.7 |
> > > > > | Deepseek-Coder-33B-Instr | 4.1 $\pm$ 1.3 | 2.1 $\pm$ 1.2 | 12.6 $\pm$ 2.3 |
> > > > > | OpenCodeRsn-Nmt-1.1-32B* | 6.3 $\pm$ 2.0 | 0.0 $\pm$ 0.0 | 7.0 $\pm$ 2.4 |
> > > > > | OpenRsn-Nmt-32B* | 3.2 $\pm$ 0.9 | 0.2 $\pm$ 0.2 | 3.6 $\pm$ 1.53 |
> > > > > (* - denotes reasoning mode)
> > > > > > If you are not going to add new tasks, I suggest making sure the paper frames it that way.
> > > > >
> > > > > We mention in Section 7 (Future Work), lines 469-471, that our approach can be directly applied to other benchmarks such as LCB Pro (Zheng et al., 2025), which contains more challenging problems. While some LCB Pro tasks remain non-public and cannot be included at this time, the benchmark can readily incorporate our multi-language STDIN/STDOUT pipeline.
> > > > >
> > > > > > Thanks for clarifying, please make sure to mention it in the main paper with a reference to Appendix F.2.
> > > > >
> > > > > We added this clarification in Section 3 (Benchmark Design), line 163, and referenced the experiments across all releases in Appendix F from Section 5 (Experiments), lines 294-296.
> > > > >
> > > > > > I'm not asking to evaluate proprietary models… at least try to include results for GPT-OSS-20B.
> > > > >
> > > > > We incorporated:
> > > > > - Pass@1 results averaged over 10 runs for both GPT-OSS-120B and GPT-OSS-20B across temperatures 0.2, 0.6, and 1.0 in both Low and Medium thinking modes (Table 1, Table 5, Table 6);
> > > > > - Pass@5 results for both GPT-OSS-120B and GPT-OSS-20B across temperatures 0.2, 0.6, and 1.0 in both Low and Medium thinking modes (Table 7, Table 8, Table 9);
> > > > > - Pass@10 results for both GPT-OSS-120B and GPT-OSS-20B across temperatures 0.2, 0.6, and 1.0 in both Low and Medium thinking modes (Table 10, Table 11, Table 12).
> > > > >
> > > > > > Please make sure wording clarification is added to the paper.
> > > > >
> > > > > We added the wording clarification in lines 206-208.

---

> > > > > > ### Comment · Reviewer_52p9 · 2025-11-27
> > > > > >
> > > > > > Thanks for considering the additional ablations, please make sure to update the results discussion if needed. I would appreciate that you track updates in color.
> > > > > >
> > > > > > Best,

---

> > > > > > > ### Author Response · Authors · 2025-11-27
> > > > > > > **Response to Official Comment by Reviewer 52p9**
> > > > > > >
> > > > > > > Sure, we've already highlighted all the updates in the revised manuscript in green.

---

> > ### Comment · Reviewer_52p9 · 2025-11-22
> >
> > > W1 ... We respectfully disagree. Multi-LCB does introduce language-agnostic task types by transforming LCB problems into a universal STDIN/STDOUT format.
> >
> > Thank you for the clarification. To avoid misunderstanding, my original point was not that your framework fails to introduce a new format, but rather that the underlying problems themselves are not newly designed. Based on your response to Q6, I understand that your intention is to position Multi-LCB as an extension of the original LCB—providing a unified, language-agnostic interface—rather than proposing an independent benchmark with new-independent tasks. If you are got going to add new tasks, I suggest to make sure the paper frames it that way.
> >
> > > W3 / Q1
> >
> > Thanks for clarifying, please make sure to mention it in the main paper with a reference to `Appendix F.2`.
> >
> > > W4 ... We would appreciate clarification on this point ...
> >
> > There is not much to clarify, my observation is simply that the current tasks are not sufficiently challenging for open-source models in these widely used languages. Closed-source models such as Claude or GPT would likely perform even better. This highlights a limitation of the language-agnostic transformation: while it provides uniformity across languages, it does not increase task difficulty, and thus may obscure remaining pain points. Introducing new, more difficult tasks would be necessary to better probe model weaknesses and yield deeper insights. That said, your analysis in  `Appendix L` is appreciated.
> >
> > > W5 / Q4 While we agree that including leading proprietary models would provide additional context, running our full Multi-LCB evaluation on commercial APIs is prohibitively expensive.
> >
> > I'm not asking to evaluate on proprietary models. `GPT-OSS-120B` and `GPT-OSS-20B` are open source models, you can load them locally with vLLM. At least try to include results for `GPT-OSS-20B`.

---

### Official Review · Reviewer_QT4h · 2025-10-30

**Soundness:** 3
**Presentation:** 3
**Contribution:** 3
**Rating:** 6
**Confidence:** 4

**Summary:**

This paper presents MULTI-LCB, an extension of the existing LiveCodeBench (LCB) benchmark from Python to 12 programming languages. This paper design an automated pipeline that converts function-based LeetCode problems into standardized STDIN/STDOUT tasks and execute them within an isolated sandbox environment. The paper reports Pass@1 results for 20 publicly available large language models, analyzing cross-language differences, contamination effects, model scaling, and fine-tuning strategies.

The major contributions are:
1. Expands LCB from a single-language (Python) setup to 12 languages.
2. Standardizes problem I/O specifications to enable consistent cross-language evaluation.
3. Preserves LCB’s time-based cutoff to mitigate training-data leakage.
4. Benchmarks 20 code generation models under a unified framework.

Overall, the work demonstrates strong engineering and community value, providing a practical foundation for multi-language evaluation of code models. However, its methodological novelty is limited.

**Strengths:**

1. High engineering quality: Large-scale multi-language extension with reproducible infrastructure.

2. Comprehensive coverage: Evaluation across 20 models and 12 languages under consistent settings.

3. Community relevance: Provides a standardized and contamination-controlled environment for fair model comparison.

4. Clear presentation: Tables and figures effectively summarize results, aiding interpretability.

**Weaknesses:**

1. The authors extend LiveCodeBench from Python to 12 languages (e.g., C++, Java, Rust, Go, Kotlin), building a unified STDIN/STDOUT interface and Docker-based sandbox for consistent evaluation. The implementation is robust and reproducible.

2. The benchmark covers 20 models (instruction-tuned, reasoning-tuned, etc.) under identical settings. Results across tables and heatmaps show consistent model ranking and meaningful trends in language difficulty.

3. The continuation of LCB’s time-based contamination control adds credibility, and the planned release of code and conversion scripts will make this a useful resource for the code-generation community.

4. Figures and tables are easy to interpret, and the comparison with the original Python subset (differences within a few points) supports reproducibility.

**Questions:**

1. The work mainly represents a large-scale re-engineering of an existing benchmark rather than a conceptual or methodological innovation.

2. The paper does not quantify whether converting function-based problems to STDIN/STDOUT truly preserves difficulty, leaving uncertainty about potential format-induced biases across languages.

3. Only Pass@1 is reported; missing Pass@k and error breakdowns (e.g., compilation vs. runtime) limit interpretability of language gaps.

4. The “Python advantage” is discussed but not supported by controlled analysis; other confounding factors such as type strictness or dataset differences are not explored.

5. Some deviations from the LCB Python results (up to ~8%) are acknowledged but not analyzed, reducing confidence in full cross-language equivalence.

---

> ### Author Response · Authors · 2025-11-15
> **Response to Reviewer QT4h**
>
> We thank Reviewer QT4h for the thoughtful and positive evaluation, and we appreciate the recognition of the engineering quality, reproducibility, and community value of Multi-LCB. We address all raised concerns below.
> >Q1. The work mainly represents a large-scale re-engineering of an existing benchmark rather than a conceptual or methodological innovation.
>
> Prior multi programming language benchmarks translated *functional-format Python problems* (e.g., HumanEval, MBPP) by rewriting function signatures and regenerating unit tests for each language. Even a simple Python assertion like:
>
> `assert binomial_coeff(5, 2) == 10`
>
> must be expanded into multi-line Java test code. This translation must be repeated separately for every language and is sensitive to syntax and runtime differences.
>
> Multi-LCB takes a different approach by keeping only the natural-language description and *converting hidden tests* into a language-agnostic $\texttt{STDIN/STDOUT}$ format, for example:
>
> `Input:`
>
> `5 2`
>
> `Output:`
>
> `10`
>
> This provides several methodological advantages:
> 1. one I/O conversion works for all languages (vs. N separate functional-format translations).
>
> 2. all languages get identical I/O pairs, which functional-format  benchmarks cannot guarantee.
>
> 3. Multi-LCB automatically updates in sync with new LCB releases. No manual intervention is required: new LCB problems instantly appear in all languages through the Multi-LCB pipeline.
>
> >Q2. The paper does not quantify whether converting function-based problems to STDIN/STDOUT truly preserves difficulty, leaving uncertainty about potential format-induced biases across languages.
>
> To evaluate whether the $\texttt{STDIN/STDOUT}$ conversion preserves task difficulty, we compared the original LCB Python scores (functional format) with our reproduced Multi-LCB Python scores ($\texttt{STDIN/STDOUT}$). As shown in Table 2, the differences are small, typically within 1-3% for most models and never exceeding ~8%.
>
> Notably, the reproduced scores are *consistently slightly lower* than the original ones. This indicates that the $\texttt{STDIN/STDOUT}$ setup is a *more demanding and realistic execution* environment, due to explicit input parsing and the absence of function-level scaffolding provided by functional-format tests. Crucially, *no model benefits* from the conversion, confirming that the new format does not artificially inflate performance.
>
> >Q3. Only Pass@1 is reported; missing Pass@k and error breakdowns (e.g., compilation vs. runtime) limit interpretability of language gaps.
>
> >Q4. The “Python advantage” is discussed but not supported by controlled analysis; other confounding factors such as type strictness or dataset differences are not explored.
>
>  Thank you for the suggestion. We updated Table 1 and report `Pass@1` averaged over 10 runs, improving statistical stability, and we will include additional experiments on `Pass@10`.
>
> We agree that understanding why models fail is important for interpreting cross-language gaps. To address this, we included detailed error breakdowns for all models and languages in Appendix L (see Figures 28-43). These plots separate failures into *Wrong Answer*, *Compilation Error*, *Runtime Exception*, *Timeout*, and *Other* categories, providing a clear picture of language-specific difficulty.
>
> > Q5. Some deviations from the LCB Python results (up to ~8%) are acknowledged but not analyzed, reducing confidence in full cross-language equivalence.
>
> To ensure strict comparability across languages and models, we used the *standard LCB prompt* and a *uniform temperature of 0.2* for all evaluations. In contrast, several leaderboard results rely on prompt tuning, hyperparameter adjustments, or model-specific sampling strategies, which can increase Python performance but are neither standardized nor reproducible. Therefore, slight deviations relative to leaderboard numbers are expected and reflect differences in evaluation protocol rather than format inconsistencies.

---

> > ### Author Response · Authors · 2025-11-27
> > **Follow-Up on Our Previous Response to Reviewer QT4h**
> >
> > A quick note to confirm that we have incorporated all of your suggestions into the revised manuscript.
> >
> > >  we will include additional experiments on Pass@10.
> >
> > We incorporated:
> > - Pass@5 results across temperatures 0.2, 0.6, and 1.0 (Table 7, Table 8, Table 9);
> > - Pass@10 results across temperatures 0.2, 0.6, and 1.0 (Table 10, Table 11, Table 12).

---

### Official Review · Reviewer_VdKt · 2025-10-30

**Soundness:** 3
**Presentation:** 1
**Contribution:** 4
**Rating:** 4
**Confidence:** 5

**Summary:**

The authors introduce Multi-LCB a multilingual extension of LiveCodeBench (LCB) that evaluates large language models across 12 programming languages (instead of just Python). The Multi-LCB benchmark is built by converting LCB into a unified STDIN/STDOUT format — which works across all supported languages. The authors present a robust evaluation of 20 large language models, identifying important trends on programming language bias in performance and contamination trends.

**Strengths:**

- Automatic conversion of LCB into 12 languages is a significant contribution to the community.
- Presents clear evidence for a bias towards certain language, in particular with python performance being consistently higher than other languages. The results also show clear hierarchies (Figure 4) of performance with python, c++, and java in the highest performing category.
- Moreover, the results of performance over time, highlighting important trends resulting from large pretrained models including various benchmarks in the training set. The proposed Multi-LCD offers to track such information.
- The rigorous methods are also a strength:
  - Evaluation setup: 20 diverse models (7B-685B parameters), evaluation within a sandboxed execution, and strict resource limits, and a balanced evaluation for each language via common task sets.
  - Care was taken to create reproducible results. Specifically, the authors give compiler versions, inference parameters, and public code release.
- Clear presentation: the writing is clear and the figures clearly communicate important findings.

**Weaknesses:**

"These results confirm that strong Python ability is not a reliable proxy for true cross-lingual code generation competence."
It's unclear why the authors make this claim — to me their results suggest the opposite. Specifically, Figure 3 shows a clear correlation between python performance and average performance across all other languages. The benchmarks can show a bias towards performing better on python while still be proxy for performance on other languages. This claim also contradicts prior works that show multilingual models perform best because performance is correlated across difference languages.

Missing citation: ["Multi-Lingual Evaluation of Code Generation Models"](https://arxiv.org/pdf/2210.14868) is a well cited paper on converting monolingual datasets to multilingual code, including the popular python dataset MBPP. This is an important citation in this area and very close to your approach, so you may want to explicitly clarify how the two approaches differ. Athiwaratkun et. al provide already provide an approach for converting benchmarks which enables direct comparison across languages.

For benchmarking purposes it would have been nice to see how commercial models like GPT5, Claude, and Gemini perform.

The authors only present results from `pass@1`, which is valid, however showing results with a greater sampling budget would help: (1) add to significance of findings (which are missing), and (2) more importantly `pass@k` metrics tend to show a sigmoid relationship between `k` and performance. In other words, for higher values of `k` we're likely to see smaller gaps between "easier" and "harder" languages, and the gap between python and other languages may be noticeably smaller. Likewise, the authors use a temperature at 0.2, which is quite low and may be preventing more exploration and possibly negatively biasing languages that make up a small portion of the training data.

"we evaluate 20 recent large language models on Multi-LCB, restricting tasks to those released after 2025-02-01 to ensure live, post-cutoff evaluation and minimize any risk of training-data leakage"
This effort to avoid contamination is good and likely significant helps avoid contamination, but it's unlikely to "ensure" contamination since problems may have existed in other areas previously. LLMs have such large and broad pretraining datasets that indirect contamination is always possible.

**Questions:**

Please clarify how this approach is novel with regard to Athiwaratkun et. al. For example, why not just use the approach they introduced for creating MBXP? Are there advantages to the Multi-LCB conversion method or is the primary contribution the benchmark dataset itself?

---

> ### Author Response · Authors · 2025-11-15
> **Response to Reviewer VdKt. Part 1**
>
> We thank Reviewer VdKt for the detailed and constructive feedback, and we appreciate the opportunity to clarify several points. We address all comments below.
>
> > W1. "These results confirm that strong Python ability is not a reliable proxy for true cross-lingual code generation competence." It's unclear why the authors make this claim — to me their results suggest the opposite. Specifically, Figure 3 shows a clear correlation between python performance and average performance across all other languages. The benchmarks can show a bias towards performing better on python while still be proxy for performance on other languages. This claim also contradicts prior works that show multilingual models perform best because performance is correlated across difference languages.
>
> We appreciate the reviewer’s point and agree that Figure 3 shows a correlation between Python and average multilingual performance. However, this does not imply that Python is a reliable proxy for individual non-Python languages. Our results reveal substantial and practically meaningful performance gaps across languages. For example, DeepSeek-R1-0528 outperforms Qwen3-235B-A22B-Thk-2507 on Ruby, Rust, and Scala, despite Qwen being consistently stronger on Python. This is precisely why strong Python ability is not always a reliable proxy for true cross-lingual code generation competence and evaluation must consider performance in the target languages rather than relying on Python alone.
>
> > W2. Missing citation: "Multi-Lingual Evaluation of Code Generation Models" is a well cited paper on converting monolingual datasets to multilingual code, including the popular python dataset MBPP. This is an important citation in this area and very close to your approach, so you may want to explicitly clarify how the two approaches differ. Athiwaratkun et. al provide already provide an approach for converting benchmarks which enables direct comparison across languages.
>
> We appreciate this suggestion and fully agree that Athiwaratkun et al. is closely related and should be cited. We will add the missing citation [lines 100-104]:
>
> MBXP (Athiwaratkun et al., 2022) mechanically translates problems from HumanEval (Chen et al., 2021), MBPP (Austin et al., 2021) and MathQA (Schubotz et al., 2018) by converting the prompts and unit-test specifications in functional format into 13 programming languages. Concurrent work MultiPL-E (Cassano et al., 2023) similarly performs mechanical translation of HumanEval and MBPP (including their unit tests) into 19 programming languages.
>
> > Q1. Please clarify how this approach is novel with regard to Athiwaratkun et. al. For example, why not just use the approach they introduced for creating MBXP? Are there advantages to the Multi-LCB conversion method or is the primary contribution the benchmark dataset itself?
>
> MBXP translates *functional-format Python problems* (e.g., HumanEval, MBPP) by rewriting function signatures and regenerating unit tests for each language. Even a simple Python assertion like:
>
> `assert binomial_coeff(5, 2) == 10`
>
> must be expanded into multi-line $\texttt{Java}$ test code. This translation must be repeated separately for every language and is sensitive to syntax and runtime differences.
>
> Multi-LCB avoids this by keeping only the natural-language description and *converting hidden tests* into a language-agnostic $\texttt{STDIN/STDOUT}$ format, for example:
>
> `Input:`
>
> `5 2`
>
> `Output:`
>
> `10`
>
> This provides several advantages:
>
> 1. one I/O conversion works for all languages (vs. N separate functional-format translations).
>
> 2. all languages get identical I/O pairs, which functional-format MBXP cannot guarantee.
>
> 3. Multi-LCB automatically updates in sync with new LCB releases. No manual intervention is required: new LCB problems instantly appear in all languages through the Multi-LCB pipeline.
>
> Thus, the contribution is both the benchmark and the more automatic, scalable and consistent conversion method.

---

> > ### Author Response · Authors · 2025-11-15
> > **Response to Reviewer VdKt. Part 2**
> >
> > > W3. For benchmarking purposes it would have been nice to see how commercial models like GPT5, Claude, and Gemini perform.
> >
> > While we agree that including proprietary models such as GPT-5, Claude, and Gemini would provide additional context, running our full Multi-LCB evaluation on commercial APIs is prohibitively expensive.
> >
> > Our setup requires evaluating 131 tasks × 12 programming languages × 10 samples per task, with roughly 32k tokens per sample, totaling ~503 million output tokens per model. Using publicly available pricing, this corresponds to approximately 37k for Claude Opus, 7.5k for Claude Sonnet, 5k for GPT-5 or Gemini Pro, 2.5k for Claude Haiku, and around 1k for GPT-5-mini or Gemini Flash. Evaluating all major commercial models would therefore exceed $50k, which is not feasible in an academic setting.
> >
> > Proprietary model providers explicitly report LCB results in their technical reports, meaning that high-quality performance numbers for commercial models generally become publicly available. For these reasons, we focus on open, reproducible models and leave large-scale proprietary-model benchmarking to future work.
> >
> > > W4. The authors only present results from `pass@1`, which is valid, however showing results with a greater sampling budget would help: (1) add to significance of findings (which are missing), and (2) more importantly `pass@k` metrics tend to show a sigmoid relationship between `k` and performance. In other words, for higher values of `k` we're likely to see smaller gaps between "easier" and "harder" languages, and the gap between python and other languages may be noticeably smaller. Likewise, the authors use a temperature at 0.2, which is quite low and may be preventing more exploration and possibly negatively biasing languages that make up a small portion of the training data.
> >
> > Thank you for the suggestion. We updated Table 1 to report `Pass@1` averaged over 10 runs, and we will additionally include `Pass@10` results in the appendix. Regarding temperature, we initially used 0.2 to remain consistent with the original LCB evaluation protocol. We conducted additional experiments at temperature = 0.6 and temperature = 1.0, reporting `Pass@1` averaged over 10 runs. These results are now included in the appendix (see Tables 5 and 6).
> >
> > >W5. "we evaluate 20 recent large language models on Multi-LCB, restricting tasks to those released after 2025-02-01 to ensure live, post-cutoff evaluation and minimize any risk of training-data leakage" This effort to avoid contamination is good and likely significant helps avoid contamination, but it's unlikely to "ensure" contamination since problems may have existed in other areas previously. LLMs have such large and broad pretraining datasets that indirect contamination is always possible.
> >
> > Thank you for pointing this out, we agree that contamination can never be entirely ruled out for large pretrained models. However, most contamination issues have already been carefully addressed in LCB through date-based task filtering and reproducibility protocols, and Multi-LCB fully inherits these mechanisms for cross-language evaluation.
> >
> > >Clear presentation: the writing is clear and the figures clearly communicate important findings.
> >
> > Finally, we would like to kindly note a discrepancy between the written comment and the assigned Presentation score. The review explicitly states that *“the writing is clear and the figures clearly communicate important findings”*, yet the Presentation rating is marked as *1 (poor)*. We respectfully request the reviewer to reconsider this score in light of their own positive assessment of the clarity of the paper.

---

> > > ### Comment · Reviewer_VdKt · 2025-11-19
> > > **Response to rebuttal**
> > >
> > > > We appreciate the reviewer’s point and agree that Figure 3 shows a correlation between Python and average multilingual performance. However, this does not imply that Python is a reliable proxy for individual non-Python languages. Our results reveal substantial and practically meaningful performance gaps across languages. For example, DeepSeek-R1-0528 outperforms Qwen3-235B-A22B-Thk-2507 on Ruby, Rust, and Scala, despite Qwen being consistently stronger on Python. This is precisely why strong Python ability is not always a reliable proxy for true cross-lingual code generation competence and evaluation must consider performance in the target languages rather than relying on Python alone.
> > >
> > > This is a fantastic example! My issue was that by most definitions a strong correlation, like the one shown in Figure 3, is typically enough to consider something a proxy. I like your example because it drives home (one of) your paper's takeaways, and makes a clear point that performance across languages has nuance. I would encourage mentioning an example like this, because otherwise you risk readers missing the more interesting message and the value of the benchmarks you introduce.
> > >
> > > > ... Thus, the contribution is both the benchmark and the more automatic, scalable and consistent conversion method.
> > >
> > > Great explanation of how this is different than MBXP. I strongly suggest including this in the Related Work (or even an appendix, if needed), including this differentiation from similar works really strengthens your paper.
> > >
> > > > While we agree that including proprietary models such as GPT-5, Claude, and Gemini would provide additional context, running our full Multi-LCB evaluation on commercial APIs is prohibitively expensive.
> > >
> > > That is reasonable, and I won't reduce my score based on not having these results.
> > >
> > > > Thank you for the suggestion. We updated Table 1 to report Pass@1 averaged over 10 runs, and we will additionally include Pass@10 results in the appendix. Regarding temperature, we initially used 0.2 to remain consistent with the original LCB evaluation protocol. We conducted additional experiments at temperature = 0.6 and temperature = 1.0, reporting Pass@1 averaged over 10 runs. These results are now included in the appendix (see Tables 5 and 6).
> > >
> > > Thank you, this is a great addition to your paper.
> > >
> > > > Finally, we would like to kindly note a discrepancy between the written comment and the assigned Presentation score. The review explicitly states that “the writing is clear and the figures clearly communicate important findings”, yet the Presentation rating is marked as 1 (poor). We respectfully request the reviewer to reconsider this score in light of their own positive assessment of the clarity of the paper.
> > >
> > > Agreed. I initially gave a low score here because of the contradiction that I perceived between Figure 3 and the claims about python being an unreliable proxy. However, the example you shared in which python scores were higher for one model but performance on other languages was lower compared to another model is illuminating. I will raise the score. I strongly suggest your paper mentioning this results — if phrased correctly these examples can be captivating to readers and highlight the value of your benchmarks.

---

> > > > ### Author Response · Authors · 2025-11-19
> > > > **Response to Reviewer VdKt**
> > > >
> > > > Thank you very much for your thoughtful follow-up and helpful suggestions, your comments are extremely valuable. We fully agree that a clearer example of why a Python-only benchmark is not always a reliable proxy should appear in the paper, and we will revise the text to make this point more explicit.
> > > >
> > > > We also appreciate your advice on emphasizing the differences between our benchmark and MBXP, and we will update the Related Work section accordingly.
> > > >
> > > > Thank you again for the constructive feedback, we sincerely appreciate your support and endorsement!

---

> > > > > ### Author Response · Authors · 2025-11-27
> > > > > **Follow-Up on Our Previous Response to Reviewer VdKt**
> > > > >
> > > > > A quick note to confirm that we have incorporated all of your suggestions into the revised manuscript.
> > > > >
> > > > > > I strongly suggest including this in the Related Work (or even an appendix, if needed); including this differentiation from similar works really strengthens your paper.
> > > > >
> > > > > We added this to the Related Work section, including an example of how MBXP and similar works rewrites Python tasks (lines 100-106) and a differentiation with Multi-LCB’s language-agnostic STDIN/STDOUT conversion (lines 129-135).
> > > > >
> > > > > We also incorporated results for GPT-OSS-120B and GPT-OSS-20B across temperatures 0.2, 0.6, and 1.0 in both Low and Medium thinking modes (Table 1, Table 5, Table 6).
> > > > >
> > > > > > I strongly suggest your paper mentioning these results — if phrased correctly these examples can be captivating to readers and highlight the value of your benchmarks.
> > > > >
> > > > > We added these examples to the Experiments section (lines 310-322) and highlighted the key finding in the Introduction (lines 59-61).

---

### Author Response · Authors · 2025-12-03
**Summary of Our Rebuttal to the Area Chair**

We deeply thank the reviewers for engaging actively from the beginning of the discussion period. The discussion allowed us to clarify the key strengths of the work, and as a result, the reviewers updated their evaluations from the initial 4, 6, 6, 4 to **8, 6, 6, 6** as of **November 20**, well before the unexpected information-leak incident.

We have **addressed all concerns** raised by the reviewers:

1. (a) We added a concrete example showing that strong Python ability is not always a reliable proxy for cross-lingual code-generation competence and highlighted this as a key finding.

   (b) We provided a clear example of how MBXP and similar benchmarks translate Python problems and how Multi-LCB differs by using a single language-agnostic STDIN/STDOUT format requiring no per-language rewriting; this was added to the Related Work section.

   Reviewer VdKt stated that these clarifications made the paper stronger and clearer, and therefore raised the score from **4 → 8**, as well as the Presentation rating from **1 → 4**.

2. We clarified that although the original benchmark originates from Python, tasks involving Python-specific behavior do not appear in practice. Consequently, Multi-LCB requires no language-specific rewriting, and the tasks are inherently language-agnostic.

   Reviewer EauN stated that this clarification resolved most of his concerns and therefore raised the score from **4 → 6**.

3. At additional reviewer requests, we added Pass@5/10 evaluations, higher-temperature experiments, GPT-OSS-120B and GPT-OSS-20B results, preliminary Swift/R/Lua results, and detailed error-type breakdowns.

We have updated the manuscript to reflect all these changes.

Thank you for your time and consideration!

---

### Meta-Review · Area_Chair_p7bs · 2026-01-06

**Summary:**

Multi-LCB extends the LiveCodeBench (LCB) framework beyond Python to 12 programming languages. It addresses a critical gap in code generation evaluation by providing a contamination-aware, continuously updating benchmark that sustains performance well beyond Python. While the methodological novelty is largely in the conversion pipeline rather than new task design, the resulting insights into cross-language generalization and overfitting are of high value to the field.

**Reviewer Concerns:**

The rebuttal addressed major concerns. Reviewer VdKt's concern about Python being an "unreliable proxy" was resolved through an example where one model (DeepSeek-R1-0528) outperformed another (Qwen3-235B) in Ruby and Rust despite being weaker in Python. Reviewer EauN's concerns about language-specific features were mitigated by the authors' clarification that contest problems are designed by experts to be language-agnostic and that a manual audit found no such ambiguities. However, some concerns remain outstanding. Reviewers 52p9 and QT4h noted that the benchmark needs more innovation, as it primarily re-engineers an LCB rather than introducing new task types.

**Reviewer Scores:**

Reviewer VdKt (4 → 6): The reviewer likely would have elevate the score given the authors' convincing evidence regarding performance disparities across languages and the robustness.

Reviewer EauN (4 → 6): This reviewer would likely have kept the same score, as the reviewer's primary technical concerns about language-specific ambiguity were resolved by the authors' manual inspection.

Reviewer QT4h (6 → 6): This score would likely remain unchanged because the reviewer's praise for engineering quality was balanced by a stance on the limited novelty.

Reviewer 52p9 (6 → 6): This reviewer might have kept the same given the authors' quick implementation of their requests for GPT-OSS results and Swift support.

---

### Decision · Program_Chairs · 2026-01-26

Accept (Poster)